# Understanding Post-Training Structural Changes in Large Language Models

## Abstract

Post-training fundamentally alters the behavior of large language models (LLMs), yet its impact on the internal parameter space remains poorly understood. In this work, we conduct a systematic singular value decomposition (SVD) analysis of principal linear layers in pretrained LLMs, focusing on two widely adopted post-training methods: *instruction tuning* and *long-chain-of-thought (Long-CoT) distillation*. Our analysis reveals two consistent and unexpected structural changes: **(1) a near-uniform geometric scaling of singular values across layers**, which theoretically modulates attention scores; and **(2) highly consistent orthogonal transformations are applied to the left and right singular vectors of each matrix.** Disrupting this orthogonal consistency leads to catastrophic performance degradation. Based on these findings, we propose a simple yet effective framework that interprets post-training as a reparameterization of fixed subspaces in the pretrained parameter space. Further experiments reveal that singular value scaling behaves as a secondary effect, analogous to a temperature adjustment, whereas the core functional transformation lies in the coordinated rotation of singular vectors. These results challenge the prevailing view of the parameter space in large models as a black box, uncovering the first clear regularities in how parameters evolve during training, and providing a new perspective for deeper investigation into model parameter changes.

## 1 Introduction

The remarkable success of large language models (LLMs) has been substantially facilitated by post-training techniques. With approaches such as instruction tuning (Ouyang et al., 2022; Zhang et al., 2024b; Peng et al., 2023), alignment training (Schulman et al., 2017; Li et al., 2023b; Rafailov et al., 2024; DeepSeek-AI et al., 2025) and knowledge distillation (Xu et al., 2024; Gu et al., 2024; McDonald et al., 2024; Yang et al., 2024), LLMs have become increasingly usable and better aligned with human intent (Guo et al., 2024; Cai et al., 2025; Feng et al., 2024). Recent research on post-training has predominantly centered on algorithmic innovations such as *Direct Preference Optimization* (DPO) (Rafailov et al., 2024), *Group Relative Policy Optimization* (GRPO) (DeepSeek-AI et al., 2025), and *Dynamic sAmpling Policy Optimization* (DAPO) (Yu et al., 2025) to enhance the reasoning capabilities of LLMs. Alternatively, *long-chain-of-thought (Long-CoT) distillation* offers a more straightforward and practiced approach, enabling smaller models to acquire reasoning ability by distilling long chains of thought from large RL-trained models (DeepSeek-AI et al., 2025).

However, despite the empirical success of post-training, its underlying impact on the internal structure of model parameters remains insufficiently understood. Although recent studies have investigated post-training mechanisms and uncovered some novel insights (Du et al., 2025; Marks & Tegmark, 2024; Jain et al., 2024; Lee et al., 2024; Panickssery et al., 2024; Stolfo et al., 2024; Katz & Belinkov, 2023; Yao et al., 2025), their studies remain indirect—relying primarily on hidden representations or behavioral observations rather than exploring fundamental structural changes. Transformations in parameter space, especially weight matrices, which we often treat as black boxes, have not been systematically examined. **The extent to which post-training reshapes the representational capacity of the parameter space remains an unresolved problem.**

In this work, we present a systematic study on how post-training affects the parameter space of LLMs. Specifically, we focus on two token-level supervised post-training methods: **instruction tuning** and

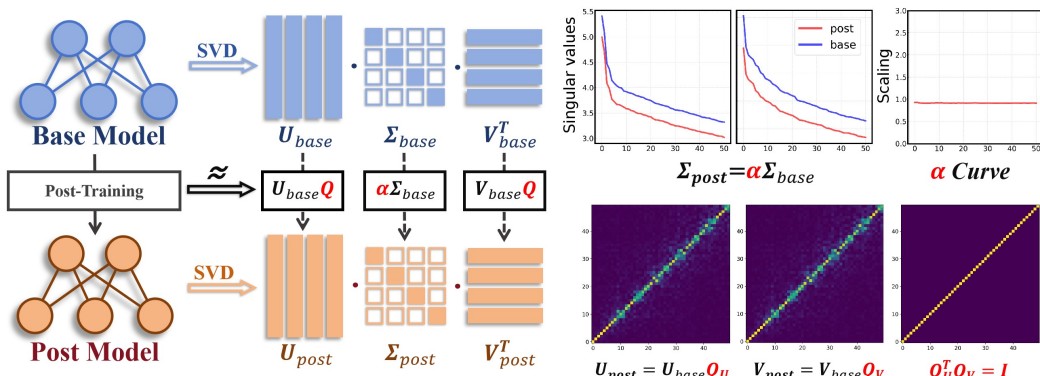

Figure 1: A simple but effective mathematical approximation to describe the effect of post-training on the parameter space. Performing SVD on weight matrices in the BASE model, post-training is equivalent to performing **linear scaling** on singular values and performing **consistent orthogonal transformations** on left and right singular vectors.

**Long-CoT distillation**[1]. These methods underpin essential capabilities like instruction-following and reasoning, forming the basis for more advanced alignment techniques. To examine the structural impact of post-training, we analyze weight matrices using singular value decomposition (SVD). **SVD decomposes each matrix into orthogonal subspaces with distinct scaling factors, thereby reducing complex weight structures into three mathematically interpretable components for systematic analysis, making the underlying geometry of large model parameters more transparent and interpretable.** We apply this framework to the weight matrices within the Self-Attention modules and Feed-Forward Networks of publicly available models, and categorize models into three types: BASE models (e.g., *Qwen2.5-Math-1.5B* (Qwen et al., 2025)), INSTRUCT models (obtained through instruction tuning), and REASONING models (trained via long-CoT distillation, such as *DeepSeek-R1-Distill-Qwen-1.5B* (DeepSeek-AI et al., 2025)). The latter two are collectively referred to as POST models. This categorization enables systematic comparison of parameter space structural changes induced by different post-training methods.

Our empirical results reveal two key effects of post-training on the model's parameter space: **(1) Near-uniform geometric scaling of singular values**: Post-training preserves the principal singular value distribution of the BASE model while applying a consistent, layer-wise linear scaling factor. We show this scaling equivalently regulates attention scores. Notably, we observe anomalous scaling in the Attention module's $W_O$ matrix, which strongly correlates with the REASONING model's superior long-chain reasoning over the INSTRUCT model; **(2) Highly consistent orthogonal transformations**: The left and right singular vectors of each matrix undergo nearly identical orthogonal transformations during post-training, exhibiting shared, coordinated rotations. This phenomenon occurs consistently across all weight matrices, **strongly suggesting that post-training preserves the subspace structure established during pre-training**.

These results indicate that post-training essentially induces highly regular structural perturbations in the parameter space. Based on the two observed phenomena, we can use a simple yet effective mathematical framework to directly approximate the impact of post-training on the parameter space (Figure 1). We experimentally demonstrate that the singular value scaling phenomenon is a temperature-controlled mechanism that does not alter the model's behavior. The consistent orthogonal transformations applied to the weight matrices are the core of post-training.

We summarize our contributions as follows:

- **To the best of our knowledge, this is the first systematic study of structural changes in LLMs before/after post-training across the entire parameter space.** Unlike prior works focusing on individual neuron activations or external behaviors, we comprehensively analyze the singular value structure of principal linear layers, revealing consistent patterns of post-training effects in the parameter space.

---

[1]For clarity and ease of reading, *post-training* hereafter refers to both *instruction tuning* and *Long-CoT distillation* in the following sections unless otherwise specified.

- **We experimentally discover two structural phenomena that are stable across multiple model families, parameter sizes, and training methods**: First, the singular values exhibit near-uniform geometric scaling; second, the left and right singular vectors of each matrix remain stable under consistent orthogonal transformations.

- **We establish a simple yet effective mathematical framework to describe the parameter change mechanism.** Our experiments have validated the importance of orthogonal transformations in post-training. This work provides new understanding of parameter evolution during post-training and lays the foundation for developing a unified theory of LLM parameter transformations.

## 2 RELATED WORK

**Interpretability of post-training.** With the growing success of post-training, researchers have increasingly sought to uncover its underlying mechanisms. Several studies have attempted to investigate the impact of post-training on LLMs by constructing task-specific or instruction-formatted datasets (Du et al., 2025; Marks & Tegmark, 2024; Jain et al., 2024; Lee et al., 2024; Panickssery et al., 2024; He et al., 2024). However, since these studies treat the models more as black boxes, they provide limited insights into the structural changes in model parameters induced by post-training. Parallel lines of research have attempted to explain the behavior of large language models by analyzing individual neurons or sparse activation patterns, uncovering phenomena such as entropy neurons and task-specific circuits (Stolfo et al., 2024; Katz & Belinkov, 2023; Yao et al., 2025; Gurnee et al., 2024; Tang et al., 2024; Chen et al., 2024; Yu & Ananiadou, 2024). While these studies offer valuable insights, their scope is inherently limited, as they are often based on earlier models such as *GPT-2* (Brown et al., 2020), reducing their relevance to contemporary architectures. Our analysis is data-agnostic, as we directly examine the full parameter space of the model rather than relying on input–output behavior. This perspective extends beyond previous studies that focus on individual neurons or isolated functional circuits, enabling a more global understanding of model structure.

**Singular value decomposition in large language models.** The optimal low-rank approximation property of SVD (Eckart & Young, 1936) has inspired a surge of SVD-based techniques for LLMs. Recent methods such as *PiSSA* (Meng et al., 2024), *SVFT* (Lingam et al., 2024) and *RaSA* (He et al., 2025) leverage dominant singular components to improve fine-tuning efficiency, while others employ SVD for quantization to reduce deployment costs (Li et al., 2024; Wang et al., 2024; Qinsi et al.; Li et al., 2023a; Yuan et al., 2023). Beyond its practical utility, SVD provides a principled framework for analyzing the structure of LLMs (Yang et al., 2023). For any weight matrix, reduced SVD produces a decomposition into two orthogonal matrices and a diagonal matrix, each of which carries a well-defined mathematical role: the orthogonal matrices span the input and output subspaces, defining bases in which the transformation operates, while the diagonal matrix applies directional scaling along these bases. In this view, the singular vectors determine how representations are aligned and projected, and the singular values quantify the relative importance of each direction. This decomposition reveals how LLMs transform information across layers, making SVD not only a tool for compression or fine-tuning, but also a window into the geometry of their internal computation. Our work leverages this perspective to investigate the structural organization of weights in LLMs.

## 3 PRELIMINARIES

This section reviews the training pipeline and architectural components of LLMs. Given a vocabulary $\mathcal{V}$, we define LLMs as $\mathcal{M} : \mathcal{T} \to \mathcal{P}$, where $\mathcal{T}$ denotes the set of input token sequences $T_i = [t_1, t_2, ..., t_n]_i \in \mathcal{T}$ and $\mathcal{P}$ is the probability space over $\mathcal{V}$. After $\mathcal{M}$ accepts sequences of input tokens $T_i$, a probability distribution $p_{\mathcal{M}} \in \mathcal{P}$ is output to predict the probability of the next token.

**Training stages of LLMs.** LLMs are typically trained following a two-stage paradigm. The first stage, known as pre-training, involves optimizing a BASE model $\mathcal{M}_{\text{base}}$ to predict the next token given previous context, based on a large-scale corpus drawn from a large-scale distribution of natural language texts (Radford et al., 2018; Sun et al., 2021; Yuan et al., 2022). The second stage, termed post-training, further fine-tunes the pretrained model to align its behavior with specific objectives, such as following user instructions (Zhang et al., 2024b) or performing complex reasoning (DeepSeek-AI et al., 2025). Depending on the post-training objective, the adapted model is referred

to as an INSTRUCT model $\mathcal{M}_{\text{Instruct}}$ or a REASONING model $\mathcal{M}_{\text{reasoning}}$. The two models under discussion are collectively referred to as POST models $\mathcal{M}_{\text{post}}$. The architectures of $\mathcal{M}_{\text{base}}$ and $\mathcal{M}_{\text{post}}$ are identical — all weight matrices share the same dimensionality, while the sole distinction lies in their respective parameterizations. In the main paper, $\mathcal{M}_{\text{base}}$ refers to *Qwen2.5-Math-1.5B*, $\mathcal{M}_{\text{Instruct}}$ to its instruction-tuned variant *Qwen2.5-Math-1.5B-Instruct*, and $\mathcal{M}_{\text{reasoning}}$ to the distilled *reasoning* model *DeepSeek-R1-Distill-Qwen-1.5B*. $\mathcal{M}_{\text{Instruct}}$ and $\mathcal{M}_{\text{reasoning}}$ can both be expressed as $\mathcal{M}_{\text{post}}$. Results for other models across different families and parameter scales are provided in the Appendix.

**Architectural components of LLMs.** We focus on decoder-only Transformer-based models, which constitute the foundation of state-of-the-art large language model systems (OpenAI et al., 2024; DeepSeek-AI et al., 2024a; Team et al., 2025). The Transformer architecture consists of two core components: the Self-Attention Module (SA) and the Feed-Forward Network (FFN) (Vaswani et al., 2023). Given an input hidden vector $h^T \in \mathbb{R}^{d_{model}}$, we consider the simplest form of attention calculation for concise illustration. The output of the SA is:

$$SA(h) = \text{softmax}\left(\frac{hW_Q \cdot [K_{\text{cache}}; hW_K]^T}{\sqrt{d}}\right) \cdot [V_{\text{cache}}; hW_V]W_O \tag{1}$$

where $W_Q, W_K, W_V, W_O \in \mathbb{R}^{d_{\text{model}} \times d_{\text{model}}}$ are learnable weight matrices, $\sqrt{d}$ is the scaling factor in the attention map, $K_{\text{cache}}$ and $V_{\text{cache}}$ are the key and value caches respectively, and $[...;...]$ denotes concatenation. While modern architectures such as *Qwen2.5* series adopt variants like GQA (Ainslie et al., 2023) to optimize attention computation, the core projection matrices remain integral to the design due to their role in defining the attention mechanism's representational capacity. Given an input vector $z^T \in \mathbb{R}^{d_{model}}$, the output of the FFN, which employs the *SwiGLU* activation function (Shazeer, 2020), is:

$$FFN(z) = (SwiGLU(z \cdot W_{gate}) \odot (z \cdot W_{up})) \cdot W_{down} \tag{2}$$

where $W_{down}^T, W_{gate}, W_{up} \in \mathbb{R}^{d_{model} \times d_{mlp}}$ are learnable weight matrices. Notably, GQA and SwiGLU-based FFNs have become fundamental building blocks adopted across numerous commercial open-source LLMs, including *Qwen* (Qwen et al., 2025), *LLaMA* (Grattafiori et al., 2024), *Mistral* (Jiang et al., 2023a), *Phi-4* (Abdin et al., 2024), *gpt-oss* (OpenAI et al., 2025), *Gemma* (Team et al., 2025) and others (GLM et al., 2024; Yang et al., 2025; DeepSeek-AI et al., 2024b). Since our work targets components common to mainstream architectures, their widespread adoption inherently ensures the generalizability and representativeness of our research focus. We specifically focus on the weight matrices in SAs and FFNs, which account for the majority of parameters in LLMs. Analyzing these linear layers further enables us to characterize the structure of the model's parameter space.

## 4 THE STRUCTURAL CHANGES OF SINGULAR SPACE AFTER POST-TRAINING

This section formally presents two regular structural changes that occur in the singular space of LLMs after post-training. Assuming that $m \leq n$, the reduced SVD of a matrix $W \in \mathbb{R}^{m \times n}$ is given by $W = U\Sigma V^T$, where $U \in \mathbb{R}^{m \times m}$ and $V^T \in \mathbb{R}^{m \times n}$ are matrices with orthogonality whose columns correspond to the left and right singular vectors respectively. The diagonal matrix $\Sigma = \text{diag}(\sigma_1, \sigma_2, \ldots, \sigma_n) \in \mathbb{R}^{n \times n}$ contains the singular values arranged in descending order.

### 4.1 NEAR-UNIFORM GEOMETRIC SCALING OF SINGULAR VALUES

We observe that post-training does not alter the overall singular value distribution established during pre-training in the BASE model, instead, It exhibits a near-uniform geometric scaling behavior, characterized by approximately consistent scaling factors across main singular values.

For the $i$-th Transformer block of $\mathcal{M}_A$ and $\mathcal{M}_B$ of the same architecture, we perform reduced SVD on weight matrix:

$$W_A^{(i)} = U_A^{(i)} \cdot diag(\sigma_{A,1}^{(i)}, \sigma_{A,2}^{(i)}, ..., \sigma_{A,n}^{(i)}) \cdot V_A^{(i)^T}$$
$$W_B^{(i)} = U_B^{(i)} \cdot diag(\sigma_{B,1}^{(i)}, \sigma_{B,2}^{(i)}, ..., \sigma_{B,n}^{(i)}) \cdot V_B^{(i)^T} \tag{3}$$

where $W_A^{(i)} \in \mathcal{M}_A$ and $W_B^{(i)} \in \mathcal{M}_B$ represent weight matrices of the same type in the $i$-th Transformer block (e.g. $W_Q$) but belonging to different models. To quantify the effect of post-training on the evolution of singular value distribution, we define the *Singular Value Scaling Matrix*

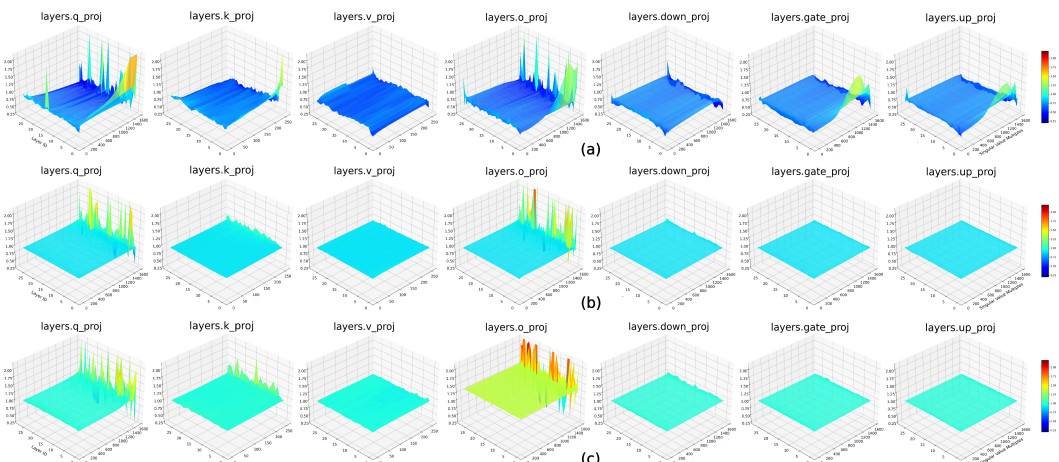

Figure 2: The heatmaps of SVSMs comparing $\mathcal{M}_{\text{base}}$ with $\mathcal{M}'_{\text{base}}$, $\mathcal{M}_{\text{Instruct}}$ and $\mathcal{M}_{\text{reasoning}}$. (a) indicates no regular pattern in the distribution of scaling factors between $\mathcal{M}'_{\text{base}}$ and $\mathcal{M}_{\text{base}}$. In both (b) and (c), the principal scaling exhibits a near-uniform distribution. While in (c), scaling factors of $W_O$ are significantly higher than those of other matrix types.

(SVSM) as:

$$SVSM(\frac{\mathcal{M}_B}{\mathcal{M}_A}) = [Div^{(1)}, Div^{(2)}, ..., Div^{(k)}], \quad Div^{(i)} = [\frac{\sigma_{B,1}^{(i)}}{\sigma_{A,1}^{(i)}}, ..., \frac{\sigma_{B,n}^{(i)}}{\sigma_{A,n}^{(i)}}]^T \tag{4}$$

where $k$ corresponds to the depth of architecture $\mathcal{M}_A$ or $\mathcal{M}_B$. $\alpha^{(i)} = \sigma_{B,j}^{(i)}/\sigma_{A,j}^{(i)}, j = 1, 2, ..., n$ is the scaling factor. SVSM actually describes the distribution of all scaling factors across layers. We plot the heatmaps of $SVSM(\frac{\mathcal{M}_{\text{Instruct}}}{\mathcal{M}_{\text{base}}})$ (Figure 2b) and $SVSM(\frac{\mathcal{M}_{\text{reasoning}}}{\mathcal{M}_{\text{base}}})$ (Figure 2c) as examples. For reference comparison, we also show heatmaps of $SVSM(\frac{\mathcal{M}'_{\text{base}}}{\mathcal{M}_{\text{base}}})$ where $\mathcal{M}'_{\text{base}}$ denotes *Qwen2.5-1.5B*, which shares the same architecture but differs in pre-training data (Figure 2a).

For $\mathcal{M}_{\text{Instruct}}$ and $\mathcal{M}_{\text{reasoning}}$ compared to $\mathcal{M}_{\text{base}}$, scaling factors are remarkably stable across principal singular values. The instability is confined to the tail, where the singular values have negligible magnitude and contribute little to the overall transformation. This phenomenon can be approximately modeled by $\Sigma_{\text{post}} \approx \alpha\Sigma_{\text{base}}$ since the scaling factors of principal singular values are almost the same. As a comparison, the cross-layer stability cannot be achieved between $\mathcal{M}'_{\text{base}}$ and $\mathcal{M}_{\text{base}}$. We further observe that scaling factors of $W_O$ in $\mathcal{M}_{\text{reasoning}}$ consistently exceed those of other matrix types, which can be used to significantly distinguish non-reasoning models. This pattern holds uniformly across all REASONING models in our study. Detailed quantitative data (Table 3) and visualizations of other models across different families and parameter scales are in Appendix A.

## 4.2 CONSISTENT ORTHOGONAL TRANSFORMATIONS OF SINGULAR VECTORS

We investigate the similarity between the singular vectors of BASE models and POST models. It is significant to find that the similarity matrices of both left and right singular vectors remain nearly identical after post-training, suggesting that the input and output subspaces undergo consistent orthogonal transformations during this process.

Combining Equation 3, the similarity matrices of $W_A^{(i)}$ and $W_B^{(i)}$ are defined as:

$$sim_U^{(i)}(\frac{\mathcal{M}_A}{\mathcal{M}_B}) = U_A^{(i)^T} \cdot U_B^{(i)}, \quad sim_V^{(i)}(\frac{\mathcal{M}_A}{\mathcal{M}_B}) = V_A^{(i)^T} \cdot V_B^{(i)} \tag{5}$$

The widely observed phenomenon can be expressed as $|sim_U^{(i)}(\frac{\mathcal{M}_{\text{base}}}{\mathcal{M}_{\text{post}}})| \approx |sim_V^{(i)}(\frac{\mathcal{M}_{\text{base}}}{\mathcal{M}_{\text{post}}})|$ (①-③ in Figure 3a), where $|\cdot|$ takes the absolute value of each matrix element to remove the possible sign ambiguity of singular vectors, which implies that the input and output subspaces of LLMs are undergoing highly symmetrical changes. Based on this inference, we can theoretically prove that the similarity matrices of the left and right singular vectors can be directly used to describe

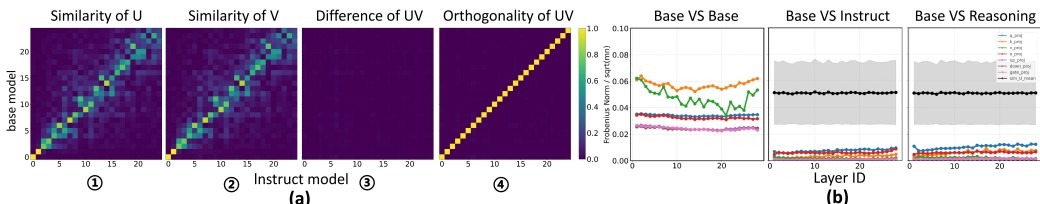

Figure 3: An example showing the orthogonality of singular vector similarity to the transformation performed. Only the first 25 dimensions are retained for clearer visualization. (a) shows the singular vector behavior of $W_O$ in the first Transformer block. Difference matrix (③) represents $\left|sim_U^{(0)} - sim_V^{(0)}\right|$, which is almost a zero matrix. ④ is $I_{orth}^{(0)}$ of $W_O^{(0)}$. Most of its diagonal elements are close to 1, and the rest are basically 0. (b) extensively verifies the approximate equality of $Q_1^{(i)}$ and $Q_2^{(i)}$ comparing $\mathcal{M}_{\text{base}}$ to $\mathcal{M}'_{\text{base}}$ and $\mathcal{M}_{\text{post}}$.

the transformation dynamics within the parameter space of LLMs, and only rotate the orthogonal bases already formed during the pre-training of LLMs. For $\mathcal{M}_{\text{base}} \rightarrow \mathcal{M}_{\text{post}}$, the change in left and right singular vectors from $W_{\text{base}}^{(i)}$ to $W_{\text{post}}^{(i)}$ can be framed as applying **coordinated orthogonal transformations** to them:

$$U_{\text{post}}^{(i)} = U_{\text{base}}^{(i)} Q_1^{(i)}, \quad V_{\text{post}}^{(i)} = V_{\text{base}}^{(i)} Q_2^{(i)}, \quad Q_1^{(i)} \approx Q_2^{(i)} = sim_{U/V}^{(i)} \tag{6}$$

where $Q_1^{(i)}$ and $Q_2^{(i)}$ are transformation matrices. The derivation of Equation 6 is given in Appendix G.2, which strongly reflects the collaborative and consistent variation of the input and output subspaces. We validate this claim by leveraging the properties of orthogonal matrices:

$$if \ \ Q_1^{(i)} = Q_2^{(i)}, \ \ then \ \ Q_1^{(i)^T} Q_2^{(i)} = (U_{\text{base}}^{(i)^T} U_{\text{post}}^{(i)})^T \cdot (V_{\text{base}}^{(i)^T} V_{\text{post}}^{(i)}) = I_{orth}^{(i)} = I \tag{7}$$

where $I \in \mathbb{R}^{n \times n}$ is the identity matrix. We quantify the orthogonality and the equality between $Q_1^{(i)}$ and $Q_2^{(i)}$ by measuring the proximity of $I_{orth}^{(i)}$ to $I$, employing the normalized Frobenius norm $\mathcal{NF}^{(i)} = \mathcal{F}^{(i)}(I_{orth}^{(i)} - I)/\sqrt{n^2} = \mathcal{F}^{(i)}(I_{orth}^{(i)} - I)/n$ as our metric. To eliminate the possibility of low $\mathcal{NF}^{(i)}$ due to insufficient training, we also plot the mean and standard deviation of $\mathcal{NF}_{\text{sim}}^{(i)} = \mathcal{F}^{(i)}((sim_U^{(i)} - I)/n)$ as line plots (shaded regions denote standard deviation) for all matrix types in each Transformer block.

④ in Figure 3a presents our visualization of $I_{orth}^{(0)}$ for $W_O^{(0)}$, and Figure 3b illustrates $\mathcal{NF}^{(i)}$ and $\mathcal{NF}_{\text{sim}}^{(i)}$ in all the weight matrices of the layers. It can be observed that for $\mathcal{M}_{\text{post}}$, the values of $\mathcal{NF}^{(i)}$ are consistently and significantly lower than those of $\mathcal{M}'_{\text{base}}$ across all layers while $\mathcal{NF}_{\text{sim}}^{(i)}$ sustains a persistently high magnitude, directly demonstrating that $Q_1^{(i)}$ and $Q_2^{(i)}$ are approximately equal orthogonal matrices throughout post-training. We can further conclude that the variation in singular vectors on the left and right can be approximately characterized by consistent orthogonal transformations with negligible deviation, a property absent in different pretrained models (see Appendix B.2). More detailed test results are in Appendix B.

## 5 ANALYSIS OF POST-TRAINING

Based on the observation of the aforementioned phenomena, we propose a simplified mathematical model of the weight changes from $\mathcal{M}_{\text{base}} \rightarrow \mathcal{M}_{\text{post}}$, which prior work has struggled to describe formally (Du et al., 2025; Marks & Tegmark, 2024; Jain et al., 2024; Lee et al., 2024). For $W_{\text{base}} \in \mathcal{M}_{\text{base}}$ and $W_{\text{post}} \in \mathcal{M}_{\text{post}}$, the changes imposed by post-training on the parameters can be approximated by **a linear factor** $\alpha$ and **an orthogonal matrix Q**:

$$W_{\text{post}} = U_{\text{post}} \Sigma_{\text{post}} V_{\text{post}}^T \approx (U_{\text{base}} Q) \cdot (\alpha \Sigma_{\text{base}}) \cdot (V_{\text{base}} Q)^T \tag{8}$$

The relation $\Sigma_{\text{post}} = \alpha \Sigma_{\text{base}}$ captures how post-training globally scales the singular values, whereas $U_{\text{post}} = U_{\text{base}} Q$ and $V_{\text{post}} = V_{\text{base}} Q$ indicate a consistent orthogonal transformation of the input and output subspaces. From this perspective, post-training can be viewed as a reparameterization of the pretrained subspaces. This section provides empirical validation that post-training a BASE model fundamentally corresponds to learning structured orthogonal rotations, where singular value scaling constitutes a secondary effect.

## 5.1 SINGULAR VALUES SCALING IS JUST A TEMPERATURE-CONTROLLED MECHANISM

Equation 8 demonstrates that post-training does not alter the singular value distribution formed during pre-training in BASE models, but merely scales it proportionally. We designed a controlled experiment to verify the impact of post-training on the singular values of POST models.

**Experiments.** A direct corollary of Equation 8 is that the singular value distribution of POST models can be approximated by combining the singular value distribution of BASE models with an appropriate linear factor. Consequently, the models before and after singular value replacement should exhibit nearly identical performance. For $\mathcal{M}_{\text{post}}$, we perform Construction 9 on each of their weight matrices across all transformer blocks, which involves replacing the singular values of $\mathcal{M}_{\text{post}}$ with those from $\mathcal{M}_{\text{base}}$ and a given linear factor $\alpha'$:

$$W_{\text{post}}^{(i)} \leftarrow U_{\text{post}}^{(i)} \cdot (\boldsymbol{\alpha'} \boldsymbol{\Sigma}_{\text{base}}^{(i)}) \cdot V_{\text{post}}^{(i)^T} \tag{9}$$

We denote the resulting model after substitution of singular values as $\mathcal{M}_{\text{post}}^{\text{replaced}}$. The choice of $\alpha'$ is shown in Table 4. We then evaluate both $\mathcal{M}_{\text{post}}$ and $\mathcal{M}_{\text{post}}^{\text{replaced}}$ on four standard benchmarks: GSM8K (Cobbe et al., 2021), MATH-500 (Hendrycks et al., 2021b), MMLU (dev split) (Hendrycks et al., 2021a), and GPQA (Rein et al., 2023). Performance is measured by pass@1 accuracy(%) with a token limit of 1024. To ensure reliability, all evaluations are conducted with three independent repetitions, and the average values are reported. The results are shown in Table 1.

Table 1: Performance comparison between original and replaced models across GSM8K, MATH-500, MMLU, and GPQA with pass@1 accuracy (%).

| BASE Models | REPLACED Types | GSM8K | MATH-500 | MMLU (dev) | GPQA |
|---|---|---|---|---|---|
| *Qwen2.5-Math-1.5B* | $\mathcal{M}_{\text{Instruct}}$ | 85.14±0.14 | 65.47±0.90 | 48.04±0.60 | 30.44±0.36 |
| | $\mathcal{M}_{\text{Instruct}}^{\text{replaced}}$ | **85.59±0.09** | **61.67±0.57** | **49.47±0.29** | **25.99±0.70** |
| | $\mathcal{M}_{\text{reasoning}}$ | 62.88±0.59 | 32.73±1.64 | 25.02±0.59 | 7.02±0.44 |
| | $\mathcal{M}_{\text{reasoning}}^{\text{replaced}}$ | **69.45±0.43** | **41.46±0.53** | **35.52±0.81** | **9.45±1.59** |

It can be observed that $\mathcal{M}_{\text{post}}^{\text{replaced}}$ maintains the performance of the $\mathcal{M}_{\text{post}}$, which once again illustrates the importance of Equation 8 and verifies that post-training does not alter the singular value distribution of the original model. Notably, we observe a significant performance gain in $\mathcal{M}_{\text{reasoning}}^{\text{replaced}}$. The underlying cause of this enhancement may lie in the reduction of the number of tokens output by the models (as shown in Table 6), which ensures that the model-generated responses are not truncated by the pre-specified token limit. The reduction in token count stems from the proposed approximate replacement operation, which enforces uniform scaling across all singular values, thereby mitigating potential noise during the training process. This in turn enables $\mathcal{M}_{\text{reasoning}}^{\text{replaced}}$ to generate more concise token sequences when addressing simple queries. Detailed experimental setups, the selection method of $\alpha'$, and results across different model scales and families are provided in Appendix C.1.

**Scaling of singular values is just a temperature-controlled mechanism.** To better visualize the change mechanism of singular values, we directly employ Construction 14 (the equivalent expression of Construction 9 when all $\alpha' = 1$) to construct $\mathcal{M}_{\text{replaced}}$ and analyze the attention score distributions of the modified model (Figure 4a). The results show that the attention score distributions remain largely consistent, exhibiting no significant shifts. Instead, the replacement appears to induce a smoothing effect that resembles a temperature-controlled process (see Appendix G.1 for proof). The measure of *attention entropy* $\mathcal{H}$ (Kumar & Sarawagi, 2019) in Figure 4b supports this potential mechanism. The attention entropy $\mathcal{H}$ of $\mathcal{M}_{\text{replaced}}$ closely matches that of the original $\mathcal{M}_{\text{Instruct}}$, suggesting that the singular value replacement does not disrupt the structural integrity of LLMs or its capacity to capture contextual dependencies. More detailed results are given in Appendix C.2.

Notably, the attention entropy before and after the replacement remains closely aligned, suggesting that the entropy transformation induced by post-training primarily serves as a secondary temperature control mechanism rather than substantially altering the model's behavior. This further implies that singular value scaling is a secondary effect accompanying the post-training process, not its primary mechanism.

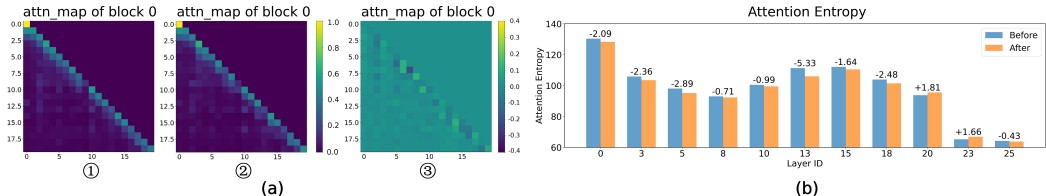

Figure 4: Visualization of the average attention patterns before and after replacing the singular values. ① in (a) shows the original attention heads, while ② presents the averaged attention heads from the modified model. ③ illustrates the differences between the original and modified attention patterns. Panel (b) suggests that this behavior corresponds to a modulation of attention entropy.

## 5.2 CONSISTENT ORTHOGONAL TRANSFORMATIONS ARE THE CORE OF POST-TRAINING

While replacing the singular values only mildly alters the model's behavior, disrupting the approximate orthogonal consistency between the input and output subspaces leads to a clear mode collapse in $\mathcal{M}_{\text{post}}$. To validate the functional importance of this coherence, we design a controlled experiment with two comparative settings.

**Experiments.** In the first setting (ABLATION), we remove the orthogonal transformation applied to the output subspaces of $W_{\text{post}}$ (Construction 10), while preserving the transformation on the input subspaces. In the second setting (RESTORATION), we restore coherence by applying to the output subspaces the same orthogonal transformation derived from the input subspaces (Construction 11).

$$W_{\text{post}}^{(i)} \leftarrow U_{\text{post}}^{(i)} \Sigma_{\text{post}} \cdot V_{\text{base}}^{(i)T} \tag{10}$$

$$W_{\text{post}}^{(i)} \leftarrow U_{\text{post}}^{(i)} \Sigma_{\text{post}} \cdot (V_{\text{base}}^{(i)} Q)^T = U_{\text{post}}^{(i)} \Sigma_{\text{post}} \cdot (V_{\text{base}}^{(i)} \cdot U_{\text{base}}^{(i)T} U_{\text{post}}^{(i)})^T \tag{11}$$

To assess the functional role of consistent orthogonal transformations, we feed the same input into $\mathcal{M}_{\text{post}}$ under three settings: the original model, the ABLATION model ($\mathcal{M}_{\text{post}}^{\text{ablation}}$), and the RESTORATION model ($\mathcal{M}_{\text{post}}^{\text{restoration}}$). All weight matrices in SAs are modified according to Constructions 10 and 11. We employ the same experimental setup as in Table 1 to evaluate the performance of restoration models across four datasets, with the results presented in Table 2:

Table 2: Performance comparison between original and RESTORATION models across GSM8K, MATH-500, MMLU, and GPQA with pass@1 accuracy (%).

| BASE Models | RESTORATION Types | GSM8K | MATH-500 | MMLU (dev) | GPQA |
|---|---|---|---|---|---|
| | $\mathcal{M}_{\text{Instruct}}$ | 85.14±0.14 | 65.47±0.90 | 48.04±0.60 | 30.44±0.36 |
| | $\mathcal{M}_{\text{Instruct}}^{\text{ablation}}$ | 0.00±0.00 | 0.00±0.00 | 0.00±0.00 | 0.00±0.00 |
| *Qwen2.5-* | $\mathcal{M}_{\text{Instruct}}^{\text{restoration}}$ | **84.53±0.25** | **66.20±0.16** | **41.28±0.44** | **27.69±0.29** |
| *Math-1.5B* | $\mathcal{M}_{\text{reasoning}}$ | 62.88±0.59 | 32.73±1.64 | 25.02±0.59 | 7.02±0.44 |
| | $\mathcal{M}_{\text{reasoning}}^{\text{ablation}}$ | 0.00±0.00 | 0.00±0.00 | 0.00±0.00 | 0.00±0.00 |
| | $\mathcal{M}_{\text{reasoning}}^{\text{restoration}}$ | **61.54±1.19** | **30.93±0.57** | **29.00±0.44** | **6.75±0.27** |

The performance of ABLATION models produce **nonsensical outputs** across different tasks, as shown in the case examples in Figure 5, leading to 0% accuracy across all evaluation metrics. In contrast, RESTORATION models recover meaningful outputs, further supporting the hypothesis of consistent orthogonal transformations in LLMs. The results across different model scales and families are provided in the Appendix D.1.

**Orthogonal Consistency and Model Integrity.** To further investigate the role of consistent orthogonal transformations in shaping the latent space across Transformer blocks, we evaluate the hidden representations of the ABLATION and RESTORATION models using *Centered Kernel Alignment* (CKA) (Kornblith et al., 2019), a standard metric for quantifying representational similarity across neural network layers. We use 100 questions from the GSM8K dataset and compute the average hidden representation at each layer across these inputs. CKA scores are then calculated between the original model (①) and the ABLATION (②) and RESTORATION (③) models, as shown in Figure 6.

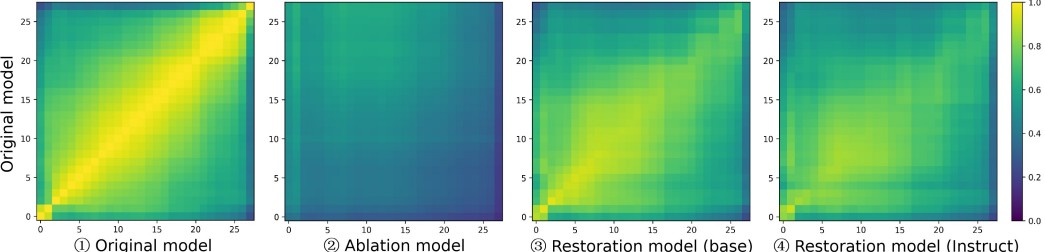

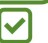

QUESTION:

Janet's ducks lay 16 eggs per day. She eats three for breakfast every morning and bakes muffins for her friends every day with four. She sells the remainder at the farmers' market daily for $2 per fresh duck egg. How much in dollars does she make every day at the farmers' market?

Okay, so Janet has ducks that lay eggs, and she uses some for her own stuff. I need to ...

Thus, Janet makes \boxed{18} dollars every day at the farmers' market.

Original response ☑

putsHost former . organizers, td);, ][ t ; modeled,, . . Story .详解 units PD Nap  t� through ther -f d fact through ~ ' . .Thing says� put 着  . specific. ., . already  ., lh ,, [, through  . /* . .,   explicit noth...

Ablation response ☒

Okay, let me try to figure out how much Janet makes every day at the farmers' market.  So, first ...

So, Janet makes **$\boxed{18}$** every day at the farmers' market.

Restoration response ☑

Figure 5: An example of model responses under three different settings. The ABLATION model outputs all garbled characters, while the RESTORATION model reconstructs the features of the original model through the orthogonal matrix of the input subspaces.

① Original model    ② Ablation model    ③ Restoration model (base)    ④ Restoration model (Instruct)

Figure 6: Heatmaps of CKA under different settings. ② corresponds to the ablation in Construction 10, which substantially disrupts the original model's representational structure. ③ and ④, corresponding to restorations via Constructions 11 and 12, effectively recover the original hidden representations.

The results reveal that the ablation (②) leads to an immediate and significant disruption of the model's representational structure starting from the very first layer. This indicates that the effect is not merely a result of cumulative downstream errors, but rather a fundamental alteration of the model's initial architecture. The restoration process (③) effectively reinstates the original representational geometry, underscoring the structural importance of the orthogonal transformations.

Additional experimental settings and results are provided in Appendix D.2. These findings suggest that the consistent orthogonal transformations between the input and output subspaces represent a central mechanism driving parameter reorganization during post-training adaptation, and offers a novel perspective that prompts us to narrow down the research scope of the impact exerted by post-training on the parameter space to the consistent rotation matrix Q.

**The equivalence of different post-training methods.** We theoretically prove that POST models initialized from the same pretrained parameters but trained on different data distributions are mutually transformable via a shared set of orthogonal transformations (see Appendix G.3 for proof). To test this hypothesis, we construct a new RESTORATION model from $\mathcal{M}_{\text{Instruct}}$ following Construction 12, and evaluate its similarity to the original model using a CKA heatmap (marked as ④ in Figure 6).

$$W_{\text{post}}^{(i)} \leftarrow U_{\text{post}}^{(i)} \Sigma_{\text{post}} \cdot (V_{\text{Instruct}}^{(i)} Q')^T = U_{\text{post}}^{(i)} \Sigma_{\text{post}} \cdot (V_{\text{Instruct}}^{(i)} \cdot U_{\text{Instruct}}^{(i)}{}^T U_{\text{post}}^{(i)})^T \quad (12)$$

This effective restoration of the latent space confirms the correctness of the hypothesis. We believe that this equivalence actually provides a parametric basis for certain universal phenomena. For example, it allows us to expose a potential mechanism behind *catastrophic forgetting*: when shared orthogonal transformations are disrupted and overwritten by new task-specific ones, the original transformations are lost, leading to performance degradation on prior tasks. We believe this inference can provide parameter-based support for understanding the forgetting mechanism of LLMs.

## 6 CONCLUSION

The paper establishes a unified and interpretable framework for understanding how post-training reshapes the internal structure of large language models. Through a comprehensive SVD analysis of linear layers, we identify two consistent transformations: a near-uniform geometric scaling of singular

values and highly consistent orthogonal transformations of singular vectors, both pervasive across model families and parameter scales. Our theoretical and empirical analyses indicate that while singular value scaling can be interpreted as a temperature-like adjustment, the essential functional change lies in the structured rotations of singular vectors, whose disruption markedly degrades performance. These findings not only provide a theoretical foundation for potential applications (see Appendix F for a related discussion), but also offer the first systematic account of the reparameterization dynamics governing large language models.

## 7 LIMITATION

While this paper identifies two structural changes in the parameter space of SAs and FFNs, our analysis primarily focuses on weight matrices in models that undergo supervised post-training. This restriction naturally raises several open questions: **Do reinforcement learning–based post-training methods exhibit the same structural phenomena? If the architecture or training paradigm of large models changes substantially, will the observed regularities persist? Do other components in LLMs with specific functions (such as normalization layers and output projection heads) follow similar patterns?** A detailed discussion in Appendix E further demonstrates the generality of these two structural changes.

Moreover, our findings also point to a deeper theoretical challenge: **what underlying mechanism gives rise to such striking regularities in LLMs?** We conjecture that a unified theoretical framework must exist—one capable of explaining the emergence and stability of these structural properties across different training paradigms. We view the pursuit of such a framework as a promising and impactful direction for future research.

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

# A SINGULAR VALUE SCALING ACROSS MODELS OF DIFFERENT FAMILIES AND SIZES

In the main paper, we introduce the SVSMs of *Qwen2.5-Math-1.5B* as the BASE model. This section continues to present comparisons of models with different post-training methods based on BASE models *Qwen2.5-Math-7B*, *Llama-3.1-8B*, and *Qwen2.5-14B* in DeepSeek-AI et al. (2025). The different POST versions of these models are described in the Appendix H.2. We will also provide a detailed analysis of the cross-layer stability of the near-uniform geometric scaling.

## A.1 SVSMs

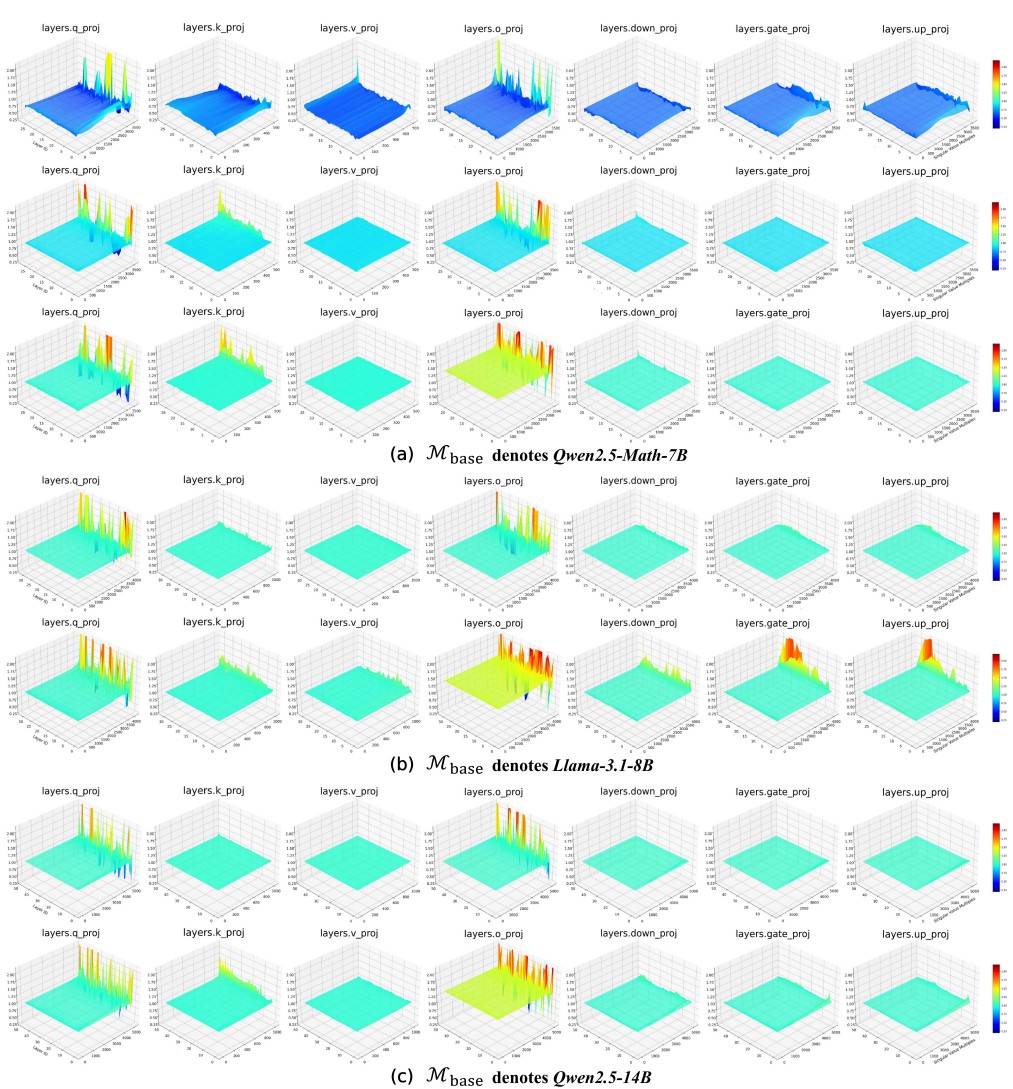

Figure 7: The heatmaps of SVSMs. The BASE models of (a), (b) and (c) are *Qwen2.5-Math-7B*, *Llama-3.1-8B* and *Qwen2.5-14B* respectively. Unlike *Qwen2.5-Math-7B* which has different pretrained versions like *Qwen2.5-7B*, only INSTRUCT version and REASONING version of the latter two models are compared.

Figure 7 shows SVSMs of different BASE models. We empirically observe a consistent pattern of singular value scaling across different post-training methods, where the principal singular values exhibit identical scaling ratios across different layers. This phenomenon universally manifests in all weight matrices. Notably, the $W_O$ matrices in all REASONING models demonstrate significantly higher overall scaling ratios compared to other weight matrices.

## A.2 Cross-layer Stability of Singular Value Scaling

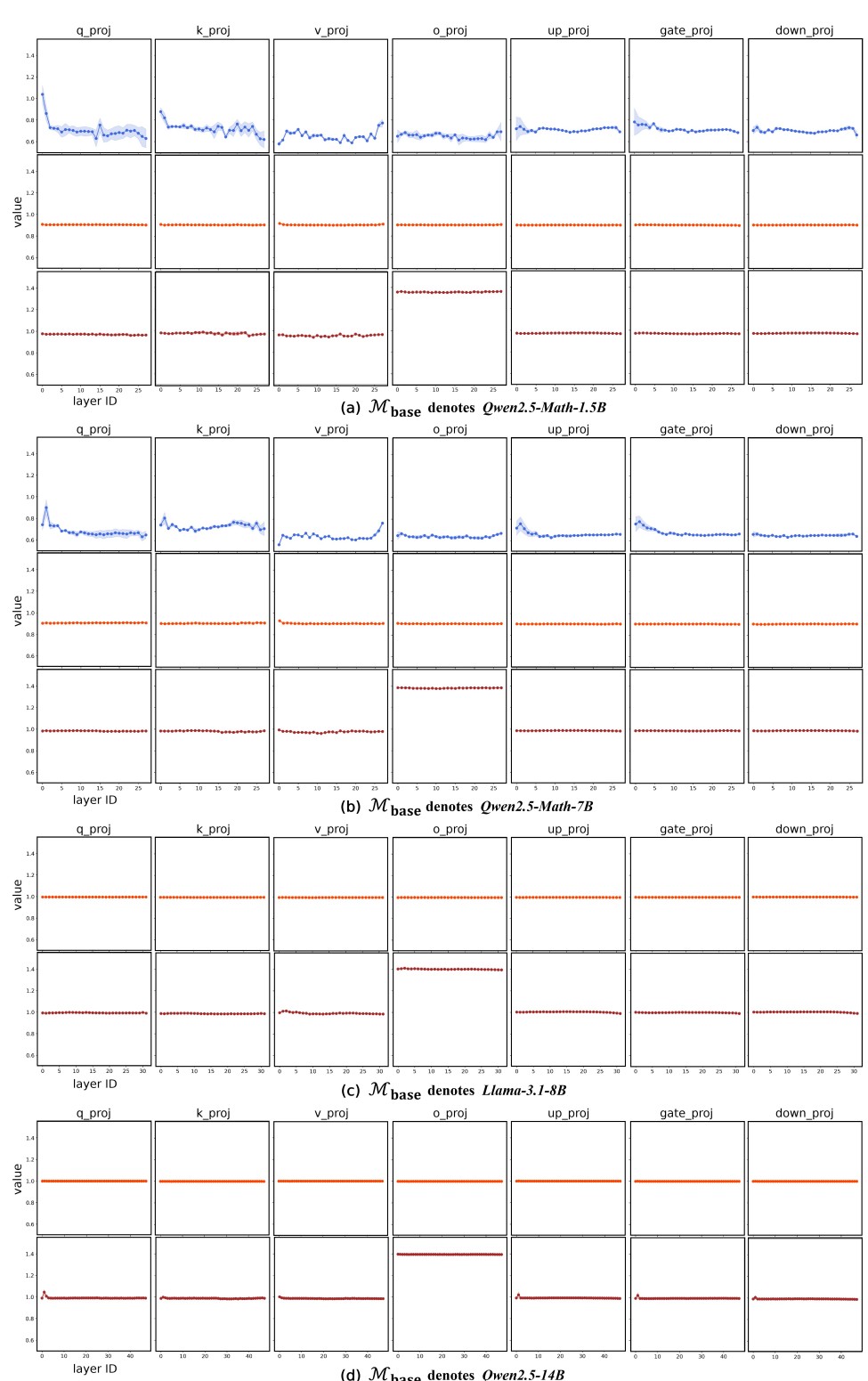

Figure 8: The bandwidth plot shows the distribution ( $mean \pm std$ ) of the scaling factors for the top 90% singular values in each layer. The blue line indicates comparison with $\mathcal{M}'_{\text{base}}$, while the light orange and brown curves correspond to comparisons with $\mathcal{M}_{\text{instruct}}$ and $\mathcal{M}_{\text{reasoning}}$ respectively.

Figure 8 shows the mean (dark line) and standard deviation (light band) of the scaling factors for the top 90% principal singular values across all Transformer blocks. As can be seen from the figure, both the INSTRUCT and REASONING models show stability in singular value scaling, which is both per-layer (almost no broadband is visible in the INSTRUCT and REASONING models) and cross-layer (the values in each layer are almost the same). Table 3 further reports the overall mean and standard deviation of the scaling factors for the top 90% singular values across all layers. As shown, the standard deviation across different BASE models is substantially larger than that between each BASE model and its corresponding POST model (e.g., 37.39× std for in *Qwen2.5-Math-1.5B* between $\mathcal{M}'_{\text{base}}$ and $\mathcal{M}_{\text{Instruct}}$), and the maximum variation of $\mathcal{M}_{\text{post}}$ remains within 1%, demonstrating the stability of the singular value scaling phenomenon and further reinforcing our claim.

Table 3: Global layer statistics of the scaling of the top 90% singular values ( $mean \pm std$ ), measured for different model families and parameter scales.

| $SVSM(\frac{\cdot}{\mathcal{M}_{\text{base}}})$ | | $W_Q$ | $W_K$ | $W_V$ | $W_O$ |
|---|---|---|---|---|---|
| *Qwen2.5-Math-1.5B* | $\mathcal{M}'_{\text{base}}$ | $0.6709 \pm 0.1728$ | $0.7017 \pm 0.0903$ | $0.6465 \pm 0.0432$ | $0.6293 \pm 0.1272$ |
| | $\mathcal{M}_{\text{Intruct}}$ | $0.9071 \pm 0.0046$ | $0.9084 \pm 0.0053$ | $0.9026 \pm 0.0036$ | $0.9041 \pm 0.0036$ |
| | $\mathcal{M}_{\text{reasoning}}$ | $0.9710 \pm 0.0131$ | $0.9723 \pm 0.0109$ | $0.9513 \pm 0.0103$ | $1.3551 \pm 0.0058$ |
| *Qwen2.5-Math-7B* | $\mathcal{M}'_{\text{base}}$ | $0.6621 \pm 0.0827$ | $0.7033 \pm 0.0688$ | $0.6388 \pm 0.0368$ | $0.6257 \pm 0.0317$ |
| | $\mathcal{M}_{\text{Intruct}}$ | $0.9074 \pm 0.0043$ | $0.9103 \pm 0.0111$ | $0.9040 \pm 0.0047$ | $0.9056 \pm 0.0027$ |
| | $\mathcal{M}_{\text{reasoning}}$ | $0.9837 \pm 0.0036$ | $0.9823 \pm 0.0072$ | $0.9737 \pm 0.0072$ | $1.3800 \pm 0.0031$ |
| *Llama-3.1-8B* | $\mathcal{M}_{\text{Intruct}}$ | $0.9960 \pm 0.0017$ | $0.9951 \pm 0.0008$ | $0.9957 \pm 0.0009$ | $0.9975 \pm 0.0027$ |
| | $\mathcal{M}_{\text{reasoning}}$ | $1.0041 \pm 0.0181$ | $0.9898 \pm 0.0058$ | $0.9930 \pm 0.0093$ | $1.4112 \pm 0.0187$ |
| *Qwen2.5-14B* | $\mathcal{M}_{\text{Intruct}}$ | $0.9990 \pm 0.0006$ | $0.9989 \pm 0.0003$ | $0.9989 \pm 0.0002$ | $0.9989 \pm 0.0002$ |
| | $\mathcal{M}_{\text{reasoning}}$ | $0.9937 \pm 0.0142$ | $0.9901 \pm 0.0064$ | $0.9861 \pm 0.0031$ | $1.3952 \pm 0.0017$ |

| $SVSM(\frac{\cdot}{\mathcal{M}_{\text{base}}})$ | | $W_{up}$ | $W_{gate}$ | $W_{down}$ |
|---|---|---|---|---|
| *Qwen2.5-Math-1.5B* | $\mathcal{M}'_{\text{base}}$ | $0.7242 \pm 0.0882$ | $0.7282 \pm 0.1179$ | $0.6967 \pm 0.0274$ |
| | $\mathcal{M}_{\text{Intruct}}$ | $0.9016 \pm 0.0010$ | $0.9018 \pm 0.0017$ | $0.9019 \pm 0.0010$ |
| | $\mathcal{M}_{\text{reasoning}}$ | $0.9720 \pm 0.0023$ | $0.9687 \pm 0.0035$ | $0.9714 \pm 0.0026$ |
| *Qwen2.5-Math-7B* | $\mathcal{M}'_{\text{base}}$ | $0.6693 \pm 0.0454$ | $0.6791 \pm 0.0514$ | $0.6495 \pm 0.0140$ |
| | $\mathcal{M}_{\text{Intruct}}$ | $0.9021 \pm 0.0014$ | $0.9025 \pm 0.0013$ | $0.9024 \pm 0.0016$ |
| | $\mathcal{M}_{\text{reasoning}}$ | $0.9847 \pm 0.0020$ | $0.9839 \pm 0.0019$ | $0.9843 \pm 0.0021$ |
| *Llama-3.1-8B* | $\mathcal{M}_{\text{Intruct}}$ | $0.9961 \pm 0.0003$ | $0.9957 \pm 0.0003$ | $0.9961 \pm 0.0003$ |
| | $\mathcal{M}_{\text{reasoning}}$ | $1.0036 \pm 0.0041$ | $0.9988 \pm 0.0033$ | $1.0035 \pm 0.0044$ |
| *Qwen2.5-14B* | $\mathcal{M}_{\text{Intruct}}$ | $0.9991 \pm 0.0021$ | $0.9991 \pm 0.0015$ | $0.9990 \pm 0.0006$ |
| | $\mathcal{M}_{\text{reasoning}}$ | $0.9922 \pm 0.0132$ | $0.9924 \pm 0.0119$ | $0.9909 \pm 0.0062$ |

# B    CONSISTENT ORTHOGONAL TRANSFORMATIONS ACROSS MODELS OF DIFFERENT FAMILIES AND SIZES

In this section, we compare $\mathcal{NF}^{(i)}$ between the BASE and POST versions of *Qwen2.5-Math-7B*, *Llama-3.1-8B*, and *Qwen2.5-14B*. We also visualize the similarity, difference, and orthogonality matrices of the left and right singular vectors of $W_Q$, $W_K$, $W_V$, and $W_O$ (using the first and last Transformer blocks as examples), and discuss whether such orthogonal consistency is already present in the pre-training stage.

## B.1    VISUALIZING ORTHOGONAL CONSISTENCY ACROSS MODELS OF DIFFERENT FAMILIES AND SIZES

As shown in Figure 9, the $\mathcal{NF}^{(i)}$ values across different POST versions consistently remain low, in contrast to the higher values observed among the pre-training variants (Figure 9a, *Base vs Base*). This indicates that, despite variations in model scale and post-training methods, each matrix exhibits a high degree of consistency in the orthogonal transformations ($Q_1^{(i)}$ and $Q_2^{(i)}$) applied to its singular vectors. This phenomenon is illustrated more clearly in Figure 10-13, where most orthogonality matrices closely approximate the identity matrix.

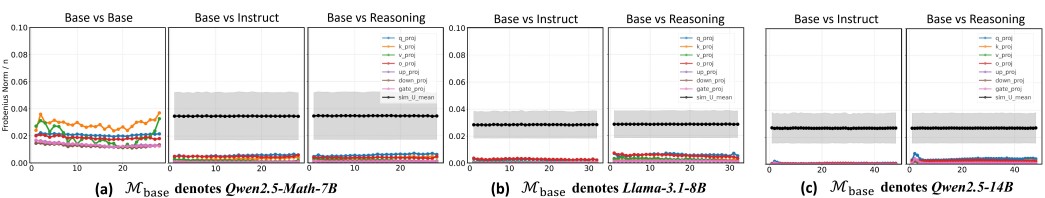

Figure 9: Extensively verifies the equality of $Q_1^{(i)}$ and $Q_2^{(i)}$ comparing $\mathcal{M}_{\text{base}}$ to $\mathcal{M}_{\text{post}}$ by $\mathcal{NF}^{(i)}$.

(a) $\mathcal{M}_{\text{base}}$ **denotes** *Qwen2.5-Math-1.5B*

(b) $\mathcal{M}_{\text{base}}$ **denotes** *Qwen2.5-Math-7B*

(c) $\mathcal{M}_{\text{base}}$ **denotes** *Llama-3.1-8B*

(d) $\mathcal{M}_{\text{base}}$ **denotes** *Qwen2.5-14B*

Figure 10: Visualizations of the similarity, difference and orthogonality matrices of the left and right singular vectors of the first and last Transformer block's $W_Q$ before and after post-training across models of different scales.

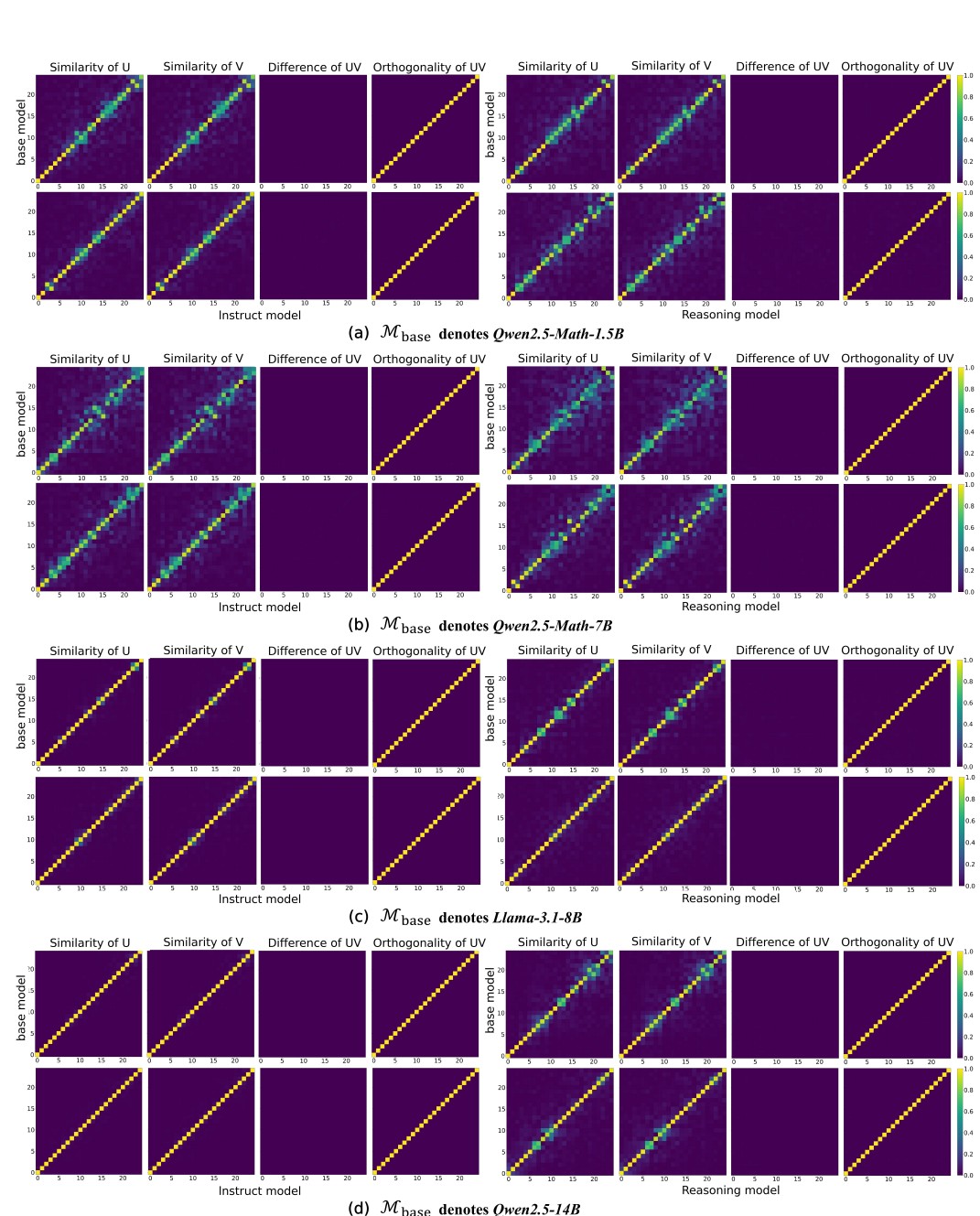

Figure 11: Visualizations of the similarity, difference and orthogonality matrices of the left and right singular vectors of the first and last Transformer block's $W_K$ before and after post-training across models of different scales.

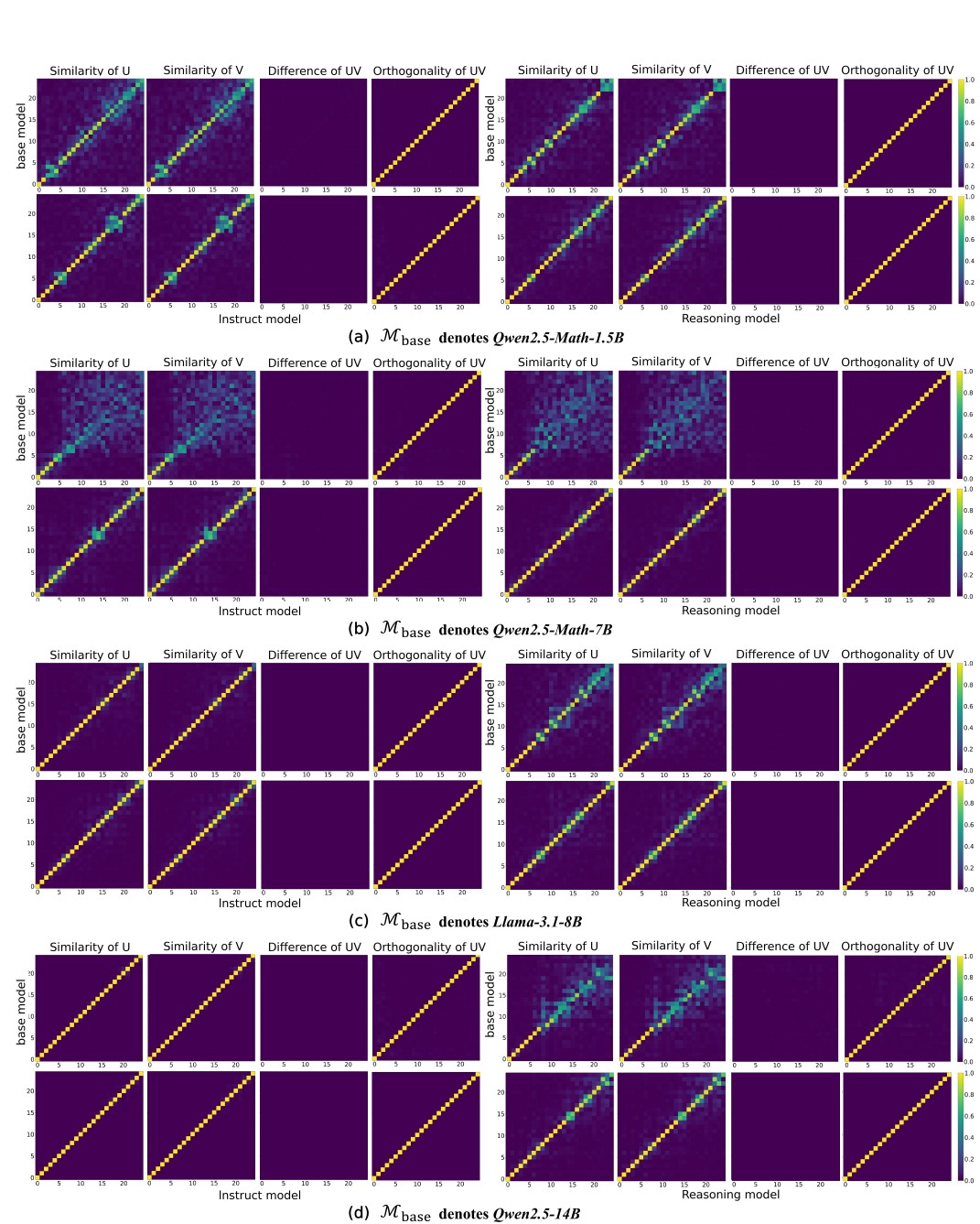

Figure 12: Visualizations of the similarity, difference and orthogonality matrices of the left and right singular vectors of the first and last Transformer block's $W_V$ before and after post-training across models of different scales.

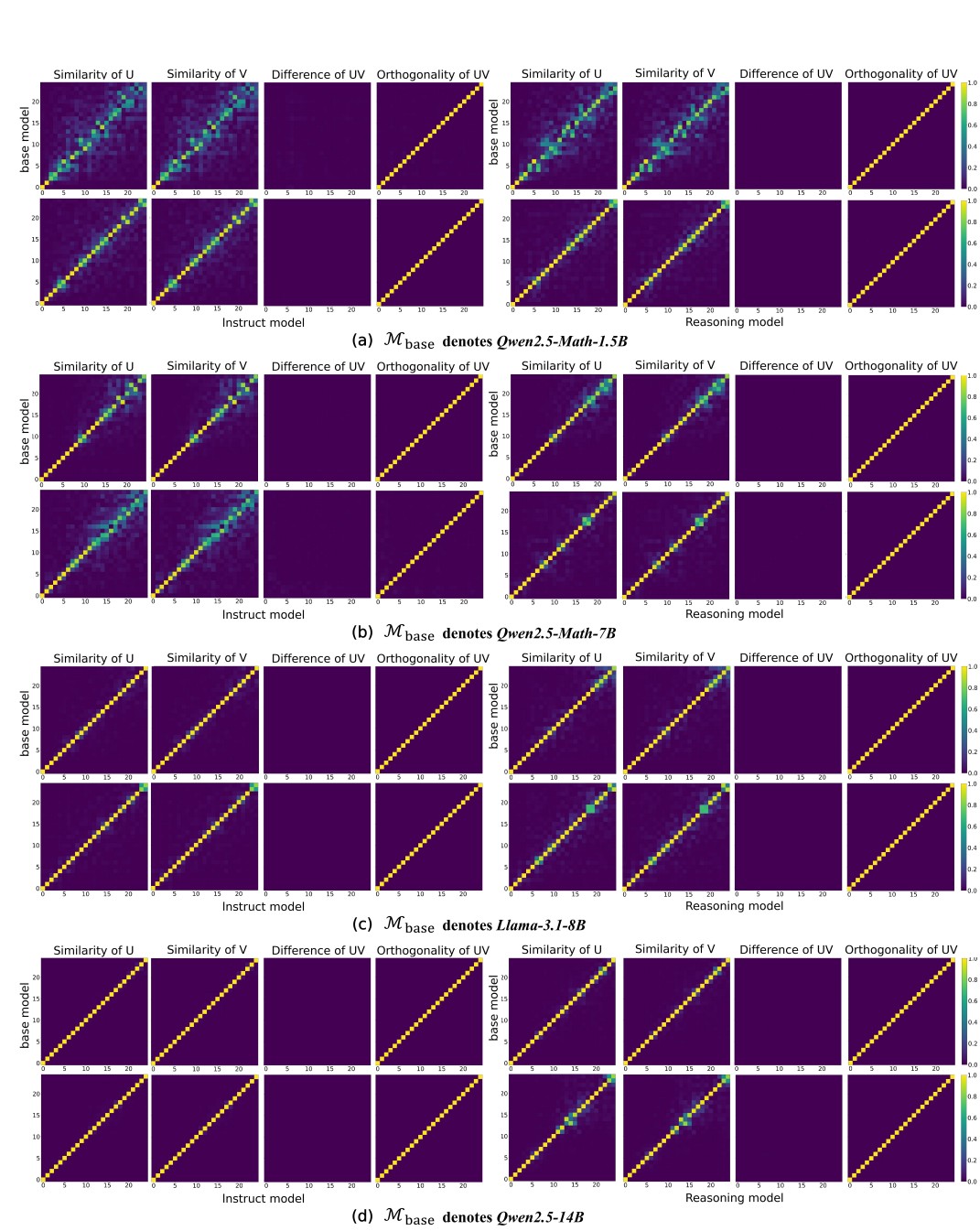

Figure 13: Visualizations of the similarity, difference and orthogonality matrices of the left and right singular vectors of the first and last Transformer block's $W_O$ before and after post-training across models of different scales.

We also observe that the similarity matrices of the left and right singular vectors are mostly concentrated along the diagonal. As shown in Appendix A, post-training does not alter the distribution of singular values of the weight matrices. When taken together with our current observation, this indirectly supports the view that post-training acts as a perturbation to the pretrained subspaces.

### B.2 TRANSFORMATIONS OF SINGULAR VECTORS DURING PRE-TRAINING

The similarity matrices of the left and right singular vectors across different BASE models do not exhibit strong diagonal dominance, suggesting substantial divergence in their pretrained subspaces (Figure 14). Despite this divergence, we observe a subtle and consistent pattern in the orthogonal transformations between the left and right singular vectors. This subtle consistency may stem from an accumulation of alignment errors, implying that the orthogonal transformations are systematically misaligned to some extent. We can calibrate $U_{\text{post}}, V_{\text{post}}$ in Equation 8:

$$
\begin{aligned}
U_{\text{post}} &= U_{\text{base}}(Q \cdot \Delta Q_1) \\
V_{\text{post}} &= V_{\text{base}}(Q \cdot \Delta Q_2)
\end{aligned}
\tag{13}
$$

The matrices $\Delta Q_1$ and $\Delta Q_2$ represent small-angle components that capture fine-grained deviations superimposed on the coordinated transformation of the left and right singular vectors during training. These residual transformation correspond to the perturbation term $I_{\text{orth}}$ in Equation 7. From this perspective, the amount of data used in post-training is substantially smaller than in pre-training. As a result, the accumulated perturbations introduced during post-training are also much smaller than the large-scale transformations of the left and right singular vectors induced during pre-training. It is reasonable to postulate that the accumulation of such errors precisely constitutes a significant factor in reshaping the subspaces of BASE models. Given that the cumulative deviations introduced by $\Delta Q_1$ and $\Delta Q_2$ remain sufficiently small, the overall transformations of the singular space can be well-approximated as coherent orthogonal rotations. This also supports the validity of the approximation made in Equation 8.

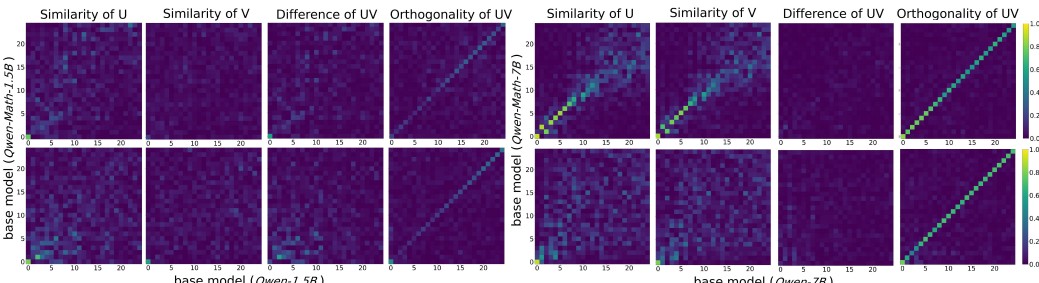

Figure 14: Visualizations of the similarity, difference and orthogonality matrices of the left and right singular vectors of the first and last Transformer block's $W_O$ between $\mathcal{M}_{\text{base}}$ and $\mathcal{M}'_{\text{base}}$.

## C EXPERIMENTS ON DIFFERENT REPLACED MODELS

This section will conduct the same experiments as presented in the main paper on models of varying scales and families, aiming to verify the universality and generalizability of the near-uniform geometric scaling phenomenon of singular values. The evaluation will include tests on four standard benchmark datasets, along with visualizations of attention entropy.

### C.1 PERFORMANCE OF DIFFERENT REPLACED MODELS

The purpose of performing Construction 9 on $\mathcal{M}_{\text{post}}$ is to verify that the singular value distribution of $\mathcal{M}_{\text{post}}$ can be reconstructed through the linear factor $\alpha'$ and the singular value distribution of $\mathcal{M}_{\text{base}}$, thereby validating the rationality of Equation 8. This verification critically depends on the selection of $\alpha'$. Our choice of $\alpha'$ is based on Table 3, as it reflects the overall distribution of singular value scaling factors. We obtain the final $\alpha'$ values for each type of weight matrix in the POST models by rounding the mean of these scaling factors, as presented in Table 4.

Table 4: $\alpha'$ values (right) assigned based on mean singular value scaling factors (left) of weight matrices per type (from Table 3).

| | POST Types | $W_Q$ | $W_K$ | $W_V$ | $W_O$ |
|---|---|---|---|---|---|
| *Qwen2.5-Math-1.5B* | $\mathcal{M}_{\text{Intruct}}$ | $0.9071 \to \mathbf{0.9}$ | $0.9084 \to \mathbf{0.9}$ | $0.9026 \to \mathbf{0.9}$ | $0.9041 \to \mathbf{0.9}$ |
| | $\mathcal{M}_{\text{reasoning}}$ | $0.9710 \to \mathbf{1.0}$ | $0.9723 \to \mathbf{1.0}$ | $0.9513 \to \mathbf{1.0}$ | $1.3551 \to \mathbf{1.4}$ |
| *Qwen2.5-Math-7B* | $\mathcal{M}_{\text{Intruct}}$ | $0.9074 \to \mathbf{0.9}$ | $0.9103 \to \mathbf{0.9}$ | $0.9040 \to \mathbf{0.9}$ | $0.9056 \to \mathbf{0.9}$ |
| | $\mathcal{M}_{\text{reasoning}}$ | $0.9837 \to \mathbf{1.0}$ | $0.9823 \to \mathbf{1.0}$ | $0.9737 \to \mathbf{1.0}$ | $1.3800 \to \mathbf{1.4}$ |
| *Llama-3.1-8B* | $\mathcal{M}_{\text{Intruct}}$ | $0.9960 \to \mathbf{1.0}$ | $0.9951 \to \mathbf{1.0}$ | $0.9957 \to \mathbf{1.0}$ | $0.9975 \to \mathbf{1.0}$ |
| | $\mathcal{M}_{\text{reasoning}}$ | $1.0041 \to \mathbf{1.0}$ | $0.9898 \to \mathbf{1.0}$ | $0.9930 \to \mathbf{1.0}$ | $1.4112 \to \mathbf{1.4}$ |
| *Qwen2.5-14B* | $\mathcal{M}_{\text{Intruct}}$ | $0.9990 \to \mathbf{1.0}$ | $0.9989 \to \mathbf{1.0}$ | $0.9989 \to \mathbf{1.0}$ | $0.9989 \to \mathbf{1.0}$ |
| | $\mathcal{M}_{\text{reasoning}}$ | $0.9937 \to \mathbf{1.0}$ | $0.9901 \to \mathbf{1.0}$ | $0.9861 \to \mathbf{1.0}$ | $1.3952 \to \mathbf{1.4}$ |

| | POST Types | $W_{up}$ | $W_{gate}$ | $W_{down}$ |
|---|---|---|---|---|
| *Qwen2.5-Math-1.5B* | $\mathcal{M}_{\text{Intruct}}$ | $0.9016 \to \mathbf{0.9}$ | $0.9018 \to \mathbf{0.9}$ | $0.9019 \to \mathbf{0.9}$ |
| | $\mathcal{M}_{\text{reasoning}}$ | $0.9720 \to \mathbf{1.0}$ | $0.9687 \to \mathbf{1.0}$ | $0.9714 \to \mathbf{1.0}$ |
| *Qwen2.5-Math-7B* | $\mathcal{M}_{\text{Intruct}}$ | $0.9021 \to \mathbf{0.9}$ | $0.9025 \to \mathbf{0.9}$ | $0.9024 \to \mathbf{0.9}$ |
| | $\mathcal{M}_{\text{reasoning}}$ | $0.9847 \to \mathbf{1.0}$ | $0.9839 \to \mathbf{1.0}$ | $0.9843 \to \mathbf{1.0}$ |
| *Llama-3.1-8B* | $\mathcal{M}_{\text{Intruct}}$ | $0.9961 \to \mathbf{1.0}$ | $0.9957 \to \mathbf{1.0}$ | $0.9961 \to \mathbf{1.0}$ |
| | $\mathcal{M}_{\text{reasoning}}$ | $1.0036 \to \mathbf{1.0}$ | $0.9988 \to \mathbf{1.0}$ | $1.0035 \to \mathbf{1.0}$ |
| *Qwen2.5-14B* | $\mathcal{M}_{\text{Intruct}}$ | $0.9991 \to \mathbf{1.0}$ | $0.9991 \to \mathbf{1.0}$ | $0.9990 \to \mathbf{1.0}$ |
| | $\mathcal{M}_{\text{reasoning}}$ | $0.9922 \to \mathbf{1.0}$ | $0.9924 \to \mathbf{1.0}$ | $0.9909 \to \mathbf{1.0}$ |

In our experiments, the output parameters of the LLMs are configured with a temperature of 0.2, a top_p of 0.95, and a maximum output token limit of 1024. This setting ensures stable generation while maintaining moderate diversity for subsequent statistical analysis. System prompts are provided in Appendix H.1. Each model is executed three times on the test set, with the final performance reported as the average score and variance. The results are presented in Table 5. The mean and variance of the average length of output tokens across three test runs are also reported in Table 6.

Table 5: Performance comparison between original and replaced models across GSM8K, MATH-500, MMLU, and GPQA with pass@1 accuracy (%).

| BASE Models | REPLACED Types | GSM8K | MATH-500 | MMLU (dev) | GPQA |
|---|---|---|---|---|---|
| *Qwen2.5-Math-7B* | $\mathcal{M}_{\text{Instruct}}$ | $95.75\pm0.12$ | $70.06\pm0.50$ | $55.90\pm0.16$ | $27.14\pm0.49$ |
| | $\mathcal{M}_{\text{Instruct}}^{\text{replaced}}$ | $\mathbf{95.25\pm0.06}$ | $\mathbf{73.00\pm0.43}$ | $\mathbf{55.20\pm0.16}$ | $\mathbf{27.22\pm0.41}$ |
| | $\mathcal{M}_{\text{reasoning}}$ | $62.70\pm1.05$ | $47.60\pm0.33$ | $58.71\pm0.91$ | $14.73\pm0.97$ |
| | $\mathcal{M}_{\text{reasoning}}^{\text{replaced}}$ | $\mathbf{72.28\pm0.42}$ | $\mathbf{53.66\pm0.81}$ | $\mathbf{60.69\pm1.03}$ | $\mathbf{18.01\pm0.87}$ |
| *Llama-3.1-8B* | $\mathcal{M}_{\text{Instruct}}$ | $34.70\pm1.24$ | $31.46\pm1.06$ | $67.48\pm0.44$ | $21.21\pm0.29$ |
| | $\mathcal{M}_{\text{Instruct}}^{\text{replaced}}$ | $\mathbf{34.92\pm0.37}$ | $\mathbf{32.60\pm1.14}$ | $\mathbf{65.26\pm0.57}$ | $\mathbf{20.11\pm0.76}$ |
| | $\mathcal{M}_{\text{reasoning}}$ | $60.17\pm0.07$ | $32.73\pm0.41$ | $52.51\pm1.47$ | $11.40\pm0.17$ |
| | $\mathcal{M}_{\text{reasoning}}^{\text{replaced}}$ | $\mathbf{68.72\pm0.43}$ | $\mathbf{29.73\pm0.90}$ | $\mathbf{52.16\pm1.29}$ | $\mathbf{9.17\pm0.51}$ |
| *Qwen2.5-14B* | $\mathcal{M}_{\text{Instruct}}$ | $94.24\pm0.29$ | $70.53\pm0.34$ | $90.63\pm0.16$ | $36.65\pm0.36$ |
| | $\mathcal{M}_{\text{Instruct}}^{\text{replaced}}$ | $\mathbf{94.11\pm0.25}$ | $\mathbf{69.13\pm0.09}$ | $\mathbf{89.93\pm1.01}$ | $\mathbf{35.60\pm1.48}$ |
| | $\mathcal{M}_{\text{reasoning}}$ | $70.61\pm0.46$ | $53.13\pm0.25$ | $77.89\pm0.76$ | $19.48\pm0.55$ |
| | $\mathcal{M}_{\text{reasoning}}^{\text{replaced}}$ | $\mathbf{79.49\pm0.42}$ | $\mathbf{52.33\pm0.25}$ | $\mathbf{75.79\pm1.03}$ | $\mathbf{19.02\pm0.32}$ |

Table 6: Comparison of average length of output tokens between Original and Replaced Models across GSM8K, MATH-500, MMLU, and GPQA.

| BASE Models | REPLACED Types | GSM8K | MATH-500 | MMLU (dev) | GPQA |
|---|---|---|---|---|---|
| *Qwen2.5-Math-1.5B* | $\mathcal{M}_{\text{Instruct}}$ | 305.01±1.54 | 542.32±1.21 | 402.60±3.13 | 633.82±5.09 |
| | $\mathcal{M}_{\text{Instruct}}^{\text{replaced}}$ | **302.92±2.54** | **527.03±4.11** | **408.09±4.31** | **610.73±8.94** |
| | $\mathcal{M}_{\text{reasoning}}$ | 539.82±6.86 | 911.55±5.55 | 619.34±13.82 | 952.00±18.83 |
| | $\mathcal{M}_{\text{reasoning}}^{\text{replaced}}$ | **427.41±5.33** | **864.71±8.03** | **590.98±15.42** | **939.18±9.91** |
| *Qwen2.5-Math-7B* | $\mathcal{M}_{\text{Instruct}}$ | 299.46±3.17 | 551.34±4.39 | 372.53±5.91 | 567.34±4.96 |
| | $\mathcal{M}_{\text{Instruct}}^{\text{replaced}}$ | **304.21±2.91** | **549.13±2.53** | **378.34±4.51** | **533.19±5.98** |
| | $\mathcal{M}_{\text{reasoning}}$ | 729.16±7.64 | 795.40±9.01 | 514.15±6.91 | 933.15±9.97 |
| | $\mathcal{M}_{\text{reasoning}}^{\text{replaced}}$ | **451.27±9.28** | **726.08±6.14** | **488.30±15.17** | **891.63±6.07** |
| *Llama-3.1-8B* | $\mathcal{M}_{\text{Instruct}}$ | 166.47±4.22 | 359.19±6.02 | 35.79±1.43 | 236.35±7.38 |
| | $\mathcal{M}_{\text{Instruct}}^{\text{replaced}}$ | **146.05±2.18** | **451.38±7.71** | **41.42±3.36** | **251.64±3.06** |
| | $\mathcal{M}_{\text{reasoning}}$ | 627.14±8.71 | 931.14±14.80 | 721.64±11.13 | 989.41±7.43 |
| | $\mathcal{M}_{\text{reasoning}}^{\text{replaced}}$ | **651.23±11.34** | **970.02±15.14** | **751.02±8.29** | **994.00±4.31** |
| *Qwen2.5-14B* | $\mathcal{M}_{\text{Instruct}}$ | 281.95±7.21 | 550.02±6.17 | 89.69±1.18 | 240.16±6.55 |
| | $\mathcal{M}_{\text{Instruct}}^{\text{replaced}}$ | **299.14±5.11** | **530.65±5.93** | **87.56±2.43** | **241.67±6.39** |
| | $\mathcal{M}_{\text{reasoning}}$ | 583.01±4.57 | 897.61±8.81 | 487.54±7.68 | 924.63±7.90 |
| | $\mathcal{M}_{\text{reasoning}}^{\text{replaced}}$ | **410.97±7.81** | **847.14±2.06** | **514.09±6.90** | **933.15±5.10** |

Experimental results demonstrate that models exhibit nearly identical performance before and after singular value replacement. This further validates that post-training does not alter the singular value distribution of pre-trained models, thereby supporting our conclusion.

We also observe that the performance of some REASONING models improves after singular value replacement. One possible explanation is that Construction 9 effectively eliminates noise arising from precision limitations or heterogeneous data during singular value adjustment of $\mathcal{M}_{\text{base}}$s' weight matrices in post-training phases. This reduction in noise consequently enables more efficient token consumption for simpler tasks (e.g., the notable decrease in output token count for $\mathcal{M}_{\text{reasoning}}^{\text{replaced}}$ of *Qwen2.5-Math-7B* on GSM8K). These observations suggest that post-training processes exert theoretically derivable influences on the singular values of weight matrices. We identify this phenomenon as a crucial direction for future theoretical investigation.

## C.2 ATTENTION ENTROPY OF DIFFERENT REPLACED MODELS

To demonstrate that singular value scaling is similar to a temperature-controlled mechanism, we perform the following operation on all weight matrices $W_{\text{post}}$ of the POST models:

$$W_{\text{post}} \leftarrow U_{\text{post}} \Sigma_{\text{base}} V_{\text{post}}^{T} \tag{14}$$

Construction 14 replaces the singular values of POST models' weight matrices with those from BASE models. To evaluate the impact of this substitution, we monitor the attention entropy $\mathcal{H}$. A substantial change in entropy suggests a shift in the distribution of attention scores, indicating a structural change. Otherwise, the effect may be interpreted as a soft temperature modulation.

We input example questions from different domains (Cobbe et al., 2021; Talmor et al., 2019; Hendrycks et al., 2021a; Rein et al., 2023) into replaced models $\mathcal{M}_{\text{replaced}}$ and observe their attention scores prior to generating the first token. Specifically, we track the average attention distribution from each attention head in Transformer blocks 0, 3, 5, 8, 10, 13, 15, 18, 20, 23, and 25, and compute the corresponding attention entropy.

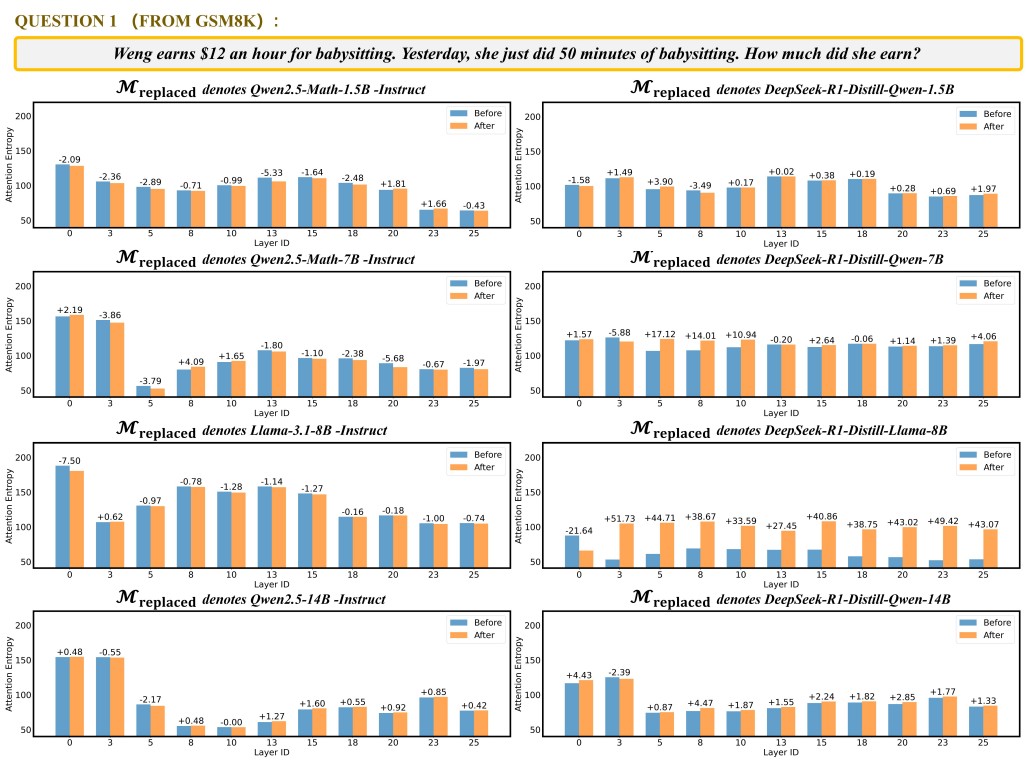

Figure 15: Attention entropy for different $\mathcal{M}_{\text{replaced}}$. The example input is from GSM8K.

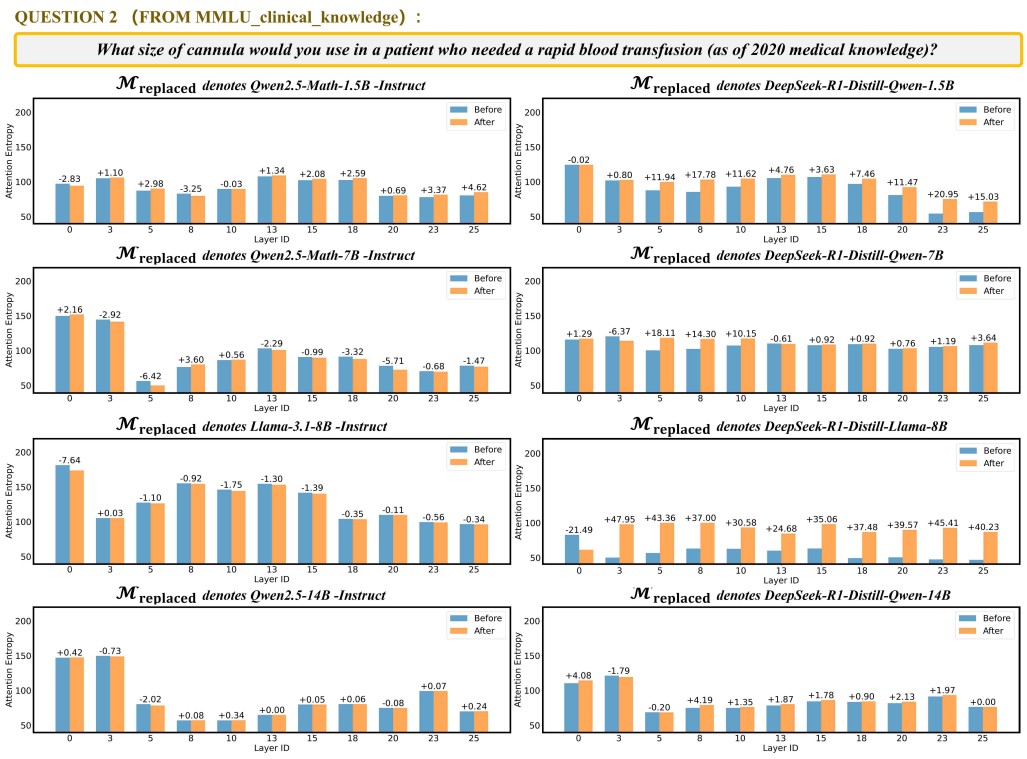

Figure 16: Attention entropy for different $\mathcal{M}_{\text{replaced}}$. The example input is from MMLU (clinical knowledge).

QUESTION 3 (FROM commonsenseQA)：

*The sanctions against the school were a punishing blow, and they seemed to what the efforts the school had made to change?*

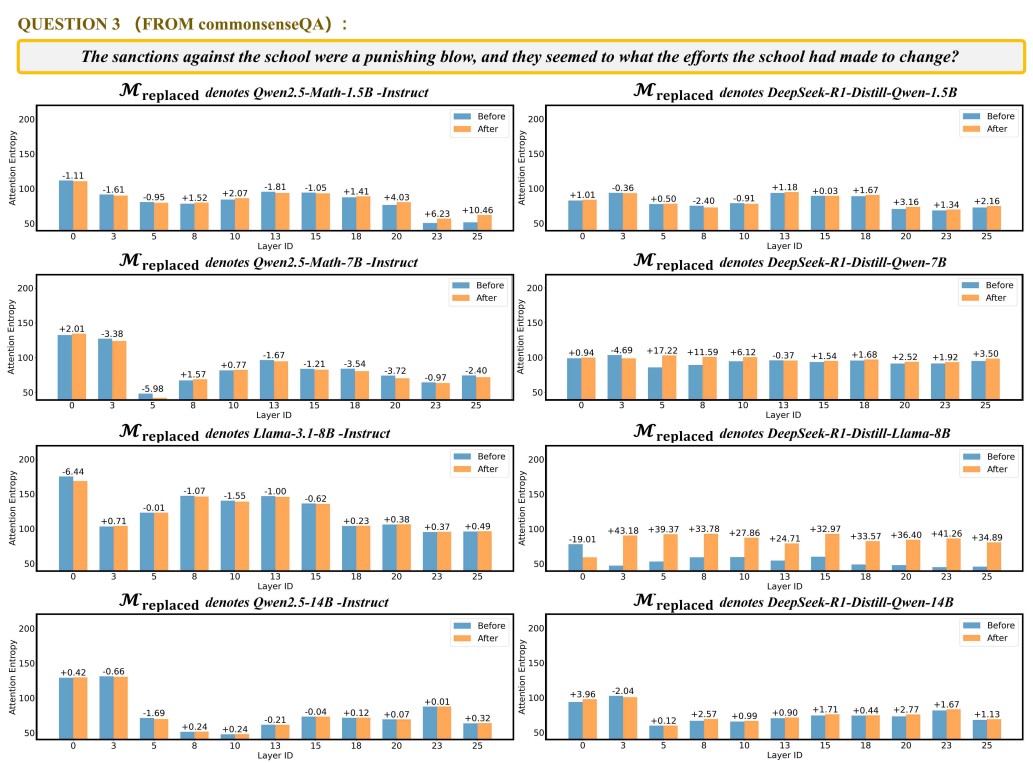

Figure 17: Attention entropy for different $\mathcal{M}_{\text{replaced}}$. The example input is from CommonsenseQA.

QUESTION 4 (FROM GPQA_diamond)：

*Two quantum states with energies E1 and E2 have a lifetime of 10^-9 sec and 10^-8 sec, respectively. We want to clearly distinguish these two energy levels. Which one of the following options could be their energy difference so that they be clearly resolved?*

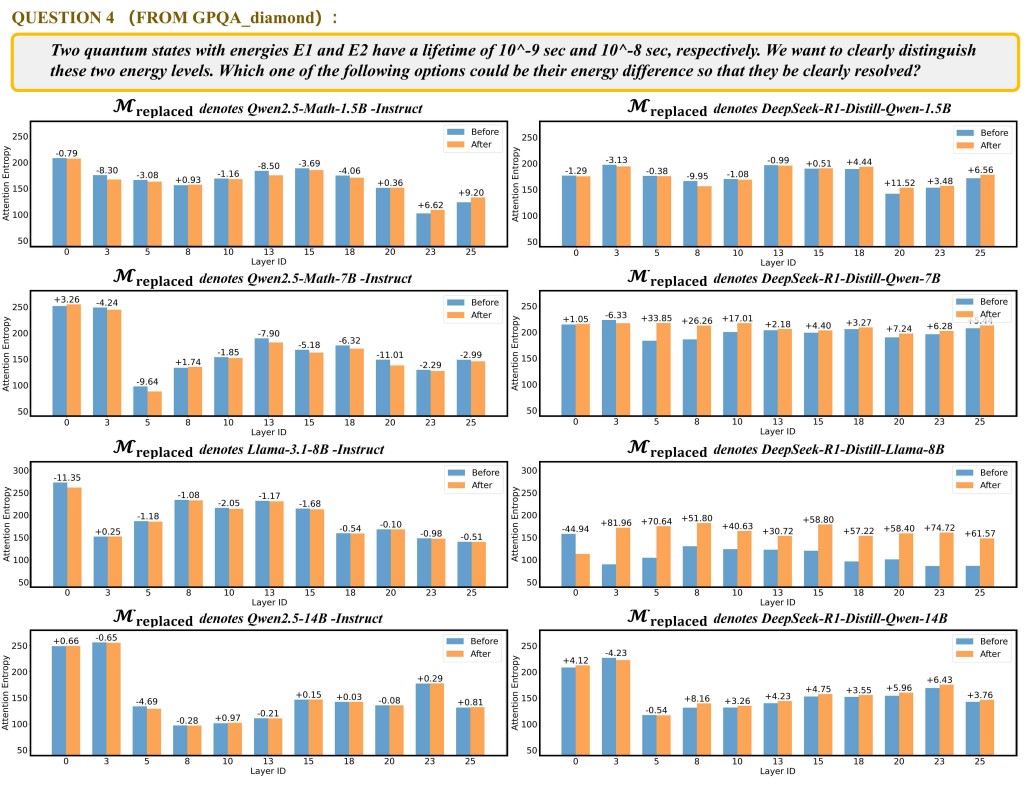

Figure 18: Attention entropy for different $\mathcal{M}_{\text{replaced}}$. The example input is from GPQA (diamond).

The replaced models $\mathcal{M}_{\text{replaced}}$, spanning diverse architectures and parameter scales, consistently preserve the attention entropy of their original counterparts across a range of examples . This robustness persists even under higher scaling of the singular values in the $W_O$ of REASONING models. In particular, *Qwen*-based models exhibit minimal sensitivity to such modifications, with attention entropy remaining largely unchanged (Figures 15, 16, 17, 18). In contrast, *LLaMA*-based REASONING models show an increase in attention entropy when the overall scale of $W_O$ singular values is reduced, consistent with a more uniform distribution of attention scores. Importantly, these effects are largely invariant to extreme amplification of singular values in the long tail of the spectrum, likely due to their negligible magnitude and limited contribution to the model's functional behavior. These findings support the interpretation of global singular value scaling as a temperature-like mechanism for modulating attention sharpness.

# D  EXPERIMENTS ON VERIFYING THE CONSISTENCY OF ORTHOGONAL TRANSFORMATIONS

This section highlights the critical importance of orthogonal consistency. While the main paper only demonstrates that disrupting orthogonal transformations in SA output subspaces can be compensated by preserving orthogonality in input subspaces, we present here a more extensive set of experimental results. We apply Construction 10 to matrices in $\mathcal{M}_{\text{post}}$ to obtain $\mathcal{M}_{\text{post}}^{\text{ablation}}$, and use Construction 11 to derive $\mathcal{M}_{\text{post}}^{\text{restoration}}$. These operations model the destruction and subsequent restoration of the output subspaces in the weight matrices. Similarly, we apply Constructions 15 and 16 to the input subspaces, as a symmetric counterpart to Constructions 10 and 11:

$$W_{\text{post}}^{(i)} \leftarrow \boldsymbol{U}_{\text{base}}^{(i)} \Sigma_{\text{post}} \cdot V_{\text{post}}^{(i)\,T} \tag{15}$$

$$W_{\text{post}}^{(i)} \leftarrow (\boldsymbol{U}_{\text{base}}^{(i)} \boldsymbol{Q}) \cdot \Sigma_{\text{post}} V_{\text{post}}^{(i)\,T} = (\boldsymbol{U}_{\text{base}}^{(i)} \cdot \boldsymbol{V}_{\text{base}}^{(i)\,T} \boldsymbol{V}_{\text{post}}^{(i)}) \cdot \Sigma_{\text{post}} V_{\text{post}}^{(i)\,T} \tag{16}$$

Constructions 10, 11, 15, and 16 provide an intuitive demonstration of the orthogonal consistency between the left and right singular vectors of each weight matrix in the model. For each $\mathcal{M}_{\text{post}}$, we apply the transformations from Constructions 10, 11, 15, and 16 to all SA or FFN modules. These operations disrupt the orthogonal transformations of either the input or output subspaces, and attempt to restore them using the corresponding orthogonal mappings. This yields eight model variants: $\mathcal{M}_{\text{ablation}}^{SA,out}$, $\mathcal{M}_{\text{restoration}}^{SA,out}$, $\mathcal{M}_{\text{ablation}}^{SA,in}$, $\mathcal{M}_{\text{restoration}}^{SA,in}$, $\mathcal{M}_{\text{ablation}}^{FFN,out}$, $\mathcal{M}_{\text{restoration}}^{FFN,out}$, $\mathcal{M}_{\text{ablation}}^{FFN,in}$, and $\mathcal{M}_{\text{restoration}}^{FFN,in}$. The superscript indicates whether the operation is applied to the input or output subspaces of all weight matrices in SAs or FFNs, while the subscript denotes whether the operation is destructive or restorative. We perform ablation and restoration operations on SAs and FFNs separately, to prevent model collapse caused by excessive cumulative errors when restoring all weight matrices simultaneously. Additionally, this approach enables independent validation of the co-rotation phenomenon between the input-output subspaces of SAs and FFNs, avoiding excessive cumulative errors that could interfere with experimental observations.

## D.1  PERFORMANCE OF DIFFERENT RESTORATION MODELS

We report the performance of all RESTORATION models on GSM8K, MATH-500, MMLU (dev split), and GPQA. All experimental configurations remain consistent with Appendix C.1, specifically with the temperature set to 0.2, top_p to 0.95, and a maximum output token length of 1024. The system prompts are as detailed in Appendix H.1. For each of the four datasets, we measure the results three times and report their pass@1 accuracy (%). All ABLATION models were unable to produce valid outputs, inevitably yielding a pass@1 accuracy of 0% in every evaluation. As these uniformly null results do not provide additional empirical insight, we refrain from reporting them in detail. The complete results are shown in Table 7 and 8.

Most RESTORATION models successfully recover the original performance, validating the consistency of co-rotational alignment between input and output subspaces and confirming Equation 8. We further observe that orthogonal substitutions in the output subspaces are more stable than in the input subspaces: $\mathcal{M}_{\text{restoration}}^{\cdot,in}$ often performs far worse than $\mathcal{M}_{\text{restoration}}^{\cdot,out}$, indicating directional rotational error (Appendix B.2). Errors appear to accumulate along the input-to-output pathway, while reverse elimination can cause collapse. This suggests an inherent asymmetry in co-rotation speed, with one subspace consistently leading the other—an intriguing phenomenon warranting further study.

Table 7: Performance comparison between original and RESTORATION models across GSM8K, MATH-500, MMLU, and GPQA with pass@1 accuracy (%). The "-" indicates model collapse.

| BASE Models | POST Types | RESTORATION Types | GSM8K | MATH-500 | MMLU (dev) | GPQA |
|---|---|---|---|---|---|---|
| Qwen2.5-Math-1.5B | $\mathcal{M}_{\text{Instruct}}$ | $\mathcal{M}_{\text{original}}$ | 85.14±0.14 | 65.47±0.90 | 48.04±0.60 | 30.44±0.36 |
| | | $\mathcal{M}_{\text{restoration}}^{SA,in}$ | 84.53±0.25 | 66.20±0.16 | 41.28±0.44 | 27.69±0.29 |
| | | $\mathcal{M}_{\text{restoration}}^{SA,out}$ | 84.03±0.29 | 66.47±1.79 | 38.25±2.30 | 29.34±2.65 |
| | | $\mathcal{M}_{\text{restoration}}^{FFN,in}$ | 61.54±0.19 | 53.00±0.20 | 31.81±0.41 | 28.79±0.83 |
| | | $\mathcal{M}_{\text{restoration}}^{FFN,out}$ | 84.51±0.18 | 66.07±0.31 | 41.17±0.88 | 22.97±1.10 |
| | $\mathcal{M}_{\text{Reasoning}}$ | $\mathcal{M}_{\text{original}}$ | 62.88±0.59 | 32.73±1.64 | 25.02±0.59 | 7.02±0.44 |
| | | $\mathcal{M}_{\text{restoration}}^{SA,in}$ | 61.54±1.19 | 30.93±0.57 | 29.00±0.44 | 6.75±0.27 |
| | | $\mathcal{M}_{\text{restoration}}^{SA,out}$ | 61.96±1.71 | 32.06±0.25 | 28.30±1.77 | 3.45±1.23 |
| | | $\mathcal{M}_{\text{restoration}}^{FFN,in}$ | 60.60±1.25 | 53.60±0.43 | 25.49±1.07 | 12.81±1.44 |
| | | $\mathcal{M}_{\text{restoration}}^{FFN,out}$ | 76.05±0.71 | 56.46±0.34 | 32.51±3.03 | 16.71±1.81 |
| Qwen2.5-Math-7B | $\mathcal{M}_{\text{Instruct}}$ | $\mathcal{M}_{\text{original}}$ | 95.75±0.12 | 70.06±0.50 | 55.90±0.16 | 27.14±0.49 |
| | | $\mathcal{M}_{\text{restoration}}^{SA,in}$ | 95.15±0.41 | 73.20±0.33 | 55.18±0.18 | 24.85±0.17 |
| | | $\mathcal{M}_{\text{restoration}}^{SA,out}$ | 94.31±0.98 | 72.40±0.53 | 53.10±1.46 | 20.80±1.60 |
| | | $\mathcal{M}_{\text{restoration}}^{FFN,in}$ | 86.10±0.53 | 68.60±1.40 | 54.04±0.61 | 25.07±0.98 |
| | | $\mathcal{M}_{\text{restoration}}^{FFN,out}$ | 94.21±0.86 | 70.93±1.51 | 55.44±3.35 | 25.89±1.44 |
| | $\mathcal{M}_{\text{Reasoning}}$ | $\mathcal{M}_{\text{original}}$ | 62.70±1.05 | 47.60±0.33 | 58.71±0.91 | 14.73±0.97 |
| | | $\mathcal{M}_{\text{restoration}}^{SA,in}$ | 63.21±0.91 | 52.80±0.28 | 58.48±0.65 | 22.99±1.19 |
| | | $\mathcal{M}_{\text{restoration}}^{SA,out}$ | 64.34±2.29 | 50.93±1.36 | 59.06±0.73 | 21.34±0.69 |
| | | $\mathcal{M}_{\text{restoration}}^{FFN,in}$ | 82.46±0.90 | 65.60±2.91 | 48.42±0.70 | 22.71±1.13 |
| | | $\mathcal{M}_{\text{restoration}}^{FFN,out}$ | 58.83±1.66 | 60.07±1.75 | 58.83±0.73 | 20.16±2.42 |
| Llama-3.1-8B | $\mathcal{M}_{\text{Instruct}}$ | $\mathcal{M}_{\text{original}}$ | 34.70±1.24 | 31.46±1.06 | 67.48±0.44 | 21.21±0.29 |
| | | $\mathcal{M}_{\text{restoration}}^{SA,in}$ | 30.15±0.82 | 30.40±0.75 | 65.49±0.43 | 22.32±0.09 |
| | | $\mathcal{M}_{\text{restoration}}^{SA,out}$ | 31.18±1.17 | 33.13±1.70 | 63.74±2.66 | 25.07±2.16 |
| | | $\mathcal{M}_{\text{restoration}}^{FFN,in}$ | 24.13±2.12 | 23.40±1.91 | 59.64±0.93 | 22.61±1.19 |
| | | $\mathcal{M}_{\text{restoration}}^{FFN,out}$ | 43.97±2.06 | 23.26±1.28 | 63.62±2.92 | 21.98±1.29 |
| | $\mathcal{M}_{\text{Reasoning}}$ | $\mathcal{M}_{\text{original}}$ | 60.17±0.07 | 32.73±0.41 | 52.51±1.47 | 11.40±0.17 |
| | | $\mathcal{M}_{\text{restoration}}^{SA,in}$ | 60.30±1.54 | 29.60±0.49 | 42.22±0.59 | 8.77±0.60 |
| | | $\mathcal{M}_{\text{restoration}}^{SA,out}$ | 61.25±0.78 | 34.87±1.17 | 47.13±2.28 | 6.81±1.63 |
| | | $\mathcal{M}_{\text{restoration}}^{FFN,in}$ | 39.87±1.13 | 15.33±3.89 | 38.95±0.70 | 8.99±2.13 |
| | | $\mathcal{M}_{\text{restoration}}^{FFN,out}$ | 38.76±1.09 | 25.00±2.31 | 47.83±1.93 | 7.53±1.50 |
| Qwen2.5-14B | $\mathcal{M}_{\text{Instruct}}$ | $\mathcal{M}_{\text{original}}$ | 94.24±0.29 | 70.53±0.34 | 90.63±0.16 | 36.65±0.36 |
| | | $\mathcal{M}_{\text{restoration}}^{SA,in}$ | 94.09±0.34 | 68.86±0.50 | 88.42±0.29 | 37.60±0.34 |
| | | $\mathcal{M}_{\text{restoration}}^{SA,out}$ | 93.91±1.52 | 73.67±0.92 | 88.07±1.95 | 32.51±0.63 |
| | | $\mathcal{M}_{\text{restoration}}^{FFN,in}$ | 93.63±0.38 | 71.33±0.83 | 82.57±3.58 | 28.89±1.66 |
| | | $\mathcal{M}_{\text{restoration}}^{FFN,out}$ | 94.87±0.64 | 73.60±1.11 | 88.30±0.73 | 34.05±3.40 |
| | $\mathcal{M}_{\text{Reasoning}}$ | $\mathcal{M}_{\text{original}}$ | 70.61±0.46 | 53.13±0.25 | 77.89±0.76 | 19.48±0.55 |
| | | $\mathcal{M}_{\text{restoration}}^{SA,in}$ | 75.72±0.25 | 56.46±0.24 | 76.37±1.85 | 21.94±0.86 |
| | | $\mathcal{M}_{\text{restoration}}^{SA,out}$ | 76.32±1.69 | 56.33±1.70 | 78.83±3.06 | 17.17±1.91 |
| | | $\mathcal{M}_{\text{restoration}}^{FFN,in}$ | - | - | - | - |
| | | $\mathcal{M}_{\text{restoration}}^{FFN,out}$ | 82.15±1.41 | 62.60±1.39 | 76.84±3.35 | 27.06±3.95 |

Table 8: Comparison of average length of output tokens between original and RESTORATION Models across GSM8K, MATH-500, MMLU, and GPQA. The "-" indicates model collapse.

| BASE Models | POST Types | RESTORATION Types | GSM8K | MATH-500 | MMLU (dev) | GPQA |
|---|---|---|---|---|---|---|
| Qwen2.5-Math-1.5B | $\mathcal{M}_{\text{Instruct}}$ | $\mathcal{M}_{\text{original}}$ | 305.01±1.54 | 542.32±1.21 | 402.60±3.13 | 633.82±5.09 |
| | | $\mathcal{M}_{\text{restoration}}^{SA,in}$ | 309.47±15.81 | 523.06±5.87 | 435.36±8.72 | 646.07±6.98 |
| | | $\mathcal{M}_{\text{restoration}}^{SA,out}$ | 287.12±6.99 | 558.05±3.83 | 447.05±8.25 | 631.88±3.64 |
| | | $\mathcal{M}_{\text{restoration}}^{FFN,in}$ | 422.87±25.85 | 587.42±7.66 | 532.19±4.54 | 792.16±7.86 |
| | | $\mathcal{M}_{\text{restoration}}^{FFN,out}$ | 320.65±8.86 | 499.06±13.76 | 443.56±1.18 | 617.73±2.57 |
| | $\mathcal{M}_{\text{Reasoning}}$ | $\mathcal{M}_{\text{original}}$ | 539.82±6.86 | 911.55±5.55 | 619.34±13.82 | 952.00±18.83 |
| | | $\mathcal{M}_{\text{restoration}}^{SA,in}$ | 504.75±24.05 | 916.60±8.58 | 659.16±8.78 | 920.66±13.58 |
| | | $\mathcal{M}_{\text{restoration}}^{SA,out}$ | 518.82±10.24 | 910.68±19.32 | 661.64±13.52 | 968.31±4.19 |
| | | $\mathcal{M}_{\text{restoration}}^{FFN,in}$ | 356.13±11.35 | 692.21±6.48 | 466.14±10.31 | 872.22±16.03 |
| | | $\mathcal{M}_{\text{restoration}}^{FFN,out}$ | 422.74±4.12 | 755.90±5.98 | 502.26±8.86 | 819.93±4.54 |
| Qwen2.5-Math-7B | $\mathcal{M}_{\text{Instruct}}$ | $\mathcal{M}_{\text{original}}$ | 299.46±3.17 | 551.34±4.39 | 372.53±5.91 | 567.34±4.96 |
| | | $\mathcal{M}_{\text{restoration}}^{SA,in}$ | 320.01±9.72 | 561.23±4.63 | 411.70±3.47 | 665.44±10.30 |
| | | $\mathcal{M}_{\text{restoration}}^{SA,out}$ | 307.38±7.85 | 565.77±15.30 | 420.34±9.38 | 672.78±7.23 |
| | | $\mathcal{M}_{\text{restoration}}^{FFN,in}$ | 382.13±8.09 | 552.38±3.86 | 642.14±10.25 | 846.68±8.97 |
| | | $\mathcal{M}_{\text{restoration}}^{FFN,out}$ | 286.25±22.59 | 510.28±11.25 | 345.16±8.75 | 535.02±5.42 |
| | $\mathcal{M}_{\text{Reasoning}}$ | $\mathcal{M}_{\text{original}}$ | 729.16±7.64 | 795.40±9.01 | 514.15±6.91 | 933.15±9.97 |
| | | $\mathcal{M}_{\text{restoration}}^{SA,in}$ | 791.97±21.19 | 617.83±4.76 | 457.57±2.16 | 863.81±2.92 |
| | | $\mathcal{M}_{\text{restoration}}^{SA,out}$ | 796.48±5.62 | 778.33±5.57 | 451.87±7.65 | 877.55±17.99 |
| | | $\mathcal{M}_{\text{restoration}}^{FFN,in}$ | 423.84±8.60 | 809.49±8.49 | 388.25±7.09 | 824.16±3.86 |
| | | $\mathcal{M}_{\text{restoration}}^{FFN,out}$ | 442.44±14.48 | 691.19±7.95 | 444.99±12.73 | 823.32±13.92 |
| Llama-3.1-8B | $\mathcal{M}_{\text{Instruct}}$ | $\mathcal{M}_{\text{original}}$ | 166.47±4.22 | 359.19±6.02 | 35.79±1.43 | 236.35±7.38 |
| | | $\mathcal{M}_{\text{restoration}}^{SA,in}$ | 183.11±8.15 | 324.01±2.05 | 32.51±8.96 | 243.30±10.17 |
| | | $\mathcal{M}_{\text{restoration}}^{SA,out}$ | 169.65±4.65 | 343.88±18.92 | 48.50±6.12 | 254.77±9.58 |
| | | $\mathcal{M}_{\text{restoration}}^{FFN,in}$ | 150.22±3.90 | 278.5±11.29 | 5.33±1.24 | 6.01±1.42 |
| | | $\mathcal{M}_{\text{restoration}}^{FFN,out}$ | 173.32±7.98 | 247.75±13.73 | 11.01±1.41 | 38.74±1.11 |
| | $\mathcal{M}_{\text{Reasoning}}$ | $\mathcal{M}_{\text{original}}$ | 627.14±8.71 | 931.14±14.80 | 721.64±11.13 | 989.41±7.43 |
| | | $\mathcal{M}_{\text{restoration}}^{SA,in}$ | 410.23±6.32 | 833.03±11.39 | 755.99±15.07 | 989.68±3.84 |
| | | $\mathcal{M}_{\text{restoration}}^{SA,out}$ | 431.48±18.15 | 888.37±17.35 | 768.72±11.06 | 998.85±6.39 |
| | | $\mathcal{M}_{\text{restoration}}^{FFN,in}$ | 309.76±24.51 | 953.37±14.71 | 684.11±19.56 | 975.54±17.14 |
| | | $\mathcal{M}_{\text{restoration}}^{FFN,out}$ | 457.27±10.21 | 833.03±11.39 | 672.14±9.32 | 972.02±4.06 |
| Qwen2.5-14B | $\mathcal{M}_{\text{Instruct}}$ | $\mathcal{M}_{\text{original}}$ | 281.95±7.21 | 550.02±6.17 | 89.69±1.18 | 240.16±6.55 |
| | | $\mathcal{M}_{\text{restoration}}^{SA,in}$ | 279.14±7.21 | 444.63±13.24 | 101.63±8.73 | 283.74±9.02 |
| | | $\mathcal{M}_{\text{restoration}}^{SA,out}$ | 182.34±4.57 | 850.45±11.08 | 99.50±5.92 | 275.19±6.80 |
| | | $\mathcal{M}_{\text{restoration}}^{FFN,in}$ | 288.07±14.29 | 442.79±4.03 | 89.41±3.21 | 188.08±5.28 |
| | | $\mathcal{M}_{\text{restoration}}^{FFN,out}$ | 282.67±6.75 | 431.10±6.25 | 120.54±11.45 | 217.08±4.71 |
| | $\mathcal{M}_{\text{Reasoning}}$ | $\mathcal{M}_{\text{original}}$ | 583.01±4.57 | 897.61±8.81 | 487.54±7.68 | 924.63±7.90 |
| | | $\mathcal{M}_{\text{restoration}}^{SA,in}$ | 538.26±6.08 | 844.46±8.89 | 442.49±12.38 | 920.88±4.77 |
| | | $\mathcal{M}_{\text{restoration}}^{SA,out}$ | 518.71±11.25 | 852.79±9.55 | 438.20±4.33 | 912.47±5.40 |
| | | $\mathcal{M}_{\text{restoration}}^{FFN,in}$ | - | - | - | - |
| | | $\mathcal{M}_{\text{restoration}}^{FFN,out}$ | 504.96±8.01 | 863.77±3.59 | 450.66±10.42 | 875.01±11.63 |

## D.2 CKA Analysis of Different Restoration Models

We then feed $N$ input examples into $\mathcal{M}_{\text{post}}$, $\mathcal{M}_{\text{post}}^{\text{ablation}}$, and $\mathcal{M}_{\text{post}}^{\text{restoration}}$, and compute the mean hidden representations $r_{\mathcal{M}}^{(i)}$ for each layer by averaging their outputs (Equation 17):

$$r_{\mathcal{M}}^{(i)} = \frac{1}{N} \sum_{j=1}^{N} \mathcal{M}^{(i)}(T_j) \tag{17}$$

where $T_j$ is the $j$-th input question, and $\mathcal{M}^{(i)}(\cdot)$ denotes the hidden representation produced by the $i$-th Transformer block in model $\mathcal{M}$. We use the first 100 examples from the GSM8K training set for analysis ($N = 100$). We compute the CKA heatmap between the average hidden representations of $\mathcal{M}_{\text{post}}$ and each ABLATION/RESTORATION variant to assess the impact of orthogonal consistency on internal representations. Figure 19 presents our experimental results.

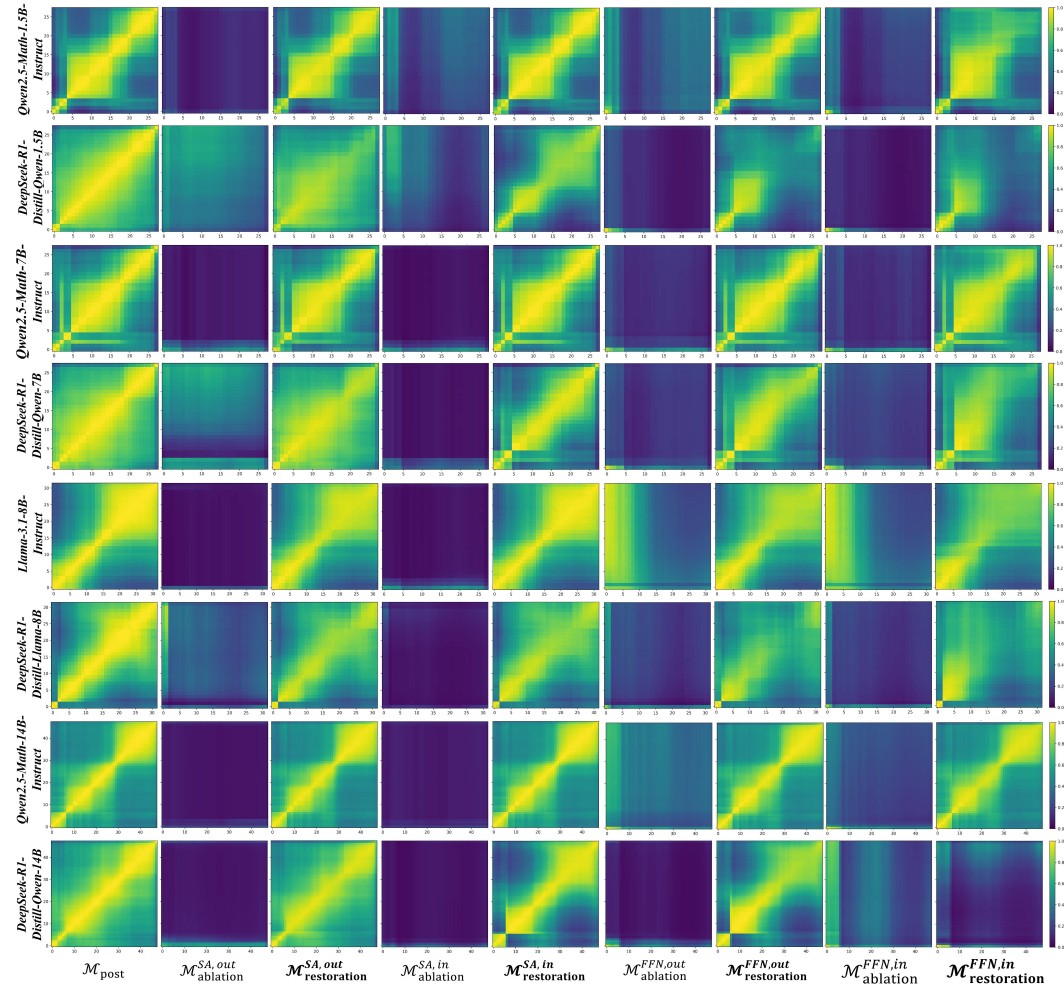

Figure 19: CKA heatmaps generated using $\mathcal{M}_{\text{post}}$ for $\mathcal{M}_{\text{post}}$, $\mathcal{M}_{\text{ablation}}$, and $\mathcal{M}_{\text{restoration}}$. The results indicate that $\mathcal{M}_{\text{Instruct}}$ exhibits stronger orthogonal alignment between input and output subspaces compared to $\mathcal{M}_{\text{reasoning}}$. Additionally, the restoration of orthogonal alignment after perturbation is more robust in the output subspaces than in the input subspaces.

Disrupting either the SAs or FFNs compromises the orthogonal alignment between input and output subspaces, impairing the internal structure of $\mathcal{M}_{\text{post}}$. Restoring this alignment leads to the reemergence of structural symmetry in the CKA heatmaps, indicating a partial recovery of the model's

hidden representations. The weight matrices of $\mathcal{M}_{\text{Instruct}}$ exhibit stronger orthogonal consistency than those of $\mathcal{M}_{\text{reasoning}}$. This is evidenced by the restoration variants of $\mathcal{M}_{\text{Instruct}}$ producing CKA heatmaps that more closely resemble those of $\mathcal{M}_{\text{post}}$. The CKA heatmaps remain only partially reducible, reflecting the fact that orthogonality is preserved only approximately. This observation is further supported by the correction introduced in Equation 13. The restoration process effectively reinstates the original representational geometry, highlighting the critical structural role of orthogonal transformations.

## E  THE STRUCTURAL CHANGES IN A BROADER RANGE OF MODELS

In the main text, as well as in Appendix A, B, C and D, we present a systematic comparison of structural changes in model weights before and after supervised post-training, with a particular focus on the *Qwen* and *LLaMA* families. We also report detailed experimental results that confirm the validity of Equation 8. These findings naturally motivate several follow-up questions:

1. How do reinforcement learning (RL)-based post-training methods influence model weights? From the perspective of parameter space, in what ways do their effects differ from those of supervised post-training, and what implications can be drawn?

2. Would modifications to the model architecture or the adoption of different training strategies affect the generalizability of the observed structural changes?

3. Do other components in LLMs with specific functions (such as normalization layers and output projection heads) follow similar patterns?

This section addresses these questions by extending our analysis to a broader set of models. The subsequent case studies provide strong evidence that the validity of Equation 8 is preserved across diverse settings—including supervised post-training, RL-based post-training, and variations in model architecture or training methodology. The two structural changes identified in the main text thus appear to generalize robustly across these scenarios. Furthermore, we observe that this phenomenon persists throughout the entire post-training phase, indicating the continuity of these two structural changes during post-training, as detailed in Appendix E.4.

### E.1  STRUCTURAL CHANGES IN LLMs INDUCED BY RL-BASED POST-TRAINING

We investigate several state-of-the-art large language models trained with advanced reinforcement learning algorithms, including *AceMath-RL-Nemotron-7B* (Liu et al., 2024), *deepseek-math-7b-rl* (Shao et al., 2024), and *Seed-X-PPO-7B* (Cheng et al., 2025). These models respectively adopt advanced reinforcement learning approaches such as GRPO (DeepSeek-AI et al., 2025) and PPO (Schulman et al., 2017), originate from different research groups, and are built upon diverse training corpora (see Table 10 for details). This diversity in both algorithmic choices and data sources provides inherent support for the generalizability of our subsequent experimental results. We compute the SVSMs between those models and their BASE versions, the $\mathcal{NF}^{(i)}$, as well as the orthogonality matrices of the singular vector (e.g., $I_{orth}^{(0)}$ in the first Transformer block), and present the corresponding visualizations in Figures 20, 21, and 22.

From the SVSM heatmaps and the lower values of $\mathcal{NF}^{(i)}$, we observe that models subjected to RL-based post-training exhibit even more consistent structural changes than those trained with SFT-based post-training. **This strongly suggests that SFT-based and RL-based post-training methods possess a high degree of parameter equivalence, meaning that the effects they impose on model parameters are essentially identical**. Building upon this conclusion, one may infer that RL-based post-training is effectively equivalent to supervised post-training, notwithstanding previous studies (Chu et al., 2025) that have highlighted the ostensibly superior generalization capacity of reinforcement learning algorithms. **We further conjecture that this generalization advantage does not arise from the intrinsic design of RL algorithms themselves, but rather from the diversity of training data generated through reinforcement learning.** For instance, GRPO encourages the model to produce more diverse responses, which are then incorporated into the training process as additional samples. This analysis further explains the effectiveness of Long-CoT distillation. Its training procedure is equivalent to that of RL-based methods, ensuring comparable effects on model

parameters, while its training data are more extensive and diverse than those of instruction tuning, enabling smaller models to achieve reasoning capabilities similar to large-scale RL-based models.

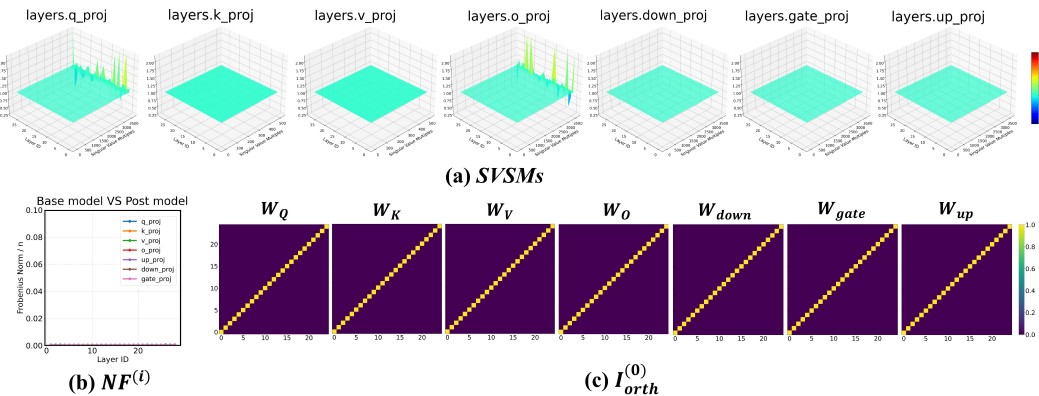

Figure 20: Visualization of structural properties of *AceMath-RL-Nemotron-7B* after post-training. (a) SVSMs reveal that the principal scaling exhibits a near-uniform distribution. (b) $\mathcal{NF}^{(i)}$ provides evidence for the consistent orthogonal transformations of the singular vectors. (c) Orthogonality matrices $I_{orth}^{(0)}$, shown as an example.

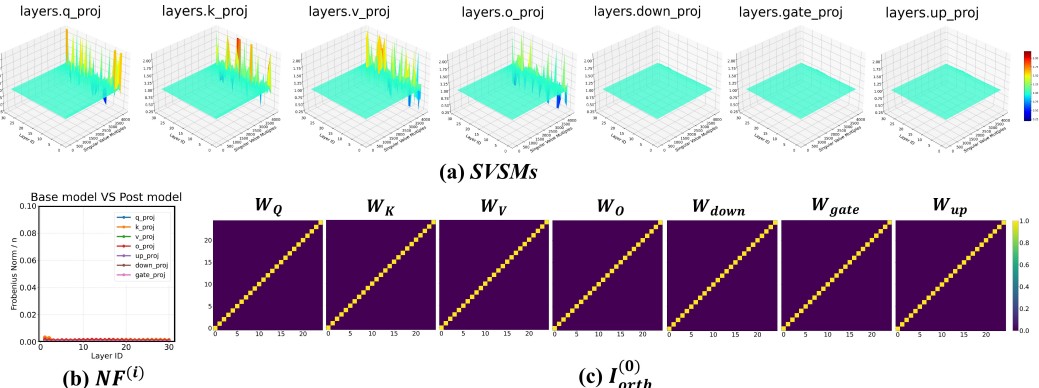

Figure 21: Visualization of structural properties of *deepseek-math-7b-rl* after post-training. The same set of analyses as in Figure 20 is presented, including SVSMs, $\mathcal{NF}^{(i)}$, and orthogonality matrices $I_{orth}^{(0)}$.

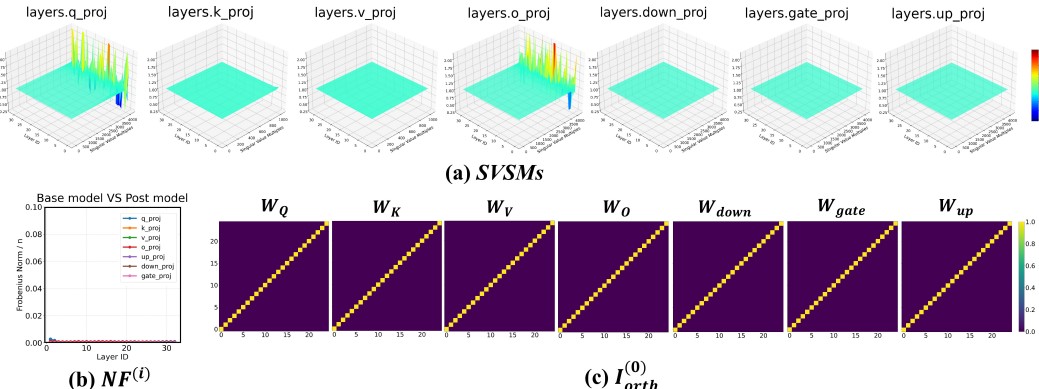

Figure 22: Visualization of structural properties of *Seed-X-PPO-7B* after post-training. The same set of analyses as in Figure 20 is presented, including SVSMs, $\mathcal{NF}^{(i)}$, and orthogonality matrices $I_{orth}^{(0)}$.

### E.2 GENERALITY OF STRUCTURAL CHANGES ACROSS TRAINING STRATEGIES AND ARCHITECTURES

We find that **regardless of architectural modifications or training strategies, LLMs consistently exhibit these two structural changes in their parameters after post-training**. To further examine the universality of this phenomenon, we extend our analysis to *Mistral-7B-Instruct-v0.1* (Albert Q. Jiang et al., 2023), *Gemma-2-2B-it* (Gemma Team et al., 2024), and *MediPhi-Instruct* (Corbeil et al., 2025), each of which incorporates distinct design improvements:

- **For *Mistral-7B-Instruct-v0.1*,** the model incorporates *Sliding Window Attention* (Beltagy et al., 2020) and a *Rolling Buffer Cache*. These mechanisms allow each layer's hidden states to access past information within a window size $W$, which is recursively stacked across layers to effectively expand the attention span. As a result, the model achieves a theoretical attention span of approximately 131K tokens. In practice, these improvements substantially reduce memory consumption and enhance computational efficiency without compromising model quality.

- **For *Gemma-2-2B-it*,** the model architecture integrates *local sliding window attention* (Beltagy et al., 2020) and *global attention* (Luong et al., 2015). Local layers operate with a window size of 4096 tokens, global layers extend to 8192 tokens. A *logit soft-capping* (Bello et al., 2017) mechanism stabilizes training across attention layers and the final layer, with soft_cap values set to 50.0 and 30.0. In post-training, the BASE model firstly undergoes supervised fine-tuning on a mixture of synthetic and human-generated English prompt–response pairs, and then proceeds to *Reinforcement learning with Human Feedback (RLHF)* (Ouyang et al., 2022), guided by a reward model trained on preference data to align behavior with human intent. The resulting models from each stage are averaged, improving stability and overall performance, and producing an instruction-tuned model optimized for both effectiveness and safety.

- **For *MediPhi-Instruct*,** the model still follows a decoder-only Transformer architecture, but the computations of its SAs and FFNs differ from the previously mentioned models. In the case of SAs, given the input $h$, the query ($Q$), key ($K$), and value ($V$) are computed using a single weight matrix $W_{QKV}$:

$$Q, K, V = chunk(QKV), \quad QKV = hW_{QKV} \tag{18}$$

where chunk($\cdot$) splits $QKV$ into $Q, K, V$ along the last dimension. Similarly, for the FFNs, *MediPhi-Instruct* also merges $W_{\text{gate}}$ and $W_{\text{up}}$. As a result, there are only four types of matrices in both the SAs and FFNs, namely $W_{QKV}$, $W_O$, $W_{\text{gate\_up}}$ and $W_{\text{down}}$. In addition to the architectural modifications, *MediPhi-Instruct* also undergoes an SFT-based post-training stage that integrates domain-specific medical knowledge. Similar to other medical instruction-tuned models such as *Aloe* (Gururajan et al., 2024) and *Med42 v2* (Christophe et al., 2024), this stage leverages medical question-answering datasets and benchmark training sets such as *PubMedQA* (Jin et al., 2019), thereby aligning the model more closely with medical reasoning and instruction-following tasks.

More detailed information regarding the aforementioned models will be presented in Table 10. We compute the SVSMs between those models and their BASE versions, the $\mathcal{NF}^{(i)}$, as well as the orthogonality matrices of the singular vector (e.g., $I_{orth}^{(0)}$ in the first Transformer block), and present the corresponding visualizations in Figures 23, 24, and 25.

The flattened SVSM heatmaps and a relatively low value of $\mathcal{NF}^{(i)}$ indicate that, regardless of whether the modifications stem from changes in the model architecture or adjustments in the training strategy, this structural property consistently persists in the linear layers of large models. In other words, **Equation 8 can be employed to characterize the parameter changes of large models before and after post-training**. This provides strong evidence for the universality of such structural transformations and further substantiates the reliability of Equation 8.

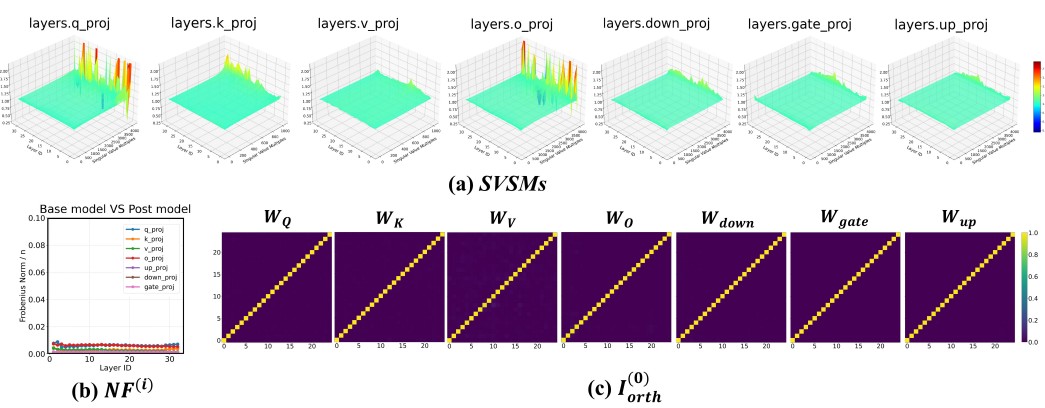

(a) *SVSMs*

(b) $NF^{(i)}$

(c) $I_{orth}^{(0)}$

Figure 23: Visualization of structural properties of ***Mistral-7B-Instruct-v0.1*** after post-training. (a) SVSMs reveal that the principal scaling exhibits a near-uniform distribution. (b) $\mathcal{NF}^{(i)}$ provides evidence for the consistent orthogonal transformations of the singular vectors. (c) Orthogonality matrices $I_{orth}^{(0)}$, shown as an example.

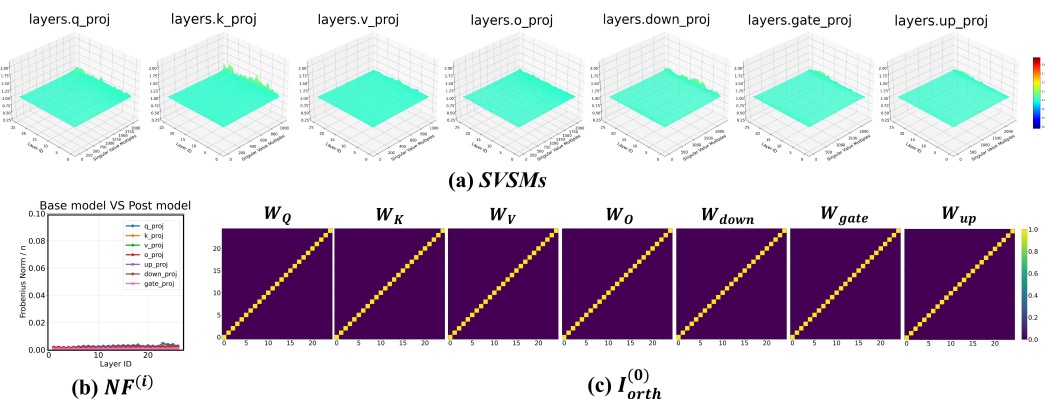

(a) *SVSMs*

(b) $NF^{(i)}$

(c) $I_{orth}^{(0)}$

Figure 24: Visualization of structural properties of ***Gemma-2-2B-it*** after post-training. The same set of analyses as in Figure 23 is presented, including SVSMs, $\mathcal{NF}^{(i)}$, and orthogonality matrices $I_{orth}^{(0)}$.

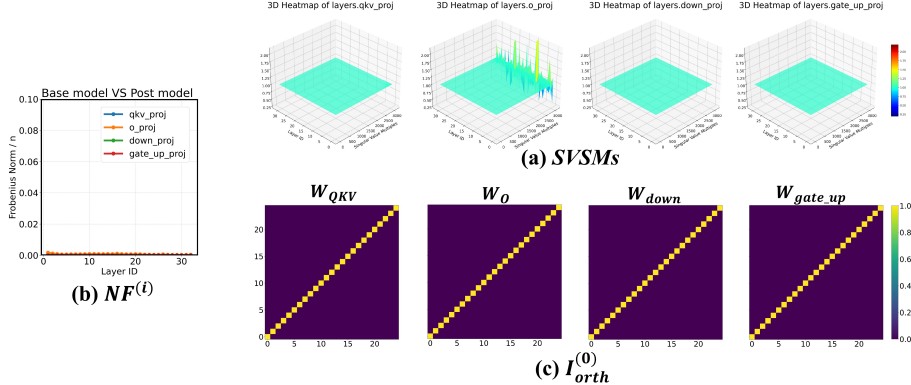

(a) *SVSMs*

(b) $NF^{(i)}$

(c) $I_{orth}^{(0)}$

Figure 25: Visualization of structural properties of ***MediPhi-Instruct*** after post-training. The same set of analyses as in Figure 23 is presented, including SVSMs, $\mathcal{NF}^{(i)}$, and orthogonality matrices $I_{orth}^{(0)}$.

### E.3   STRUCTURAL CHANGES IN OTHER COMPONENTS OF LLMS

We investigate the structural changes of the main linear layers in LLMs in the main text. Although these layers constitute nearly the entire parameter space, other components also play crucial roles. This subsection therefore extends the exploration to the structural changes in the parameter space of functionally important components such as normalization layers and output projection heads. Specifically, we focus on the models listed in Table 9, where each transformer block employs two RMSnorm layers (Jiang et al., 2023b) that serve as the pre-norms for the attention and FFN modules, respectively, to enhance training stability, and an output projection head is added to the final block to convert hidden vectors into a vocabulary distribution.

We visualize the features of normalization layers and output projection heads and unexpectedly find that **these components still roughly adhere to the parameter law described in Equation 8**, yet exhibit subtle differences.

For normalization layers, since the weight often exists as a one-dimensional vector $w$, we consider performing reduced SVD on it:

$$w = a * \sigma * v^T = 1 * ||w|| * \frac{w}{||w||} \tag{19}$$

For a vector $w$, its left singular vector reduces to $\pm 1$ (assumed to be 1), its right singular vector becomes the normalized unit vector $\frac{w}{||w||}$, and its singular value is $||w||$. For the corresponding normalized weight $w_{\text{post}}$ of the POST model, if Equation 8 holds in Equation 20, it implies that the rotation matrix $Q$ of the right singular vector degenerates. In this one-dimensional case, $Q$ becomes a $1 \times 1$ matrix whose sole element is identical to the cosine similarity between $w$ and $w_{\text{post}}$, which is exactly 1. we can derive that:

$$v^T v_{\text{post}} = \frac{w}{||w||} \cdot \left(\frac{w_{\text{post}}}{||w_{\text{post}}||}\right)^T = a^T a_{\text{post}} = 1 \tag{20}$$

We have experimentally verified this point, as shown in Figure 26a. It can be observed that the cosine similarity between the weights of the normalization layers in the POST models and the BASE models remains consistently at 1. **It mathematically proves that the normalization layer of each Transformer block only shows uniform and globally consistent scaling during post-training, rather than the channel-wise selective filtering we anticipated**. However, there is some fluctuation in the scaling of their singular values (norms), as shown in Figure 26b. We speculate that this may be related to the unique function of normalization, which involves dynamically adjusting the expressive capacity of the hidden vectors. When the subspace is fixed, this can only be achieved by globally scaling the vector norms, making it difficult for the norms to maintain uniformly consistent scaling across layers.

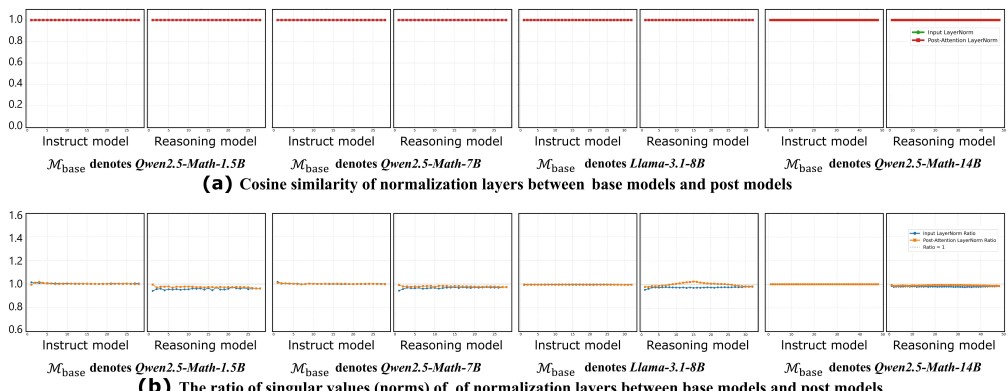

Figure 26: (a) The cosine similarity between the corresponding normalization layers of the BASE models and POST models was calculated. The vast majority of values were equal to 1. (b) The magnitudes of the normalization layers are approximately uniformly scaled but exhibit some fluctuations.

Regarding the output projection heads, we plot the left and right similarity matrices against the overall singular value scaling, as shown in Figure 27. We observe that certain subspaces within the input

and output spaces of this component still do not exhibit strong co-rotation. We hypothesize that this stems from the specific function of output projection heads: since they are responsible for mapping hidden states directly to the vocabulary space, **their parameters are updated directly under the influence of external supervision signals**. As a result, unlike other main linear layers that propagate information through hidden representations, this component experiences greater perturbation of its space during post-training. This makes some of its internal subspaces more susceptible to being reshaped by external supervision, thereby partially hindering appropriate co-rotation. Nevertheless, due to the limited scale of post-training, the structure of the majority of subspaces remains preserved, allowing the output projection heads to largely maintain co-rotation across their subspaces.

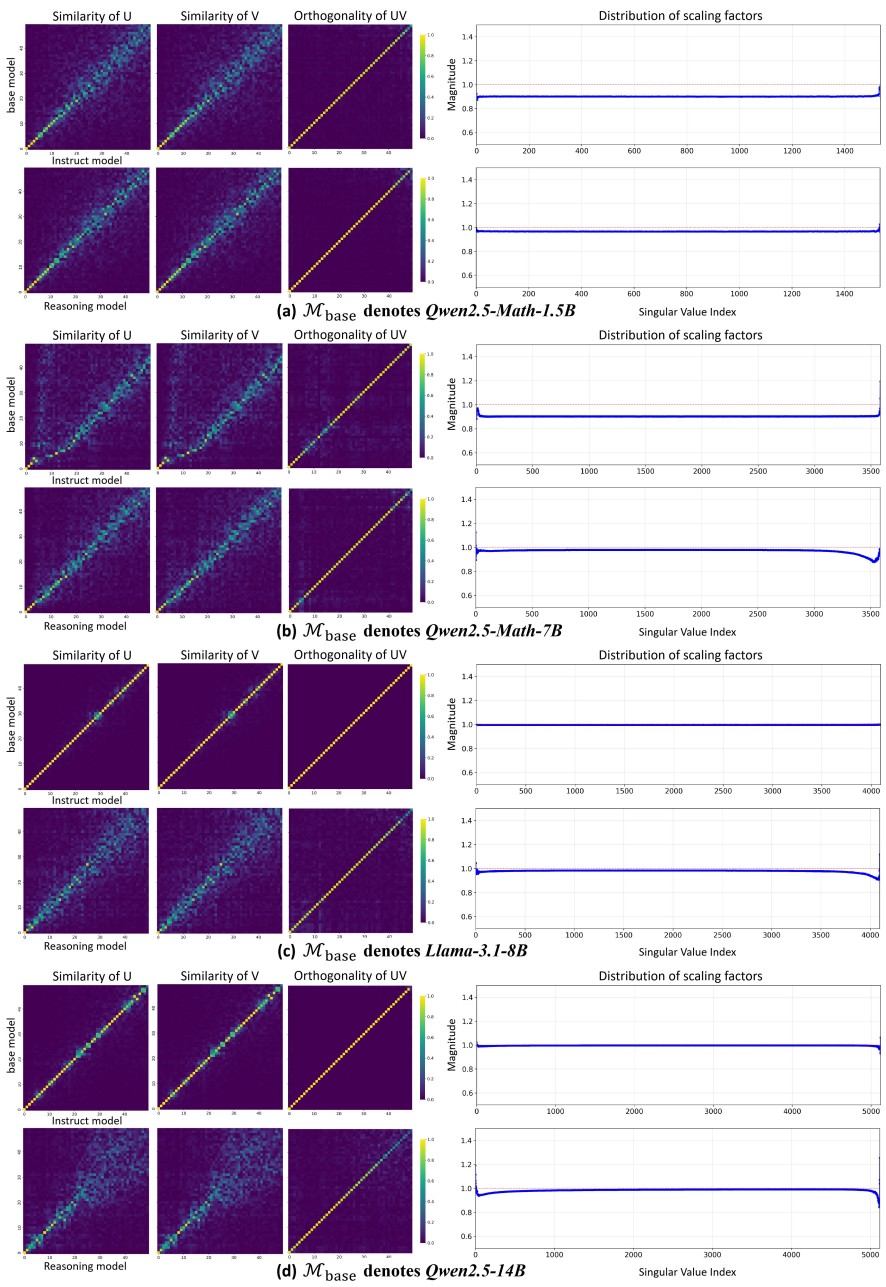

Figure 27: Visualization of the evolution of output projection head properties across model scales. We show the similarity/orthogonality of singular vectors and scaling of singular values before and after post-training.

### E.4 STRUCTURAL CHANGES DURING POST-TRAINING

To determine whether this phenomenon arises during the post-training process or is specific to the final convergence stage, we design a preliminary investigation. We fine-tune the *Qwen2.5-Math-1.5B* model on the complex dataset *s1K-1.1* (Muennighoff et al., 2025) for 5 epochs using supervised learning. Checkpoints are saved after each training epoch. We subsequently compute the $\mathcal{NF}^{(i)}$ metric and the *SVSMs* between these intermediate checkpoints and the original pre-trained *Qwen2.5-Math-1.5B* model. The training configuration is as follows: a maximum sequence length (max_length) of 1024, a batch size of 16, the *AdamW* optimizer (Loshchilov & Hutter, 2019), a learning rate of $2 \times 10^{-5}$, and no gradient accumulation. The evolution of $\mathcal{NF}^{(i)}$ and SVSMs throughout the post-training phase is depicted in Figure 28.

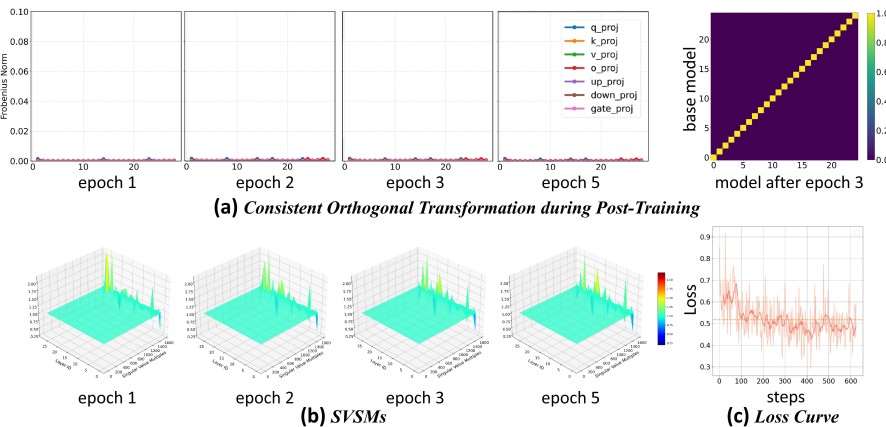

Figure 28: Observation of metrics during the post-training process. (a) presents the $\mathcal{NF}^{(i)}$ for each checkpoint relative to the BASE model, all of which remain at an extremely low level. We also display the $I_{orth}$ of $W_o$ between the first Transformer block of the checkpoints corresponding to epoch 3 and the BASE model, indicating that consistent orthogonal transformations are highly established. (b) shows SVSMs during post-training, and (c) depicts the loss curve, which gradually converges over epochs.

It can be observed that during the training process, the parameter space of the model still closely adheres to the principle of structural transformation mentioned in the main text. This indicates that this phenomenon is an inherent characteristic of the changes in model parameters, rather than a property that only emerges after model convergence.

## F POTENTIAL APPLICATIONS OF OUR FINDINGS

While our primary focus is to characterize the structural transformations of LLMs induced by post-training, our analysis also points to several promising avenues for application. This section outlines a set of illustrative directions, intended not as definitive claims but as conceptual extensions of our findings, with the goal of inspiring future research and advancing the understanding of parameter-level transformations. An overview of these potential applications is provided in Figure 29.

**Fine-grained initialization strategies.** From a post-training perspective, the observed coordinated rotation of singular vectors could inspire more fine-grained weight initialization strategies. A novel approach, termed *PiSSA* (Meng et al., 2024), preserves key components of singular vectors and singular values by initializing them as LoRA weights, while retaining and freezing the remaining singular components. However, *PiSSA* primarily fine-tunes the principal components corresponding to the top-$k$ singular directions. Our analysis of $sim_U$ and $sim_V$ (Figures 10–13) reveals that the singular vectors associated with the largest singular values ($\sigma_{\max}$) exhibit minimal rotation during post-training. This observation implies that the dominant singular components are not the primary targets of fine-tuning. Consequently, as shown in Figure 29a, directing fine-tuning toward the middle-$k$ components rather than the top-$k$ may yield improved performance.

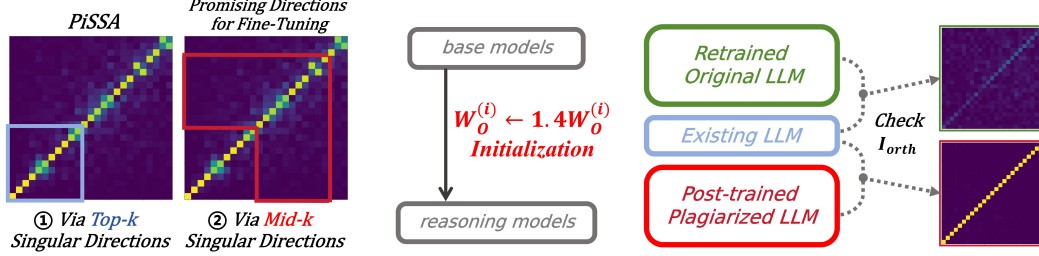

Figure 29: Illustrative overview of potential applications suggested by our findings: (a) fine-grained initialization strategies; (b) accelerated convergence in REASONING models; (c) model fingerprinting based on the detection of $I_{orth}$.

**Potentially accelerated convergence in REASONING models.** We find that the singular value dynamics of REASONING models exhibits unique scaling patterns, particularly in matrices such as $W_O$ (as demonstrated in Figures 2 and 7). Motivated by this observation, one may hypothesize that simple rescaling of pretrained singular values could accelerate convergence during reasoning-oriented training. For instance, initializing $W_O$ as $\alpha W_O$ with $\alpha = 1.4$ provides a lightweight mechanism to impose reasoning-like spectral properties in a single step, potentially reducing the number of iterations required to reach stable performance. While speculative, this perspective highlights the potential to exploit post-training geometry for more efficient model development.

**Model fingerprints under fully parameterized testing.** Appendix B.2 demonstrates that the weight matrices of the same model architecture exhibit markedly different behaviors in $I_{\text{orth}}$ after undergoing distinct pre-training and post-training procedures. This observation provides a practical criterion for distinguishing whether a large language model has been fully developed from scratch or merely obtained through post-training on another model. As illustrated in Figure 29c, this distinction can be achieved simply by measuring the deviation between $I_{\text{orth}}$ and the identity matrix $I$. Importantly, since disrupting the coordinated rotational structure directly leads to model collapse, potential plagiarists cannot eliminate the discrepancy between their model and the original one by deliberately altering this property. Consequently, $I_{\text{orth}}$ serves as a robust and discriminative fingerprint for model identification. Moreover, because this method relies solely on parameter-level analysis, it does not require the design of evaluation datasets as in representation-based fingerprinting approaches such as *REEF* (Zhang et al., 2024a). This line of investigation highlights a promising avenue for safeguarding the intellectual property rights of LLM developers.

While the potential applications discussed above represent relatively straightforward extensions of our observations, their concrete implementation and validation require more rigorous empirical investigation. Nevertheless, we hope that these preliminary intuitions will serve to inspire future research and provide readers with a deeper understanding of the broader implications of our findings for model design, optimization, and interpretability.

## G  PROOF

This section mainly integrates all the mathematical proofs mentioned in the main paper.

### G.1  SINGULAR VALUE SCALING MODULATES THE ATTENTION SCORE

Under near-uniform geometric scaling with singular values, Equation 8 can be restated as $W_{\text{post}} \approx \alpha \cdot U_{\text{post}} \Sigma_{\text{base}} V_{\text{post}}^T = \alpha \cdot W'_{\text{post}}$, which means scaling the singular values has the same effect as scaling the entire weight matrix. We uniformly apply this linear scaling effect to all weight matrices in SAs and FFNs, resulting in the following modified forms of Equations 1 and 2:

$$SA(h) \approx \text{softmax}\left(\frac{\boldsymbol{\alpha^2} \cdot hW'_Q \cdot [K'_{\text{cache}}; hW'_K]^T}{\sqrt{d}}\right) \cdot [V'_{\text{cache}}; hW'_V] \cdot W'_O \cdot \boldsymbol{\alpha\alpha_O} \qquad (21)$$

$$FFN(z) \approx (SwiGLU(z \cdot W'_{gate} \cdot \boldsymbol{\alpha}) \odot (z \cdot W'_{up})) \cdot W'_{down} \cdot \boldsymbol{\alpha^2} \tag{22}$$

The term $\boldsymbol{\alpha}^2$ in Equation 21 corresponds to the inverse of the *attention temperature* (Vaswani et al., 2023), **which can be directly expressed by $T = 1/\boldsymbol{\alpha^2}$.** In SAs, all $\alpha$ except $\alpha_O$ of REASONING models are consistently below 1 after post-training (demonstrated in Table 3), which corresponds to a higher attention temperature. This causes the softmax function to produce more uniformly distributed attention scores, encouraging the model to attend more evenly across all tokens and thereby enhancing its ability to capture global contextual information.

### G.2 TRAINING IS TO PERFORM ORTHOGONAL TRANSFORMATION ON $U$ AND $V$ MATRICES

Considering $\mathcal{M}_A \to \mathcal{M}_B$ as the model training process, left singular vectors of $W_A \in \mathcal{M}_A$, $W_A \in \mathbb{R}^{m \times n}$ can be regarded as performing different transformations $Q_U$:

$$U_B = U_A Q_U \tag{23}$$

We first prove that $Q_U$ is an orthogonal matrix. For $Q_U$, we have:

$$U_A^T U_B = U_A^T U_A \cdot Q_U = I \cdot Q_U = Q_U \tag{24}$$

$Q^T Q = I$ is a necessary and sufficient condition for $Q$ to be an orthogonal matrix. We calculate $Q_U^T Q_U$ then have:

$$Q_U^T Q_U = (U_A^T U_B)^T \cdot (U_A^T U_B) = U_B^T \cdot (U_A U_A^T) \cdot U_B = I \tag{25}$$

Therefore $Q_U$ is an orthogonal matrix.

Through experiments, we observe that $V_A^T V_B$ is nearly identical to $Q_U = U_A^T U_B$. Under the condition that $V_A^T V_B$ is an orthogonal matrix, we aim to prove that the column spaces of $V_A$ and $V_B$ have the same subspace structure, i.e., $\text{col}(V_A) = \text{col}(V_B)$, and that $V_B$ can be obtained from $V_A$ through an orthogonal transformation. Specifically, we will prove that there exists an orthogonal matrix $Q_V$ such that $V_B = V_A Q_V$, where $Q_V = V_A^T V_B$.

Because $V_A$ and $V_B$ have orthonormal columns, $V_A^T V_B$ is an $m \times m$ matrix. We are given that $Q_V = V_A^T V_B$ is orthogonal, hence

$$Q_V^T Q_V = I \tag{26}$$

We define the orthogonal projector onto the column space of $V_A$ as $P_{V_A} = V_A V_A^T$. Decompose $V_B$ into the sum of its projection onto $\text{col}(V_A)$ and the orthogonal remainder:

$$V_B = P_{V_A} V_B + (I - P_{V_A}) V_B = V_A (V_A^T V_B) + (I - V_A V_A^T) V_B \tag{27}$$

Using the definition $Q_V = V_A^T V_B$ this becomes

$$V_B = V_A Q_V + (I - V_A V_A^T) V_B \tag{28}$$

To show $(I - V_A V_A^T) V_B = 0$, consider its Frobenius norm:

$$\|(I - V_A V_A^T) V_B\|_F^2 = \text{tr}\left(V_B^T (I - V_A V_A^T) V_B\right) \tag{29}$$

Expand the trace:

$$\text{tr}\left(V_B^T (I - V_A V_A^T) V_B\right) = \text{tr}\left(V_B^T V_B\right) - \text{tr}\left(V_B^T V_A V_A^T V_B\right) \tag{30}$$

Since $V_B$ has orthonormal columns, $V_B^T V_B = I$, so the first term equals $\text{tr}(I) = m$. For the second term use cyclicity of trace and the definition of $Q_V$:

$$\text{tr}\left(V_B^T V_A V_A^T V_B\right) = \text{tr}\left((V_A^T V_B)^T (V_A^T V_B)\right) = \text{tr}\left(Q_V^T Q_V\right) \tag{31}$$

Because $Q_V$ is orthogonal, $Q_V^T Q_V = I$, hence

$$\text{tr}\left(Q_V^T Q_V\right) = \text{tr}(I) = m \tag{32}$$

Combining these equalities gives

$$\|(I - V_A V_A^T) V_B\|_F^2 = m - m = 0 \tag{33}$$

Therefore

$$(I - V_A V_A^T)V_B = 0 \tag{34}$$

and consequently

$$V_B = V_A Q_V \tag{35}$$

From $V_B = V_A Q_V$ and the fact that $Q_V$ is invertible (orthogonal), the column spaces are identical:

$$\text{col}(V_B) = \text{col}(V_A Q_V) = \text{col}(V_A) \tag{36}$$

This completes the proof. From this perspective, the orthogonal bases utilized during the post-training are essentially **the same as those formed in the BASE models**. This fundamentally implies that post-training does not disrupt the output subspaces constructed during pre-training, strongly suggesting that it constitutes merely a reparameterization process of the BASE models.

### G.3 PROOF OF DIFFERENTLY POST-TRAINED MODELS SHARING A SET OF CONSISTENT ORTHOGONAL TRANSFORMATIONS

We theoretically prove that different POST models initialized from the same pretrained parameters and post-trained on data from different distributions can be transformed into each other through a set of shared orthogonal transformations. Assuming there are two POST models $\mathcal{M}_{\text{post}}, \mathcal{M}'_{\text{post}}$, combining equations 6 and 8, we have:

$$U_{\text{post}} = U_{\text{base}} Q_{\text{post}}, \quad V_{\text{post}} = V_{\text{base}} Q_{\text{post}} \tag{37}$$

$$U'_{\text{post}} = U_{\text{base}} Q'_{\text{post}}, \quad V'_{\text{post}} = V_{\text{base}} Q'_{\text{post}} \tag{38}$$

Substituting Equation 37 into 38, we have:

$$\begin{aligned}
U'_{\text{post}} &= (U_{\text{post}} Q_{\text{post}}^T) \cdot Q'_{\text{post}} = U_{\text{post}} \cdot (Q_{\text{post}}^T Q'_{\text{post}}) \\
V'_{\text{post}} &= (V_{\text{post}} Q_{\text{post}}^T) \cdot Q'_{\text{post}} = V_{\text{post}} \cdot (Q_{\text{post}}^T Q'_{\text{post}})
\end{aligned} \tag{39}$$

Let $Q_{\text{combined}} = Q_{\text{post}}^T Q'_{\text{post}}$, then we observe that:

$$Q_{\text{combined}}^T Q_{\text{combined}} = (Q_{\text{post}}^T Q'_{\text{post}})^T (Q_{\text{post}}^T Q'_{\text{post}}) = I \tag{40}$$

$Q_{\text{combined}}$ is an orthogonal matrix. This directly shows that the conversion from $\mathcal{M}_{\text{post}} \to \mathcal{M}'_{\text{post}}$ can be transformed using an approximately consistent orthogonal matrix $Q_{\text{combined}}$.

This significant corollary reveal that both in-distribution fine-tuning (e.g., instruction tuning) and out-of-distribution fine-tuning (e.g., Long-CoT distillation) induce equivalent transformations in parameter space—specifically, different post-training methods can be mutually converted through shared orthogonal transformations. This equivalence explains why LLMs can be fine-tuned on arbitrary data distributions to improve task-specific performance: **the model's input and output subspaces undergo orthogonal transformations optimized for the target task distribution**.

We believe this insight offers significant promise for future research, particularly in developing methods to mitigate forgetting while preserving adaptability.

## H SETTINGS

This section will delve into more detailed experimental setups, including the different system prompts used for various datasets and the precision of models.

### H.1 SYSTEM PROMPTS

The datasets used in this study include GSM8K, MATH-500, MMLU, and GPQA. Due to time and cost constraints, we limit the output tokens to 1024. If a simple system prompt is used directly, models (particularly REASONING models) often require more tokens to generate correct answers when handling challenging datasets like GPQA. This would result in truncated outputs due to the token limit, preventing us from obtaining valid results for performance evaluation. Therefore, we need to design distinct system prompts for different datasets to facilitate observation of the outcomes.

Additionally, since some datasets provide descriptive ground-truth answers (e.g., GSM8K and MATH-500) while others present multiple-choice questions (e.g., MMLU and GPQA), we must also process the inputs differently across datasets to ensure accurate performance validation.

For the simple dataset (GSM8K) mentioned in this article, the unified system prompt we adopted is:

> ***Please put your final answer within \boxed{}.***

Additionally, all visualization results, including the tracking of attention entropy and the analysis of CKA heatmaps, also adopt this simple system prompt. This is attributed to the fact that during visual analysis of the model, comprehensive output results or testing performance metrics are not required for evaluation purposes.

For hard datasets (MATH-500, MMLU and GPQA) mentioned in this article, the unified system prompt we adopted is:

> ***Please put your final answer within \boxed{} and keep your thought process as short as possible.***

This system prompt will enable us to effectively measure the performance on hard datasets of models within limited token computations.

For the multiple-choice question datasets (MMLU and GPQA) mentioned in this text, the template we adopted for all input prompts is as follows:

> {***ORIGINAL QUESTION***}
> *You have four options, and they are:*
> *A.*{***CHOICE A***}
> *B.*{***CHOICE B***}
> *C.*{***CHOICE C***}
> *D.*{***CHOICE D***}
> *Please select the correct option and just give A, B, C or D. For example, if you think the answer is A, just give \boxed{A} as the answer.*

This template design enables us to use the same validation evaluator for both multiple-choice and open-ended answer datasets, thereby reducing our engineering complexity.

## H.2    INTRODUCTION TO THE MODELS AND MODEL PRECISION SETTINGS

The different POST versions corresponding to the different BASE models are shown in Table 9 and 10. All experiments in this paper were conducted on two NVIDIA A100 GPUs with 40GB of memory each.

Table 9: Different POST versions of different BASE models used in Appendix A, B, C and D.

| BASE Models | POST Types | POST Models | Developer |
|---|---|---|---|
| *Qwen2.5-Math-1.5B* | $\mathcal{M}_{\text{Instruct}}$ $\mathcal{M}_{\text{reasoning}}$ | *Qwen2.5-Math-1.5B-Instruct* *DeepSeek-R1-Distill-Qwen-1.5B* | *Qwen Team* *DeepSeek* |
| *Qwen2.5-Math-7B* | $\mathcal{M}_{\text{Instruct}}$ $\mathcal{M}_{\text{reasoning}}$ | *Qwen2.5-Math-7B-Instruct* *DeepSeek-R1-Distill-Qwen-7B* | *Qwen Team* *DeepSeek* |
| *Llama-3.1-8B* | $\mathcal{M}_{\text{Instruct}}$ $\mathcal{M}_{\text{reasoning}}$ | *Llama-3.1-8B-Instruct* *DeepSeek-R1-Distill-Llama-8B* | *Meta* *DeepSeek* |
| *Qwen2.5-14B* | $\mathcal{M}_{\text{Instruct}}$ $\mathcal{M}_{\text{reasoning}}$ | *Qwen2.5-14B-Instruct* *DeepSeek-R1-Distill-Qwen-14B* | *Qwen Team* *DeepSeek* |

All $\mathcal{M}_{\text{base}}$ and $\mathcal{M}_{\text{Instruct}}$ use BF16 parameter storage, while $\mathcal{M}_{\text{reasoning}}$ employ FP32. To address potential precision truncation, we consistently convert all parameters to FP32 before experimentation, ensuring unified numerical precision throughout our evaluations.

Table 10: Different POST versions of different BASE models used in Appendix E.

| BASE Models | POST Models | post-training method | Developer |
|---|---|---|---|
| *DeepSeek-R1-Distill-Qwen-7B* | *AceMath-RL-Nemotron-7B* | RL-based (GRPO) | *Nvidia* |
| *deepseek-math-7b-base* | *deepseek-math-7b-rl* | RL-based (GRPO) | *Deepseek* |
| *Seed-X-Instruct-7B* | *Seed-X-PPO-7B* | RL-based (PPO) | *ByteDance* |
| *Mistral-7B-v0.1* | *Mistral-7B-Instruct-v0.1* | SFT-based | *Mistral AI* |
| *gemma-2-2b* | *gemma-2-2b-it* | SFT-based | *Google* |
| *MediPhi* | *MediPhi-Instruct* | SFT-based | *Microsoft* |

# I USE OF LARGE LANGUAGE MODELS

We acknowledge the use of LLMs for minor editorial assistance. Specifically, **LLMs were only employed to polish the language and correct grammatical errors in the manuscript**. No LLMs were involved in generating the research ideas, designing experiments, conducting analyses, or drawing conclusions.

