# OpenReview forum: "Understanding Post-Training Structural Changes in Large Language Models"
_ICLR.cc/2026/Conference — Submitted to ICLR 2026_

### Official Review · Reviewer_YWpN · 2025-10-17

**Soundness:** 3
**Presentation:** 3
**Contribution:** 4
**Rating:** 8
**Confidence:** 3

**Summary:**

This paper investigates how the weights of large language models change under instruction fine-tuning and Long-CoT distillation. The authors propose that the post-finetuning weight matrix can be approximately expressed as
$$ W_{\text{post}} = U_{\text{post}} \Sigma_{\text{post}} V_{\text{post}}^{T} \approx (U_{\text{base}} Q)   \alpha \Sigma_{\text{base}} (V_{\text{base}} Q)^{T} $$
suggesting that fine-tuning primarily induces a structured transformation on the singular vectors of the base model, characterized by a rotation $Q$ and a rescaling $\alpha \Sigma_{\text{base}}$.

**Strengths:**

1) The decomposition provides a clear and interpretable picture of how instruction fine-tuning modifies model representations.

2) The scaling relation $\alpha \Sigma_{\text{base}}$ is convincingly shown in the combination of Figure 2 and Table 1.

3) The claim that a single rotation matrix $Q$ is sufficient to describe the transition is less visually convincing in Figure 3b, but Table 2 provides substantial additional support.

**Weaknesses:**

I am not convinced by Figure 3b, as, to my understanding, the simple assumption that fine-tuning leads to only small changes in the weight may produce this figure.

I would suggest adding the zero values to Table 2 to enhance readability.

**Questions:**

No Questions.

---

> ### Author Response · Authors · 2025-11-19
>
> We sincerely appreciate your favorable assessment and helpful suggestions. We have implemented all your recommended revisions, as detailed in our point-by-point responses below.
>
> ---
>
> >***W1**: I am not convinced by Figure 3b, as, to my understanding, the simple assumption that fine-tuning leads to only small changes in the weight may produce this figure.*
>
> We fully agree with your point. If post-training only induces slight modifications to the weights, the resulting rotation of the input/output subspaces would also be minimal, also leading to very small values of $NF$. **To clarify this distinction, we have added a new black line in the original Figure 3b, which illustrates the $NF_{sim}$ between the left singular vector similarity matrix $sim^{(i)}_U$ and the identity matrix $I$.** If the small $NF$ values were due to insufficient training of the LLMs, then $NF_{sim}$ should also remain very low (since $sim^{(i)}_U$ would be close to an identity matrix if the model were barely trained). In practice, however, we observe that this curve consistently maintains a high value, indicating that the trained model has substantially deviated from the original one. In this case, the lower $NF$ values strongly support the conclusion that the input and output subspaces of the large model undergo coordinated rotation.
>
> We have elaborated on these points in Lines 279–300 and supplemented the corresponding Figure 9. We believe this addition further strengthens the credibility of Figure 3b. We sincerely appreciate your valuable and important feedback.
>
> ---
>
> >***W2**: I would suggest adding the zero values to Table 2 to enhance readability.*
>
> Thank you for your suggestions! We have added the description of $M_{ablation}$ in Table 2, and distinguished the data rows for $M_{ablation}$ and $M_{restoration}$ using red and green cells, respectively, to improve the readability and interpretability of the table. Similarly, we have applied the same treatment to Table 1, highlighting the data of $M_{replaced}$ in green to enhance clarity.
>
> ---
>
> We are truly thankful for your comments, which have allowed us to present a more polished revisions. We trust that the revisions meet with your approval and await your response.

---

> > ### Comment · Reviewer_YWpN · 2025-11-20
> >
> > Thank you for your detailed rebuttal. I have no further questions.

---

> > > ### Author Response · Authors · 2025-11-20
> > >
> > > We appreciate your feedback. Thank you once again for your valuable comments on our manuscript.

---

### Official Review · Reviewer_1Wyf · 2025-10-23

**Soundness:** 2
**Presentation:** 2
**Contribution:** 2
**Rating:** 4
**Confidence:** 4

**Summary:**

The paper analyzes how post-training methods (instruction tuning and long chain-of-thought distillation) change the weight matrices inside large language models. Using SVD on attention and FFN layers, the authors report two regularities: (1) singular values are scaled by nearly uniform factors across layers, and (2) the left and right singular vector bases undergo highly consistent, shared rotations.

**Strengths:**

**Clear conceptual framing**: The paper offers a simple and intuitive way to describe post-training dynamics in LLMs. This conceptual framing is easy to understand and potentially useful for future theoretical or empirical work.

**Extensive and systematic experimentation**: The authors support their claims with a wide range of experiments across multiple model families and sizes.

**Weaknesses:**

**Potential over-interpretation**: Because post-training typically induces only small parameter drift compared to pre-training, the observed “near-identity rotation” and “near-uniform scaling” could be calibration artifacts rather than evidence of a deep mechanism. The evidence also leans on visual flatness and relative comparisons (SVSM, NF).

**Mechanistic gap**: The work characterizes what changes (rotation + scaling) but says little about why.

**Lack of actionable insight**: The paper does not concretely demonstrate how these observations can be operationalized. Although it hints at possible implications, the ideas remain at the suggestion level.

**Questions:**

Minor Errors:

Line 91: strongly suggests that -> strongly suggesting that, the subspaces structure -> the subspace structure

Line 81: DeepSeek R1-Distill-Qwen-1.5B -> DeepSeek-R1-Distill-Qwen-1.5B

Line 278: show -> shows

Line 281: are closed to -> are close to

Lines 336, 340, 399, 1220, and 1461: accuracy(%) -> accuracy (%)

Line 620: Svdqunat -> SVDQuant

Line 167: The notation seems sloppy. Here the symbols refer to single model instances, but they are used as if they were sets. This could be clarified to avoid confusion.

---

> ### Author Response · Authors · 2025-11-19
> **Response [1/3]**
>
> We thank you for your review and for highlighting the importance of clearly defining the scope and value of our work. We agree that this is crucial, and we appreciate the chance to provide further clarification in our responses and through revisions to the text.
>
> ---
>
> >***W1**: Potential over-interpretation: Because post-training typically induces only small parameter drift compared to pre-training, the observed “near-identity rotation” and “near-uniform scaling” could be calibration artifacts rather than evidence of a deep mechanism. The evidence also leans on visual flatness and relative comparisons (SVSM, NF).*
>
> Your concern is well-founded and rigorous. Therefore, we would like to share some additional insights.
>
> 1. Regarding co-rotation, **we have actually observed minor co-rotation phenomena across different base models** (as shown in Figure 14 of the revised manuscript). This led us to propose a simpler hypothesis (Appendix B.2): the difference between post-training and pre-training mainly stems from the accumulation of co-rotation errors. During post-training, LLMs tend to prioritize co-rotating subspaces to fit the data distribution, at which point the co-rotation is minimal and negligible. However, when the existing rotation can no longer encode richer data distributions, the accumulated error forces certain subspaces of the LLM to be reshaped to adapt to the new pre-training data distribution. **Co-rotation should be regarded as a fundamental and important behavior linking parameter space and data distribution, rather than as a calibration artifact in the early training stage;** otherwise, such subtle co-rotation would not occur during pre-training. From this perspective, explaining this phenomenon must involve a deeper underlying mechanism.
>
> 2. As for uniform scaling of singular values, if it were merely a calibration artifact of post-training compared to pre-training (i.e., an insufficient phase), then two distinct post-training methods (instruction tuning and Long-CoT distillation) should also exhibit insufficient scaling. **However, the anomalous increase in the singular values of the $W_O$ matrix in reasoning models contradicts this: after Long-CoT distillation, the singular values of $W_O$ increased by nearly 1.4 times (equivalent to a 40% magnitude impact!). Yet, despite this substantial effect, the singular values still maintained uniform scaling.** The "conflict" inherent in this phenomenon highlights the uniqueness of post-training. Under the same base model, instruction tuning does not exhibit this characteristic, indicating that the training data distribution induces different structural changes in post-training, rather than being a potential calibration artifact. The emergence of these two phenomena necessitates a possible underlying mechanism to unify and explain them.
>
> Furthermore, we have noted your questions regarding our methodology. In fact, the visualization results are intended to help readers better understand our study intuitively. The causal experiments in Section 5 are the core of our paper. These experiments validate the correctness of Eq. 8 through parameter manipulation and performance observation. Additionally, we extensively employ relative comparison methods in our work because this approach allows us to decouple the changes brought by post-training from the original complex model parameters, making it easier to directly observe the transformation dynamics across the singular space.
>
> ---

---

> > ### Author Response · Authors · 2025-11-19
> > **Response [2/3]**
> >
> > ---
> >
> > >***W2**: Mechanistic gap: The work characterizes what changes (rotation + scaling) but says little about why.*
> >
> > We sincerely appreciate your insightful comments. Accordingly, we wish to further clarify the positioning of this work:
> >
> > *Our primary contribution lies in verifying the universality, stability, and consistency of these two structural transformations across different model families, scales, and various post-training methods, while also providing a new mathematical framework (Eq 8) for understanding the evolution of parameter spaces—challenging the conventional view of treating parameter space as a disordered black box.*
> >
> > We argue that **discovering general patterns is a necessary prerequisite** for the development of any mechanistic theory. Without a solid empirical characterization of these consistently occurring changes, attempts to investigate their underlying causes would lack a firm foundation. **Our main contribution is the rigorous identification and quantification of these structural phenomena through extensive measurements and causal experiments (Section 5 and Appendices),** ensuring the generalizability and robustness of the observed phenomena.
> >
> > At the same time, we have taken initial steps toward mechanistic reasoning. For instance, in W1 and as discussed with you, we considered the cumulative effect of co-rotation errors and the potential role of data distribution in shaping the singular space, along with the key insight that *post-training can be viewed as a reparameterization process within the pretrained subspace* (Eq 8). Although this remains a simple and direct insight, we believe it will provide important inspiration for future work and offer valuable guidance for subsequent research directions, as mentioned in the Limitations section.
> >
> > We hope the above clarifications help in better understanding the intended scope and core contributions of our study.
> >
> > ---

---

> > > ### Author Response · Authors · 2025-11-19
> > > **Response [3/3]**
> > >
> > > >***W3**: Lack of actionable insight: The paper does not concretely demonstrate how these observations can be operationalized. Although it hints at possible implications, the ideas remain at the suggestion level.*
> > >
> > > We thank you for the insightful comments. We fully acknowledge the concern regarding "actionability." However, we would like to emphasize that **the primary objective of this work is not to propose a task-specific algorithm, but rather to leverage extensive experiments to uncover the intrinsic geometric structure governing parameter space changes during post-training of LLMs, and to provide an approximate description of the training dynamics via Eq 8.** Formally characterizing the orderly evolution within the parameter space constitutes the core contribution of our study.
> > >
> > > We believe that establishing an accurate mathematical framework to describe a particular class of universal phenomena is a key value of our paper. **This approach aligns with the well-regarded research paradigm in LLM interpretability.** For instance, analyzing knowledge storage mechanisms has enhanced the understanding of factuality and refusal behaviors [1], the discovery of knowledge circuits has provided a structural basis for subsequent knowledge editing [2], and studies on reasoning graph topology have offered new analytical perspectives for LRMs [3]. **While these works did not directly propose applied algorithms, their theoretical contributions have substantially supported the experimental foundations of later practical methods.**
> > >
> > > Furthermore, **our discussions on potential applications are not speculative but are strictly grounded in our experimental findings.** For example, the stability of singular vector similarity along the direction of the largest singular value directly inspired our proposed improvement for PiSSA initialization [4]. The structural differences in $I_{orth}$ and $NF$ between pre-training and post-training naturally suggest their use as candidate features for model fingerprinting. These are not generic "potential applications" but direct deductions from our experimental observations, which we are already validating as part of our ongoing research.
> > >
> > > We sincerely appreciate your valuable feedback, which has prompted us to clarify the potential contributions of our work to the community. Accordingly, in the revised manuscript, we have added descriptions of the scope and impact of post-training studies in Lines 475–480 and Lines 465–467 (highlighted in blue) to better underscore the value of our research.
> > >
> > > [1] How Post-Training Reshapes LLMs: A Mechanistic View on Knowledge, Truthfulness, Refusal, and Confidence, accepted to *COLM 2025*.
> > >
> > > [2] Knowledge Circuits in Pretrained Transformers, accepted to *NeurIPS 2024*.
> > >
> > > [3] Topology of Reasoning: Understanding Large Reasoning Models through Reasoning Graph Properties, accepted to *NeurIPS 2025*.
> > >
> > > [4] PiSSA: Principal Singular Values and Singular Vectors Adaptation of Large Language Models, accepted to *NeurIPS 2024*.
> > >
> > > ---
> > >
> > > >***Q1**: Some writing mistakes.*
> > >
> > > We sincerely appreciate your meticulous feedback on the writing of our manuscript! We have carefully revised the text to address the writing issues and refined certain expressions for greater precision (highlighted in blue). Thank you once again for your thorough review of our paper.
> > >
> > > ---
> > >
> > > We sincerely thank you for your thoughtful comments. We hope our point-by-point responses have adequately clarified our paper's key contributions. We are ready to address any remaining points.

---

### Official Review · Reviewer_DYuD · 2025-10-28

**Soundness:** 3
**Presentation:** 2
**Contribution:** 2
**Rating:** 4
**Confidence:** 4

**Summary:**

This paper presents a systematic SVD analysis of LLMs before and after post-training (instruction tuning, Long CoT training). The authors identify two recurring structural changes: 1) a near-uniform geometric scaling of singular values, and 2) highly consistent orthogonal transformations applied to the left and right singular vectors. They argue that the orthogonal transformation is the core mechanism, while singular value scaling is a secondary, temperature-like effect.

**Strengths:**

1. The empirical evaluation is extensive, including multiple model families (Qwen, LLaMA), scales (1.5B to 14B). The scale of the validation is a strength.

2. The discovery of coordinated orthogonal transformations (U_post = U_base Q, V_post = V_base Q) is novel and interesting. The ablation experiments showing that disrupting this coordination causes catastrophic failure are convincing evidence of its importance.

3. The paper is generally well-written and the proposed mathematical framework is clear and intuitive.

**Weaknesses:**

1. The paper only describes what happens but fails to explain why it happens. The work establishes a strong correlation between the structural patterns and post-training, but does not prove that these patterns are the cause of improved performance.  The deepth of the analysis is constrained, thus making is below the bar of acceptance.

2. Besides, the findings don't motivate any methods which also limits its practical value. "To perform post-training, one must learn a coordinated orthogonal transformation."  The paper does not provide a new, more efficient algorithm, a better initialization scheme, or a method to control this process. The "potential applications" in Appendix F are speculative and lack empirical validation.

3. The Singular Value Scaling Argument is Weak: Dismissing singular value scaling as a "secondary temperature effect" undermines the work's own findings. The data in Table 1 and Table 5 shows that for reasoning models (M_reasoning), replacing singular values often improves performance. This suggests the scaling is not just a passive effect but is actively detrimental in some cases, and its "correction" is beneficial.

**Questions:**

NA

---

> ### Author Response · Authors · 2025-11-19
> **Response [1/3]**
>
> We sincerely thank you for your thorough review and for highlighting the fundamental questions about our study's significance. We have carefully reflected on these points and provide a detailed response below to better justify our work's positioning.
>
> ---
>
> >***W1**: The paper only describes what happens but fails to explain why it happens. The work establishes a strong correlation between the structural patterns and post-training, but does not prove that these patterns are the cause of improved performance.*
>
> Thank you for raising this insightful question. We would like to further clarify the positioning and academic contributions of this work. Our primary contribution lies in the discovery and **robust validation** of a fundamental phenomenon that has been underexplored, namely, how post-training shapes the parameter space of LLMs, along with the identification of two important structural principles therein. This establishes a new mathematical framework (Eq 8) for understanding the evolution of the parameter space, and **challenges the conventional view that treats the parameter space as an unstructured black box.**
>
> **We argue that rigorously describing a universal phenomenon is a necessary foundation for building any causal theory**. If the phenomenon itself is fragile, subsequent mechanistic interpretations would lack a reliable basis. Therefore, the main depth of this work is reflected in the extensive and rigorous visualization experiments and causal analyses (Section 5 and Appendix), which systematically validate the universality and robustness of the identified principles and the correctness of Eq 8. This provides the community with a solid starting point.
>
> Additionally, we wish to clarify that performance improvement is not the focus of our study. Rather, our core interest is in *how post-training shapes the parameter structure*. We observe that disrupting the co-rotation leads to subspace collapse within LLMs, while restoring rotation in the symmetric space can recover the LLMs' output (i.e., the behaviors learned during post-training). This strongly supports that co-rotation is an essential mechanism through which post-training influences the parameter space, **while performance serves only as a supporting indicator for validating the phenomenon.**
>
> We hope these clarifications help convey the original intention and academic value of our study.
>
> ---

---

> > ### Author Response · Authors · 2025-11-19
> > **Response [2/3]**
> >
> > ---
> >
> > >***W2**: Besides, the findings don't motivate any methods which also limits its practical value ... ,The "potential applications" in Appendix F are speculative and lack empirical validation.*
> >
> > We thank the reviewer for raising this important point. We would like to clarify that, the main contribution of **our work is not focused on the development of task-specific application, but rather on deepening the community's understanding of parameter space changes in LLMs during post-training**. Therefore, we have conducted extensive experiments to validate the correctness and effectiveness of the proposed framework (Eq 8), rather than developing a specific algorithm.
> >
> > We believe that these interesting scientific findings and detailed experimental validations constitute the value of our paper, *which aligns with the respected research paradigm in LLM interpretability*. For instance, [1] provided important guidance for studying knowledge storage, factuality, refusal, and confidence in LLMs; [2] identified knowledge circuits in LLMs through observations on GPT-2 and TinyLlama, enhancing the community's understanding of knowledge editing; and [3] offered new perspectives on retrospective reasoning in LRMs by studying properties of their reasoning graphs. These works demonstrate that fundamental discoveries about internal model mechanisms are key to advancing the field, even when they do not propose specific application algorithms.
> >
> > Moreover, **the potential applications proposed in our Appendix F are not baseless conjectures.** It is based on our observation of the stability of  $sim_U$  and  $sim_V$ in the singular vectors corresponding to the largest singular values that we suggest a possible improvement to the initialization of PiSSA [4]-specifically, using mid-$k$ singular vectors instead of top-$k$ ones for initialization. Additionally, due to the distinct behaviors of $I_{\text{orth}}$ and $\text{NF}$ between pre-training and post-training, these two metrics naturally serve as feature carriers for full parameterized model fingerprints. The content presented in our potential applications is grounded in solid research.
> >
> > We thank you again for the valuable feedback. Your comments have also prompted us to re-examine the manuscript and more clearly articulate the contributions to the community. Accordingly, in the revised manuscript, we have added descriptions of the contributions of post-training in Lines 475–480 and Lines 465–467 (highlighted in blue) to better clarify the value of our research.
> >
> > [1] How Post-Training Reshapes LLMs: A Mechanistic View on Knowledge, Truthfulness, Refusal, and Confidence, accepted to *COLM 2025*.
> >
> > [2] Knowledge Circuits in Pretrained Transformers, accepted to *NeurIPS 2024*.
> >
> > [3] Topology of Reasoning: Understanding Large Reasoning Models through Reasoning Graph Properties, accepted to *NeurIPS 2025*.
> >
> > [4] PiSSA: Principal Singular Values and Singular Vectors Adaptation of Large Language Models, accepted to *NeurIPS 2024*.
> >
> >
> > ---

---

> ### Author Response · Authors · 2025-11-19
> **Response [3/3]**
>
> ---
>
> >***W3**: The Singular Value Scaling Argument is Weak: Dismissing singular value scaling as a "secondary temperature effect" undermines the work's own findings ... This suggests the scaling is not just a passive effect but is actively detrimental in some cases, and its "correction" is beneficial.*
>
> We sincerely appreciate your insightful observation and would like to take this opportunity to clarify and supplement our perspective.
>
> We attribute the scaling of singular values to a *secondary effect analogous to temperature adjustment*, **based on the premise that such scaling does not inherently alter the output structure of LLMs.** For instance, even if we manually scale the singular values of the $W_O$ matrix in instruct models by a factor of 1.4 (matching the scaling observed in reasoning models),  the instruct models do not develop a similar output structure (such as exhibiting retrospective reflection or generating "wait" tokens). Instead, only the number of output tokens changes.
>
> Since we have mathematically shown that this linear scaling is equivalent to adjusting the attention temperature (Eq 21 in the revised manuscript), and since the observable effect of attention temperature is also a change in output token count, we consider this phenomenon as a form of temperature regulation.
>
> **The performance improvement observed in reasoning models, as you keenly noted, may be viewed as a secondary effect of the change in token count.** As shown in Table 6, reasoning models derived from Qwen2.5-Math-7B show significant performance improvements on GSM8K and MATH-500, accompanied by a marked decrease in average token count. Conversely, reasoning models based on Llama-3.1-8B exhibit no performance gain on tasks such as MMLU and GPQA, with token counts comparable to or exceeding the base model. **This discrepancy stems from the inference token limit: shorter outputs reduce truncation, facilitating correct answer matching, whereas longer outputs risk truncating key content, causing evaluation failures.** Absent this constraint, the actual performance gap would be negligible. This interpretation is consistent with instruct models (Tables 5 and 6), which use fewer and more stable token counts, avoiding truncation and maintaining consistent performance across datasets.
>
> We have added descriptions regarding token count and performance in Lines 356–363 of the revised manuscript, highlighted in blue, hoping that this will better elucidate our insights.
>
> ---
>
> We extend our sincere thanks for your comments, which have helped us better articulate our core contributions. We hope the rebuttal adequately reflects this. Please let us know if anything requires elaboration.

---

### Official Review · Reviewer_Gn1W · 2025-10-30

**Soundness:** 3
**Presentation:** 3
**Contribution:** 3
**Rating:** 6
**Confidence:** 4

**Summary:**

The paper aims to characterize how base LLMs change their weights during post-training (instruction tuning and long chain-of-thought). For that, SVD decomposition of weight matrices is used. The paper shows that base weights and post-trained weights differ only up to an orthogonal transformation of the left and right eigenvectors and a near-uniform scaling of the singular values. Furthermore, the near-uniform singular value scaling is proposed to act as a temperature parameter for attention matrices, where a higher scaling induces lower attention entropy. Then, the effect of singular value scaling and orthogonal transformation is assessed by reconstructing post-trained models from base models, showing comparable performance. Finally the representational similarity of base and reconstructed LLMs is assessed using Centered Kernel Alignment (CKA).

**Strengths:**

* The approach is simple and models very accurately the parameter changes in LLMs during post-training.
* The claims are largely supported by the experiments, showing how near-uniform scaling and orthogonal rotations underlie post-training.
* The findings have a big impact on understanding post-training and its learning dynamics, potentially allowing a simplification of current post-training techniques.

**Weaknesses:**

* The results in Table 1 are a bit circular, first a scaling factor is estimated, then adding this scaling factor restores the benchmark scores.
Isn’t this trivial since this parameter was estimated to transform a set of weights into the other? I get that the point of this analysis is to show that near-uniform is enough, but could this be shown by efficiently grid-searching/optimizing the scaling factor?

* A crucial control which seems unexplored is whether any fine-tuning (fine-tune a LLM to better know capital cities) would result in such structural changes, or whether this is specific to instruction/reasoning.

**Questions:**

* Is there any distinguishable feature in the orthogonal transformations that can distinguish instruct vs reasoning models?

* Is there any regularity in the orthogonal transformations across layers and parameter types? Is the space equally rotated, when does it rotate more with respect to the base model? Is it possible to recreate something like Figure 2?

---

> ### Author Response · Authors · 2025-11-19
> **Response [1/2]**
>
> Thank you for your engaging review and the thoughtful questions you have raised, which encourage a deeper discussion of our work. We respond to each of your points below to clarify our perspective and explore these important ideas.
>
> ---
>
> >***W1**: The results in Table 1 are a bit circular, first a scaling factor is estimated, then adding this scaling factor restores the benchmark scores. Isn’t this trivial since this parameter was estimated to transform a set of weights into the other?*
>
> Your point is highly important. We clarify that the purpose of the experiments related to Table 1 **is not to search for an optimal scaling factor to restore or improve model performance, but rather to validate the correctness of Eq 8.** That is, to demonstrate that the singular value distribution of the post models can be reconstructed using the scaling factor $\alpha$ and the singular value distribution of the base models.
>
> Consider the following scenario: if post-training significantly alters the singular value distribution of the base models, it would be fundamentally impossible to reconstruct the singular value distribution of the post-training models using only a scaling factor *(A more concise mathematical example: we cannot use a scaling factor multiplied by a uniform distribution to reconstruct a normal distribution)*. Moreover, in such a case, it would be impossible for the model before and after applying Construction 9 to achieve similar performance, since different singular value distributions would drastically alter the entire parameter distribution of $M_{\text{post}}$, inevitably affecting its performance and potentially causing model collapse.
>
> Therefore, **we must control for other variables that could influence the experimental outcomes to verify the hypothesis that the singular value distributions of $M_{\text{post}}$ and $M_{\text{base}}$ are nearly identical.** We use the estimated $\alpha$ to ensure that this variable remains consistent, preventing any discrepancy in the scaling factor from interfering with the validation of our hypothesis. Only when the scaling factor is held constant can we focus the experimental analysis on differences in the singular value distributions, thereby more rigorously confirming the correctness of Eq 8.
>
> ---
>
> >***W2**: A crucial control which seems unexplored is whether any fine-tuning (fine-tune a LLM to better know capital cities) would result in such structural changes, or whether this is specific to instruction/reasoning.*
>
> We thank you for raising this important question. While our study primarily focuses on *instruction tuning* and *Long-CoT distillation*, **we believe the structural changes we observed represent a more general and fundamental mechanism in *post-training*, rather than being specific to certain tasks**.
>
> We have included a preliminary experiment (see updated Appendix E.4), in which *Qwen-Math-1.5B* was fully fine-tuned for 5 epochs on the *s1K-1.1* dataset covering complex tasks, and structural changes in the checkpoints after each epoch were examined. The observed *coordinated rotation* and *uniform scaling of singular values* throughout training demonstrate that the two phenomena described in our paper are indeed universal.
>
> We argue that domain-specific fine-tuning of LLMs is essentially a form of supervised fine-tuning on either *out-of-distribution* (OOD) or *in-distribution* (ID) data. *Long-CoT distillation* is distinctly OOD in nature. It involves distilling long-chain reasoning sequences from large RL-trained models, which are often from general domains and far exceed the complexity of pre-training data, into smaller models (e.g., *DeepSeek-R1-Distill-Qwen-1.5B* distills general *Long-CoT* data into *Qwen2.5-Math-1.5B*, which is pre-trained only on mathematical datasets). The fact that even such strong OOD training does not violate the general patterns we identified suggests that simpler tasks (e.g., memorizing capitals) are even less likely to deviate from these patterns.
>
> ---

---

> ### Author Response · Authors · 2025-11-19
> **Response [2/2]**
>
> ---
>
> >***Q1**: Is there any distinguishable feature in the orthogonal transformations that can distinguish instruct vs reasoning models?*
>
> This question is insightful. **We agree that there must exist orthogonal transformation features capable of distinguishing Instruct models from reasoning models.**
>
> The key finding of our work is that post-training essentially performs a set of consistent orthogonal transformations within the input and output subspaces of weight matrices, and such rotational patterns are **necessarily induced by the training data distribution during post-training**. Given the significant divergence between the data distributions of instruction-following tasks and complex reasoning tasks, they inevitably induce distinct orthogonal transformation patterns in the parameter space, thereby shaping the models' different external behaviors.
>
> However, precisely characterizing and quantifying the differences between these two types of orthogonal transformation patterns. For instance, identifying which rotational trajectories in the subspace are unique to instruction models or reasoning models, remains an important and challenging new research direction. We believe this will serve as a promising starting point for future work.
>
> ---
>
> >***Q2**: Is there any regularity in the orthogonal transformations across layers and parameter types? Is the space equally rotated, when does it rotate more with respect to the base model? Is it possible to recreate something like Figure 2?*
>
> This is indeed a crucial and insightful question, which aligns with important directions we are currently exploring.
>
> Based on our observations, we can only draw some preliminary conclusions: although the rotations of the subspaces of different weight matrices remain coordinated, the specific forms of rotation for individual matrices vary considerably, even among matrices of the same type (e.g., the differences in the similarity matrices of the first/last layers shown in Fig 10–13).
>
> We argue that such rich inconsistency in rotations is reasonable. Since the core of post-training lies in coordinated rotation, the directions of these subspace rotations naturally **encode the data distribution** encountered during post-training. This results in a more diverse set of coordinated rotation patterns. If all rotations were identical, the variety of rotational combinations available to fit the data distribution would be reduced, potentially causing LLMs to underfit on new training data, as the rotational patterns would fail to capture the richness of the training distribution.
>
> From this perspective, the more diverse the training data distribution, the greater the extent of rotation, and the more rotational patterns available to encode the data.
>
> We believe this idea may offer an interesting entry point for studying data distributions: it is generally challenging to quantify the diversity of a training dataset, however, if such diversity could be encoded and explicitly represented by the extent of rotation across various matrices in a well-trained model (i.e., an “LLM trainer” fully trained on that data), it might provide a useful quantitative measure for assessing data diversity. This could also be a promising direction for future research.
>
> ---
>
> Thank you once more for your time and the valuable comments that have enhanced our manuscript. We are happy to provide any additional information if required.

---

### Official Review · Reviewer_CkKS · 2025-11-01

**Soundness:** 4
**Presentation:** 3
**Contribution:** 3
**Rating:** 8
**Confidence:** 3

**Summary:**

- The paper investigates how post-training methods, specifically instruction tuning and Long-CoT distillation, affect the internal parameter structure of large language models (LLMs).
- Uses SVD to analyze principal linear layers in the attention and feedforward components to compare pretrained and post-trained models.
- The authors identify two consistent structural effects of post-training: Near-uniform geometric scaling of singular values across layers, and consistent orthogonal transformations on the left and right singular vectors, reflecting coordinated rotations of subspaces.
- They also demonstrate that disrupting these orthogonal transformations can cause severe performance degradation, indicating their functional importance.
- Their experiments mainly cover models based on Qwen2.5-Math-1.5B, as well as additional models such as Qwen2.5-Math-7B, Llama-3.1-8B, Qwen2.5-14B, etc in the appendix, while evaluating on GSM8K, MAtH-500, MMLU, and GPQA

**Strengths:**

- Presents an interesting and original perspective by analyzing post-training effects through the structural properties of model weights directly.
- Provides clear and consistent empirical findings: the identification of two structural phenomena, singular value scaling and orthogonal vector rotation, across diverse models and post-training methods highlights strong regularity and robustness.
- The investigation is thorough and rigorous, supported by extensive experiments spanning multiple model families and post-training scenarios.
- Includes a discussion on potential applications and implications in the appendix, adding useful context for future exploration.

**Weaknesses:**

- While the paper identifies consistent orthogonal transformations as a key structural component of post-training, it would benefit from a deeper discussion or empirical analysis of how these transformations influence model outputs or distributional behavior. Understanding this connection could strengthen the interpretability and practical relevance of the findings.

**Questions:**

- Is there any insight on how the other parameters of the model (eg. normalization layers and output projection head) change as a result of post-training?
- Is there evidence that these structural patterns persist throughout training, or do they emerge only after convergence of post-training?

---

> ### Author Response · Authors · 2025-11-19
> **Response [1/3]**
>
> We sincerely appreciate your recognition of our work and thank you for your constructive suggestions! We believe that addressing your suggestions will substantially improve our work.
>
> ---
>
> >***W1**: It would benefit from a deeper discussion or empirical analysis of how these transformations influence model outputs or distributional behavior. Understanding this connection could strengthen the interpretability and practical relevance of the findings.*
>
> This point is highly insightful. While our study does not directly examine the effect of consistent orthogonal transformations on data distributions, we can provide an indirect analysis related to them.
>
> 1. One important corollary we derive is the discovery of **the equivalent effect induced by post-training data from different distributions on co-rotation**. For different post-training data, assuming different data distributions induce distinct orthogonal transformations in the parameter subspaces, these orthogonal transformations should be convertible via the same set of orthogonal transformations. From this perspective, if a model undergoes post-training with one dataset and is then exposed to another dataset, the original orthogonal transformation may be overwritten by a new one, thereby replacing the parametric characteristics obtained under the initial post-training. We believe this corollary may offer a novel parametric perspective on *catastrophic forgetting*. We have added relevant descriptions in Lines 475-480.
>
> 2. Another key relevance is framing post-training as a reparameterization process of the subspaces formed in the pre-training space. That is, **regardless of the data distribution used in post-training, the structure of the subspaces formed during pre-training remains intact**. Our experimental results show that $Q^{(i)}_2$ can approximate the same orthogonal matrix as $Q^{(i)}_1$, which further rigorously demonstrates that the subspaces generated by post-training shares nearly identical geometric structure with the output subspaces of the base model (i.e., $\mathrm{col}(U_A) = \mathrm{col}(U_B)$, $\mathrm{col}(V_A) = \mathrm{col}(V_B)$). This strongly implies that post-training approximately preserves the original input/output subspaces of LLMs, reinforcing the reparameterization insight and validating the correctness of Eq 8. To strengthen our explanation, we have incorporated more rigorous mathematical constraints and descriptions in the revised version (Lines 268--284) and added a detailed mathematical proof to elaborate the "subspaces reparameterization" insight (see Appendix G.2 of the revised manuscript).
>
> Although these findings and arguments are preliminary, we firmly believe they can offer valuable guidance for future research. We thank you again for your highly insightful question. All revisions have been highlighted in blue.
>
> ---

---

> > ### Author Response · Authors · 2025-11-19
> > **Response [2/3]**
> >
> > ---
> >
> > >***Q1**: Is there any insight on how the other parameters of the model (eg. normalization layers and output projection head) change as a result of post-training?*
> >
> > Thank you for your comment. We have added Appendix E.3 and included additional experiments to further investigate the results. In summary, we observe that these components largely follow the same parameter scaling law.
> >
> > 1.  **For normalization layers**, we provide a theoretical analysis: since its parameters form a vector, applying reduced SVD to the normalization layer causes the left singular vector to reduce to $1$, the singular value to reduce to the vector norm, and the right singular vector to reduce to a unit vector. If Eq 8 holds, this implies that the cosine similarity "matrix" formed by the right singular vectors should also equal $1$. We have empirically verified this intriguing phenomenon. However, we note fluctuations in the scaling of their "singular values" (i.e., the norms). We hypothesize that this relates to the function of normalization layers, which dynamically adjust the expressive capacity of hidden vectors. When the subspace is fixed, the layer can only modulate overall magnitude scaling to fulfill its role, making it difficult for the "singular values" to maintain uniform scaling across layers.
> >
> > 2.  **For the output projection head**, we observe weaker co-rotation in some of its subspaces. We speculate that this is tied to its function. Since this layer maps hidden vectors to the vocabulary space, it is directly influenced by external supervision signals, whereas other main linear layers propagate information through hidden representations. As a result, post-training perturbations affect the output projection head more significantly than the main linear layers, making some of its internal subspaces more susceptible to reshaping by external signals. This, in turn, inhibits strong co-rotation. Nevertheless, due to the limited scale of post-training, most subspace structures remain largely preserved, allowing the output projection head to largely maintain the co-rotation phenomenon.
> >
> > Although these preliminary observations reveal interesting and exciting phenomena, a deeper understanding of these components remains an important and open direction for future research. We hope that our additional explanations can contribute to a more thorough understanding of the parameter space of these components. All supplementary experiments can be found in the updated Appendix E.3.
> >
> > ---

---

> ### Author Response · Authors · 2025-11-19
> **Response [3/3]**
>
> ---
>
> >***Q2**: Is there evidence that these structural patterns persist throughout training, or do they emerge only after convergence of post-training?*
>
> **We argue that this structural pattern may persist throughout the entire training process**. We have supplemented a preliminary experiment: using *Qwen2.5-Math-1.5B* as the base model, we performed full fine-tuning for 5 epochs on the complex dataset *s1K-1.1*, and visualized the $NF$ and SVSMs between intermediate models and the base model every other epoch. It can be observed that the $NF$ remains consistently low, while the SVSMs remains almost unchanged during post-training. This preliminary experiment confirms that co-rotation exists throughout the training process.
>
> From this perspective, model convergence may also correspond to the convergence of co-rotation: the training process can be viewed as the pretrained subspace seeking an optimal rotation pattern that minimizes the discrepancy between its output and the distribution of the training data. We have added the relevant experiments in Appendix E.4 to provide a more comprehensive understanding of this phenomenon.
>
> ---
>
> Once again, we sincerely appreciate your comments, which have significantly improved the quality and clarity of our work. Please do not hesitate to contact us if further clarifications are needed.

---

### Author Response · Authors · 2025-11-27

Dear Reviewers,

Thank you once again for your thoughtful feedback on our manuscript. We have carefully reviewed all comments and revised the submission accordingly.

With the discussion period closing in less than a week, we wanted to gently follow up to see if you have any further questions or need clarification on any of the revisions.

We truly appreciate the time and effort you have dedicated to the review process. Your insights have been invaluable, and we welcome any final comments you may have before the deadline.

We look forward to your response.

Best regards,

The Authors of Paper #12861

---

### Author Response · Authors · 2025-11-29
**Rebuttal Summary**

Dear AC,

We sincerely appreciate your tremendous efforts in reviewing our paper under the special circumstances of ICLR. To save your time and workload, we provide a concise summary of our rebuttal for your reference during the evaluation process.

**Reviewer-identified strengths of our paper:**

- Our paper characterizes two robust structural phenomena in the parameter space of LLMs: *a near-uniform geometric scaling of singular values across layers* and *highly consistent orthogonal transformations of the singular vectors*. We further present an intuitive mathematical framework that offers a clear perspective for understanding the impact of post-training on the parameter space. The reviewers generally acknowledged that the perspectives raised in the paper are both clear and novel (`CkKS`, `Gn1W`, `DYuD`, `1Wyf`, `YWpN`).
- The reviewers noted that our experiments are comprehensive, systematic and extensive, sufficiently validating the universality of the discovered patterns and the correctness of the established mathematical framework (`CkKS`, `DYuD`, `1Wyf`). Ablation studies further provide compelling evidence for the critical role of these structural phenomena (`DYuD`).
- The reviewers considered our findings to be of significant theoretical importance for understanding the learning dynamics of post-training (`Gn1W`), and to offer clear practical implications for simplifying and guiding future post-training techniques (`CkKS`, `Gn1W`).

**Response to Reviewer `CkKS`:**
- We have supplemented the discussion on data distribution and variation patterns in **W1** with more rigorous theoretical deductions and rigorous proofs.
- Supplementary experiments have been conducted to address **Q1** regarding the variation patterns of parameters such as normalization layers.
- Supplementary experiments confirm the persistence of structural patterns raised in **Q2**.

**Response to Reviewer `Gn1W`:**
- We have elaborated on the logic of our experimental design to clarify the misunderstanding of *"circular reasoning"* in **W1**.
- Supplementary experiments validate the universality of structural changes concerned in **W2**.
- We acknowledge the theoretical possibility raised in **Q1** regarding distinguishing orthogonal transformation features between instruct models and reasoning models.
- The issue of rotational patterns raised in **Q2** has been discussed.

**Response to Reviewer `DYuD`:**
- We clarified in **W1** that the core contribution of our work lies in the robust discovery and validation of fundamental patterns, rather than direct performance improvement.
- In **W2**, we elaborated that our research focuses on investigating internal patterns of LLMs and establishing a mathematical framework Eq 8 (rather than algorithm design), while justifying the potential applicability.
- The misunderstanding in **W3** was addressed by providing a detailed analysis of output token counts, reaffirming that singular value "temperature modulation" is a secondary effect.

**Response to Reviewer `1Wyf`:**
- We clarified the misunderstanding in **W1** that *"structural changes may be calibration artifacts"* by providing experimental evidence and theoretical explanations.
- In **W2**, we emphasized that our core contribution is the robust discovery and rigorous validation of a fundamental phenomenon, laying a solid foundation for future theoretical developments.
- We clarified in **W3** that the focus of this work is on providing fundamental discoveries and rigorous experimental validation, rather than algorithm design. *(Note: W2 and W3 of Reviewer `1Wyf` are similar to W1 and W2 of Reviewer `DYuD`.)*

**Response to Reviewer `YWpN`:**
- Controlled experiments were added to **W1** to strengthen the robustness of the conclusions in Figure 3b.
- We adopted the suggestion in **W2** by improving the visualization of tables.

**All supplementary experiments and revisions have been incorporated into the revised manuscript and highlighted in blue.** We thank all reviewers for their valuable contributions to improving our paper, and express our gratitude again for your dedicated efforts during this exceptional period. We hope this summary assists in your assessment.

Best regards,

The Authors of Paper #12861

---

### Meta-Review · Area_Chair_1TJf · 2026-01-04

**Summary:**

The paper received two scores marginally below acceptance, one marginally above, and two “accept” ratings. Reviewers raised several concerns, including the depth of analysis, practical applicability, weak motivation for the Singular Value Scaling Argument, and a general lack of insight. Notably, both reviewers who gave an “8 – accept” did not provide substantive justification for their scores. In particular, the comments from reviewer YWpN, who rated the paper as 8, are extremely brief and should be given limited weight in the decision. Overall, the weaknesses outweigh the strengths of the paper. After reviewing the authors’ responses, many of the issues raised by the reviewers remain unclear. The paper requires improvement before it can be considered for acceptance.

**Reviewer Concerns:**

Several concerns—including the depth of analysis, practical applicability, weak motivation for the Singular Value Scaling Argument, and a general lack of insight—have not been adequately addressed.

**Reviewer Scores:**

Notably, both reviewers who gave an “8 – accept” did not provide substantive justification for their scores. In particular, the comments from reviewer YWpN, who rated the paper as 8, are extremely brief and should be given limited weight in the decision. Based on the rebuttal, it is unlikely that the reviewers will revise their scores to positive ratings.

---

### Decision · Program_Chairs · 2026-01-26

Reject